# Truly Deterministic Policy Optimization

**Ehsan Saleh[1], Saba Ghaffari[1], Timothy Bretl[1,2], Matthew West[3]**
[1]Department of Computer Science
[2]Department of Aerospace Engineering
[3]Department of Mechanical Science and Engineering
University of Illinois Urbana-Champaign
ehsans2,sabag2,tbretl,mwest@illinois.edu

## Abstract

In this paper, we present a policy gradient method that avoids exploratory noise injection and performs policy search over the deterministic landscape, with the goal of improving learning with long horizons and non-local rewards. By avoiding noise injection all sources of estimation variance can be eliminated in systems with deterministic dynamics (up to the initial state distribution). Since deterministic policy regularization is impossible using traditional non-metric measures such as the KL divergence, we derive a Wasserstein-based quadratic model for our purposes. We state conditions on the system model under which it is possible to establish a monotonic policy improvement guarantee, propose a surrogate function for policy gradient estimation, and show that it is possible to compute exact advantage estimates if both the state transition model and the policy are deterministic. Finally, we describe two novel robotic control environments—one with non-local rewards in the frequency domain and the other with a long horizon (8000 timesteps)—for which our policy gradient method (TDPO) significantly outperforms existing methods (PPO, TRPO, DDPG, and TD3). Our implementation with all the experimental settings and a video of the physical hardware test is available at https://github.com/ehsansaleh/tdpo.

Policy Gradient (PG) methods can be broadly characterized by three defining elements: the policy gradient estimator, the regularization measures, and the exploration profile. For gradient estimation, episodic [58], importance-sampling-based [50], and deterministic [54] gradients are some of the most common estimation oracles. As for regularization measures, either an Euclidean distance within the parameter space [58, 54, 35], or dimensionally consistent non-metric measures [50, 25, 52, 26, 59] have been frequently adapted. Common exploration profiles include Gaussian [50] and stochastic processes [35]. These elements form the basis of many model-free and stochastic policy optimization methods successfully capable of learning high-dimensional policy parameters.

Both stochastic and deterministic policy search can be useful in applications. A stochastic policy has the effect of smoothing or filtering the policy landscape, which is desirable for optimization. Searching through stochastic policies has enabled the effective control of challenging environments under a general framework [50, 52]. The same method could either learn robotic movements or play basic games (1) with minimal domain-specific knowledge, (2) regardless of function approximation classes, and (3) with less human intervention (ignoring reward engineering and hyper-parameter tuning) [13]. Using stochasticity for exploration, although it imposes approximations and variance, has provided a robust way to actively search for higher rewards. Despite many successes, there are practical environments which remain challenging for current policy gradient methods. For example, non-local rewards (e.g., those defined in the frequency domain), long time horizons, and naturally-resonant environments all occur in realistic robotic systems [34, 38, 47] but can present issues for policy gradient search.

36th Conference on Neural Information Processing Systems (NeurIPS 2022).

To tackle challenging environments such as these, this paper considers policy gradient methods based on deterministic policies and deterministic gradient estimates, which could offer advantages by allowing the estimation of global reward gradients on long horizons without the need to inject noise into the system for exploration. To facilitate a dimensionally consistent and low-variance deterministic policy search, a compatible policy gradient estimator and a metric measure for regularization should be employed. For gradient estimation we focus on Vine estimators [50], which can be easily applied to deterministic policies. As a metric measure, we use the Wasserstein distance, which can measure meaningful distances between deterministic policy functions that have non-overlapping supports (in contrast to the Kullback-Liebler (KL) divergence and the Total Variation (TV) distance).

The Wasserstein metric has seen substantial recent application in a variety of machine-learning domains, such as the successful stable learning of generative adversarial models [4]. Theoretically, this metric has been studied in the context of Lipschitz-continuous Markov decision processes in reinforcement learning [21, 14]. Pirotta et al. [45] defined a policy gradient method using the Wasserestein distance by relying on Lipschitz continuity assumptions with respect to the policy gradient itself. Asadi et al. [5] and Rachelson and Lagoudakis [48] used the Wasserstein distance to derive model-based value-iteration and policy-iteration methods, respectively. On a more practical note, Ciosek et al. [9] introduced a method which in the deterministic mode optimized policies using the Wasserstein distance. Pacchiano et al. [43] utilized Wasserstein regularization for behavior-guided stochastic policy optimization. Moreover, Abdullah et al. [1] has proposed another robust stochastic policy gradient formulation. Estimating the Wasserstein distance for general distributions is more complicated than typical KL-divergences [56]. This fact constitutes and emphasizes the contributions of Abdullah et al. [1] and Pacchiano et al. [43]. However, for our deterministic observation-conditional policies, closed-form computation of Wasserstein distances is possible without any approximation.

Existing deterministic policy gradient methods (e.g., DDPG and TD3) use *deterministic policies* [54, 35, 15], meaning that they learn a deterministic policy function from states to actions. However, such methods still use *stochastic search* (i.e., they add stochastic noise to their deterministic actions to force exploration during policy search). In contrast, we will be interested in a method which not only uses *deterministic policies*, but also uses *deterministic search* (i.e., without constant stochastic noise injection). We call this method *truly deterministic policy optimization* (TDPO) and it may have lower estimation variances and better scalability to long horizons, as we will show in numerical examples.

Scalability to long horizons is one of the most challenging aspects of policy gradient methods that use stochastic search. This issue is sometimes referred to as the *curse of horizon* in reinforcement learning [36]. General worst-case analyses suggest that the sample complexity of reinforcement learning is exponential with respect to the horizon length [27, 31, 30]. Deriving polynomial lower-bounds for the sample complexity of reinforcement learning methods is still an open problem [24]. Lower-bounding the sample complexity of reinforcement learning for long horizons under different settings and simplifying assumptions has been a topic of theoretical research [10, 57]. Some recent work has examined the scalability of importance sampling gradient estimators to long horizons in terms of both theoretical and practical estimator variances [36, 28, 29]. All in all, long horizons are challenging for all reinforcement learning methods, especially the ones suffering from excessive estimation variance due to the use of stochastic policies for exploration, and our truly deterministic method may have advantages in this respect.

In this paper, we focus on continuous-domain robotic environments with reset capability to previously visited states. The main contributions of this work are: (1) we introduce a Deterministic Vine (DeVine) policy gradient estimator which avoids constant exploratory noise injection; (2) we derive a novel deterministically-compatible surrogate function and provide monotonic payoff improvement guarantees; (3) we show how to use the DeVine policy gradient estimator with the Wasserstein-based surrogate in a practical algorithm (TDPO: Truly Deterministic Policy Optimization); (4) we illustrate the robustness of the TDPO policy search process in robotic control environments with non-local rewards, long horizons, and/or resonant frequencies.

# 1   Background

**MDP preliminaries.**   An infinite-horizon discounted Markov decision process (MDP) is specified by $(\mathcal{S}, \mathcal{A}, P, R, \mu, \gamma)$, where $\mathcal{S}$ is the state space, $\mathcal{A}$ is the action space, $P : \mathcal{S} \times \mathcal{A} \to \Delta(\mathcal{S})$ is the transition dynamics, $R : \mathcal{S} \times \mathcal{A} \to [0, R_{\max}]$ is the reward function, $\gamma \in [0, 1)$ is the discount factor,

and $\mu(s)$ is the initial state distribution of interest (where $\Delta(\mathcal{F})$ denotes the set of all probability distributions over $\mathcal{F}$, otherwise known as the Credal set of $\mathcal{F}$). The transition dynamics $P$ is defined as an operator which produces a distribution over the state space for the next state $s' \sim P(s, a)$. The transition dynamics can be easily generalized to take distributions of states or actions as input (i.e., by having $P$ defined as $P : \Delta(\mathcal{S}) \times \mathcal{A} \to \Delta(\mathcal{S})$ or $P : \mathcal{S} \times \Delta(\mathcal{A}) \to \Delta(\mathcal{S})$). We may abuse the notation and replace $\delta_s$ and $\delta_a$ by $s$ and $a$, where $\delta_s$ and $\delta_a$ are the deterministic distributions concentrated at the state $s$ and action $a$, respectively. A policy $\pi : \mathcal{S} \to \Delta(\mathcal{A})$ specifies a distribution over actions for each state, and induces trajectories from a given starting state $s$ as follows: $s_1 = s$, $a_1 \sim \pi(s_1)$, $r_1 = R(s_1, a_1)$, $s_2 \sim P(s_2, a_2)$, $a_2 \sim \pi(s_2)$, etc. We will denote trajectories as state-action tuples $\tau = (s_1, a_1, s_2, a_2, \ldots)$. One can generalize the dynamics (1) by using a policy instead of an action distribution $\mathbb{P}(\mu_s, \pi) := \mathbb{E}_{s \sim \mu_s}[\mathbb{E}_{a \sim \pi(s)}[P(s, a)]]$, and (2) by introducing the $t$-step transition dynamics recursively as $\mathbb{P}^t(\mu_s, \pi) := \mathbb{P}(\mathbb{P}^{t-1}(\mu_s, \pi), \pi)$ with $\mathbb{P}^0(\mu_s, \pi) := \mu_s$, where $\mu_s$ is a distribution over $\mathcal{S}$. The visitation frequency can be defined as $\rho_\mu^\pi := (1-\gamma) \sum_{t=1}^\infty \gamma^{t-1} \mathbb{P}^{t-1}(\mu, \pi)$. Table 2 of the Supplementary Material summarizes all MDP notation.

The value function of $\pi$ is defined as $V^\pi(s) := \mathbb{E}[\sum_{t=1}^\infty \gamma^{t-1} r_t \mid s_1 = s; \pi]$. Similarly, one can define $Q^\pi(s, a)$ by conditioning on the first action. The advantage function can then be defined as their difference (i.e. $A^\pi(s, a) := Q^\pi(s, a) - V^\pi(s)$). Generally, one can define the advantage/value of one policy with respect to another using $A^\pi(s, \pi') := \mathbb{E}[Q^\pi(s, a) - V^\pi(s) \mid a \sim \pi'(\cdot|s)]$ and $Q^\pi(s, \pi') := \mathbb{E}[Q^\pi(s, a) \mid a \sim \pi'(\cdot|s)]$. Finally, the payoff of a policy $\eta_\pi := \mathbb{E}[V^\pi(s); s \sim \mu]$ is the average value over the initial states distribution of the MDP.

**Probabilistic and mathematical notations.** Sometimes we refer to $\int f(x)g(x)dx$ integrals as $\langle f, g \rangle_x$ Hilbert space inner products. Assuming that $\zeta$ and $\nu$ are two probabilistic densities, the Kulback-Liebler (KL) divergence is $D_{\mathrm{KL}}(\zeta|\nu) := \langle \zeta(x), \log(\frac{\zeta(x)}{\nu(x)}) \rangle_x$, the Total-Variation (TV) distance is $\mathrm{TV}(\zeta, \nu) =: \frac{1}{2} \langle |\zeta(x) - \nu(x)|, 1 \rangle_x$, and the Wasserstein distance is $W(\zeta, \nu) = \inf_{\gamma \in \Gamma(\zeta, \nu)} \langle \|x - y\|, \gamma(x, y) \rangle_{x,y}$ where $\Gamma(\zeta, \nu)$ is the set of couplings for $\zeta$ and $\nu$. We define $\mathrm{Lip}(f(x, y); x) := \sup_x \|\nabla_x f(x, y)\|_2$ and assume the existence of $\mathrm{Lip}(Q^\pi(s, a); a)$ and $\mathrm{Lip}(\nabla_s Q^\pi(s, a); a)$ constants. Under this notation, the Rubinstein-Kantrovich (RK) duality states that the $|\langle \zeta(x) - \nu(x), f(x) \rangle_x| \leq W(\zeta, \nu) \cdot \mathrm{Lip}(f; x)$ bound is tight for all $f$. For brevity, we may abuse the notation and denote $\sup_s W(\pi_1(\cdot|s), \pi_2(\cdot|s))$ with $W(\pi_1, \pi_2)$ (and similarly for other measures). For parameterized policies, we define $\nabla_\pi f(\pi) := \nabla_\theta f(\pi)$ where $\pi$ is parameterized by the vector $\theta$. Table 1 of the Supplementary Material summarizes all these mathematical definitions.

**Policy gradient preliminaries.** The advantage decomposition lemma provides insight into the relationship between payoff improvements and advantages [25]. That is,

$$\eta_{\pi_2} - \eta_{\pi_1} = \frac{1}{1 - \gamma} \cdot \mathbb{E}_{s \sim \rho_\mu^{\pi_2}}[A^{\pi_1}(s, \pi_2)]. \tag{1}$$

We will denote the current and the candidate next policy as $\pi_1$ and $\pi_2$, respectively. Taking derivatives of both sides with respect to $\pi_2$ at $\pi_1$ yields

$$\nabla_{\pi_2} \eta_{\pi_2} = \frac{1}{1 - \gamma} \left[ \langle \nabla_{\pi_2} \rho_\mu^{\pi_2}(\cdot), A^{\pi_1}(\cdot, \pi_1) \rangle + \langle \rho_\mu^{\pi_1}(\cdot), \nabla_{\pi_2} A^{\pi_1}(\cdot, \pi_2) \rangle \right]. \tag{2}$$

Since $\pi_1$ does not have any advantage over itself (i.e., $A^{\pi_1}(\cdot, \pi_1) = 0$), the first term is zero. Thus, the Policy Gradient (PG) theorem is derived as

$$\nabla_{\pi_2} \eta_{\pi_2} \Big|_{\pi_2 = \pi_1} = \frac{1}{1 - \gamma} \cdot \mathbb{E}_{s \sim \rho_\mu^{\pi_1}}[\nabla_{\pi_2} A^{\pi_1}(s, \pi_2)] \Big|_{\pi_2 = \pi_1}. \tag{3}$$

For policy iteration with function approximation, we assume $\pi_2$ and $\pi_1$ to be parameterized by $\theta_2$ and $\theta_1$, respectively. One can view the PG theorem as a Taylor expansion of the payoff at $\theta_1$.

A brief introduction of the Conservative Policy Iteration (CPI) [25], the Trust Region Policy Optimization (TRPO) [50], the Proximal Policy Optimization (PPO) [51], the Deep Deterministic Policy Gradient (DDPG) [35], and the Twin-Delayed Deterministic Policy Gradient (TD3) [15] policy gradient methods is left to the Supplementary Material. Whether using deterministic policy gradients (e.g., DDPG and TD3) or stochastic policy gradients (e.g., TRPO and PPO), all these methods still perform stochastic search by adding stochastic noise to the deterministic policies to force exploration.

## 2 Monotonic Policy Improvement Guarantee

We use the Wasserstein metric because it allows the effective measurement of distances between probability distributions or functions with non-overlapping support, such as deterministic policies, unlike the KL divergence or TV distance which are either unbounded or maximal in this case. The physical transition model's smoothness enables the use of the Wasserstein distance to regularize deterministic policies. Therefore, we make the following two assumptions about the transition model:

$$W(\mathbb{P}(\mu, \pi_1), \mathbb{P}(\mu, \pi_2)) \leq L_\pi \cdot W(\pi_1, \pi_2), \tag{4}$$

$$W(\mathbb{P}(\mu_1, \pi), \mathbb{P}(\mu_2, \pi)) \leq L_\mu \cdot W(\mu_1, \mu_2). \tag{5}$$

Also, we make the dynamics stability assumption $\sup \sum_{l=1}^{t} \hat{L}_{\mu,\pi_1,\pi_2}^{(l-1)} \prod_{i=l+1}^{t-1} \tilde{L}_{\mu,\pi_1,\pi_2}^{(i)} < \infty$, with the definitions of the new constants and further discussion of the implications deferred to the Supplementary Material where we also discuss Assumptions 4 and 5 and the existence of other Lipschitz constants which appear as coefficients in the final lower bound.

The advantage decomposition lemma can be rewritten as

$$\eta_{\pi_2} = \eta_{\pi_1} + \frac{1}{1-\gamma} \cdot \mathbb{E}_{s \sim \rho_\mu^{\pi_1}}[A^{\pi_1}(s, \pi_2)] + \frac{1}{1-\gamma} \cdot \langle \rho_\mu^{\pi_2} - \rho_\mu^{\pi_1}, A^{\pi_1}(\cdot, \pi_2) \rangle_s. \tag{6}$$

The $\langle \rho_\mu^{\pi_2} - \rho_\mu^{\pi_1}, A^{\pi_1}(\cdot, \pi_2) \rangle$ term has zero gradient at $\pi_2 = \pi_1$, which qualifies it to be crudely called "the second-order term". We dedicate a full section of our Supplementary Material to the theoretical derivations and proofs necessary to lower-bound this second-order term into an objective in a form well-suited for practical optimization. Next, we present the final bound:

$$\mathcal{L}_{\pi_1}^{\sup}(\pi_2) = \eta_{\pi_1} + \frac{1}{1-\gamma} \mathbb{E}_{s \sim \rho_\mu^{\pi_1}}[A^{\pi_1}(s, \pi_2)] - C_2 \cdot \sup_s \left[ W(\pi_2(a|s), \pi_1(a|s))^2 \right]$$

$$- C_1 \cdot \sup_s \left[ \left\| \nabla_{s'} W \left( \frac{\pi_2(a|s') + \pi_1(a|s)}{2}, \frac{\pi_2(a|s) + \pi_1(a|s')}{2} \right) \Big|_{s'=s} \right\|_2^2 \right]. \tag{7}$$

For brevity, we denote the $\left\| \nabla_{s'} W(\cdots) \right|_{s'=s} \right\|_2^2$ expression as $\mathcal{L}_{G^2}(\pi_1, \pi_2; s)$ in the rest of the paper. We have $\eta_{\pi_2} \geq \mathcal{L}_{\pi_1}^{\sup}(\pi_2)$ and $\mathcal{L}_{\pi_1}^{\sup}(\pi_1) = \eta_{\pi_1}$. This facilitates the application of Theorem 2.1 as an instance of Minorization-Maximization algorithms [22].

**Theorem 2.1.** *Successive maximization of $\mathcal{L}^{\sup}$ yields non-decreasing policy payoffs.*

*Proof.* With $\pi_2 = \arg\max_\pi \mathcal{L}_{\pi_1}^{\sup}(\pi)$, we have $\mathcal{L}_{\pi_1}^{\sup}(\pi_2) \geq \mathcal{L}_{\pi_1}^{\sup}(\pi_1)$. Thus,

$$\eta_{\pi_2} \geq \mathcal{L}_{\pi_1}^{\sup}(\pi_2) \text{ and } \eta_{\pi_1} = \mathcal{L}_{\pi_1}^{\sup}(\pi_1) \implies \eta_{\pi_2} - \eta_{\pi_1} \geq \mathcal{L}_{\pi_1}^{\sup}(\pi_2) - \mathcal{L}_{\pi_1}^{\sup}(\pi_1) \geq 0. \quad \square \tag{8}$$

Successive optimization of $\mathcal{L}_{\pi_1}^{\sup}(\pi_2)$ generates non-decreasing payoffs. However, it is impractical due to the large number of "sup" terms that need to be expensively statistically estimated and implemented as constraints [50]. To mitigate this, we take a similar approach to TRPO and replace the maximums with expectations over the observations.

The Truly Deterministic Policy Optimization (TDPO) method is given in Algorithm 1. In the *basic variant*, the coefficients $C_1$ and $C_2$ are constant and a trust region is used. The coefficients $C_1$ and $C_2$ are dynamics-dependent and the Supplementary Material provides practical notes on their choice. In the *advanced variant*, we use a line search similar to Schulman et al. [50] and an adaptive tuning of the exploration scale using an importance sampling derivative estimate, as described in Algorithm 1.

We note that, for deterministic policies, the squared Wasserstein distance $W(\pi_2(a|s), \pi_1(a|s))^2$ degenerates to the Euclidean distance over the action space. Any policy defines a sensitivity matrix at a given state $s$, which is the Jacobian matrix of the policy output with respect to $s$. The policy sensitivity term $\mathcal{L}_{G^2}(\pi_1, \pi_2; s)$ is essentially the squared Euclidean distance over the action-to-observation Jacobian matrix elements. In other words, our surrogate prefers to step in directions where the action-to-observation sensitivity is preserved within updates.

## 3 Model-Free Estimation of Policy Gradient

The DeVine advantage estimator is formally defined in Algorithm 2. Unlike DDPG and TD3, the DeVine estimator allows our method to perform *deterministic search* by not consistently injecting

---

**Algorithm 1** Truly Deterministic Policy Optimization (TDPO)

---

**Require:** Initial policy $\pi_0$ and exploration scale $\sigma$.
**Require:** Advantage estimator and sample collector oracle $\mathbb{A}^\pi$ of Algorithm 2.
 1: **for** $i = 0, 1, 2, \ldots$ **do**
 2:     Collect trajectories and construct $\mathbb{A}^{\pi_i}$ using Algorithm 2.
 3:     Compute the policy gradient $g$ at $\theta_i$ : $g \leftarrow \nabla_{\theta'} \mathbb{A}^{\pi_i}(\pi')|_{\pi'=\pi_i}$
 4:     Construct a surrogate Hessian vector product oracle $v \to H \cdot v$ such that for $\theta' = \theta_i + \delta\theta$,

$$\mathbb{E}_{s \sim \rho_\mu^{\pi_i}} \left[ W(\pi'(a|s), \pi_i(a|s))^2 \right] + \frac{C_1}{C_2} \mathbb{E}_{s \sim \rho_\mu^{\pi_i}} \left[ \mathcal{L}_{G^2}(\pi', \pi_i; s) \right] = \frac{1}{2} \delta\theta^T H \delta\theta + \text{h.o.t.}, \quad (9)$$

where h.o.t. denotes higher order terms in $\delta\theta$.
 5:     Find the optimal update direction $\delta\theta^* = H^{-1}g$ using the Conjugate Gradient algorithm.
 6:     **(Basic Variant)** Determine the best step size $\alpha^*$ within the trust region:

$$\alpha^* = \arg\max_\alpha g^T(\alpha\delta\theta^*) - \frac{C_2}{2}(\alpha\delta\theta^*)^T H(\alpha\delta\theta^*)$$

$$\text{s.t.} \quad \frac{1}{2}(\alpha^*\delta\theta^*)^T H(\alpha^*\delta\theta^*) \le \delta_{\max}^2 \quad (10)$$

 7:     **(Advanced Variant)** Determine the best step size $\alpha^*$ using a line-search procedure and pick the best one; each coefficient's performance can be evaluated by sampling from the environment.
 8:     **(Advanced Variant)** Update the exploration scale $\sigma$ in $\mathbb{A}^\pi$ using the collected samples.
 9:     Update the policy parameters: $\theta_{i+1} \leftarrow \theta_i + \alpha^*\delta\theta^*$.
10: **end for**

---

**Algorithm 2** Deterministic Vine (DeVine) Policy Advantage Estimator

---

**Require:** The number of workers $K$, policy $\pi$, initial state distribution $\mu(s)$, and discount factor $\gamma$.
**Require:** An exploration index set distribution $\nu$, exploration scale $\sigma$, and maximal horizon $H$.
 1: Sample an initial state $s_0$ from $\mu$, and then roll out a trajectory $\tau = (s_0, a_0, s_1, a_1, \cdots)$ using $\pi$.
 2: Sample the exploration indices set $\mathcal{X}_K := \{(t_1, j_1), (t_2, j_2), \cdots, (t_K, j_K)\}$ from $\nu$.
 3: **for** $k = 1, 2, \cdots, K$ **do**
 4:     Compute the value $V^{\pi_1}(s_{t_k}) = \sum_{i=t_k}^\infty \gamma^{t_k-i} R(s_i, a_i)$.
 5:     Reset the initial state to $s_{t_k}$, set $a'_{t_k} := \pi(s_{t_k}) + \sigma \cdot \mathbf{e}_{j_k}$, and use $\pi$ for the rest of the trajectory, with $\mathbf{e}_j$ being the $j^{th}$ basis element for $\mathcal{A}$. This will create $\tau' = (s_{t_k}, a'_{t_k}, s'_{t_k+1}, a'_{t_k+1}, \cdots)$.
 6:     Compute the value $Q^{\pi_1}(s_{t_k}, a'_{t_k}) = \sum_{i=t_k}^\infty \gamma^{t_k-i} R(s'_i, a'_i)$.
 7:     Compute the advantage $A^{\pi_1}(s_{t_k}, a'_{t_k}) = Q^\pi(s_{t_k}, a'_{t_k}) - V^\pi(s_{t_k})$.
 8: **end for**
 9: Define $\mathbb{A}^{\pi_1}(\pi_2) := \frac{1}{K} \sum_{k=1}^K \frac{\dim(\mathcal{A}) \cdot H \cdot \gamma^{t_k}}{\nu(\mathcal{X}_K)} \cdot \frac{(\pi_2(s_{t_k}) - a_{t_k})^T (a'_{t_k} - a_{t_k})}{(a'_{t_k} - a_{t_k})^T (a'_{t_k} - a_{t_k})} \cdot A^{\pi_1}(s_{t_k}, a'_{t_k})$.
10: Return $\mathbb{A}^{\pi_1}(\pi_2)$ and $\nabla_{\pi_2}\mathbb{A}^{\pi_1}(\pi_2)$ as unbiased estimators for $E_{s \sim \rho_\mu^{\pi_1}}[A^{\pi_1}(s, \pi_2)]$ and the PG.

---

noise in actions for exploration. Algorithm 2 uses an exploration index sampler $\nu$, which samples a set of time-steps and action dimensions for exploration perturbation. The truly deterministic version of TDPO uses the deterministic $\nu_{\text{det}}$ which always returns the complete covering of $\{1, \cdots, \dim(\mathcal{A})\} \times \{1, \cdots, H\}$. Using $\nu_{\text{det}}$, DeVine produces exact policy gradients in the limit of small $\sigma$ as stated in Theorem 3.1, whose proof is deferred to the Supplementary Material.

**Theorem 3.1.** *Assume a finite horizon MDP with both deterministic transition dynamics $P$ and initial distribution $\mu$, with maximal horizon length of $H$. Define $K = H \cdot \dim(\mathcal{A})$ and $\nu := \nu_{det}$, where $\nu_{det}$ always returns the complete covering of $\{1, \cdots, \dim(\mathcal{A})\} \times \{1, \cdots, H\}$. Then we have*

$$\lim_{\sigma \to 0} \nabla_{\pi_2}\mathbb{A}^{\pi_1}(\pi_2)\big|_{\pi_2=\pi_1} = \nabla_{\pi_2}\eta_{\pi_2}\big|_{\pi_2=\pi_1}. \quad (11)$$

Although this theorem sets the stage for computing a fully deterministic gradient, stochastic approximation can be used in Algorithm 2 by randomly sampling a small set of states from for advantage

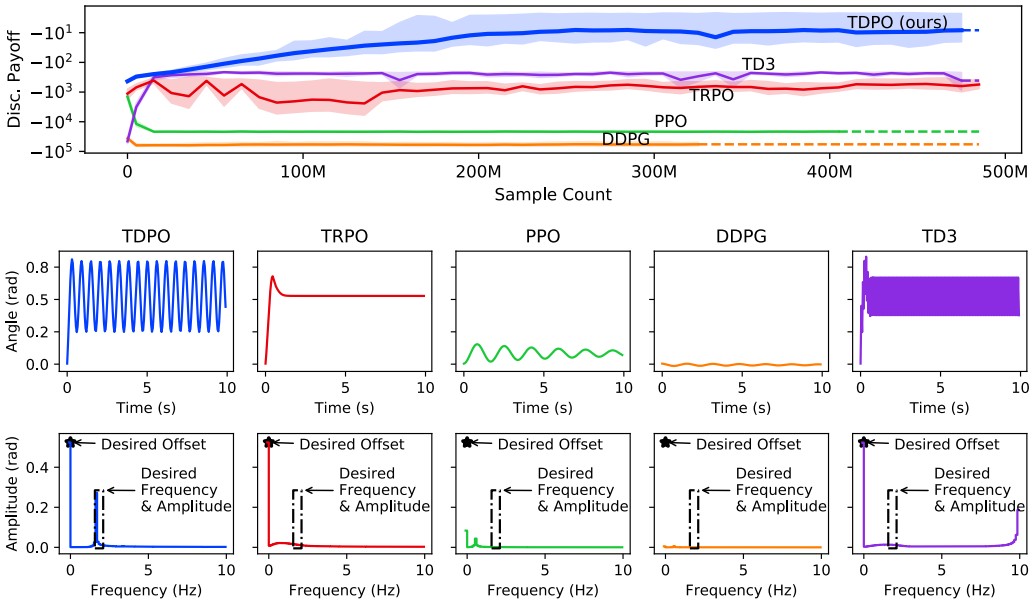

Figure 1: Results for the simple pendulum with non-local rewards. Upper panel: training curves with empirical discounted payoffs. Lower panels: trajectories in both the time domain and frequency domain, showing target values of oscillation frequency, amplitude, and offset. The basic variant of our method (non-adaptive exploration scales and update coefficients) was used in this experiment. The initial agent payoffs indicate the performance after the first epoch.

estimation. In other words, Theorem 3.1 would use $\nu$ to deterministically sample all trajectory states, whereas this is not a practical requirement for Algorithm 2 and the gradients are still unbiased if a random set of vine branches is used.

The DeVine estimator can be advantageous in at least two scenarios. First, in the case of rewards that cannot be decomposed into summations of immediate rewards. For example, overshoot penalizations or frequency-based rewards as used in robotic systems are non-local. DeVine can be robust to non-local rewards as it is insensitive to whether the rewards were applied immediately or after a long period. Second, DeVine can be an appropriate choice for systems that are sensitive to the injection of noise, such as high-bandwidth robots with natural resonant frequencies. In such cases, using white (or colored) noise for exploration can excite these resonant frequencies and cause instability, making learning difficult. DeVine avoids the need for constant noise injection.

# 4    Experiments

The next three subsections show challenging robotic control tasks including frequency-based non-local rewards, long horizons, and sensitivity to resonant frequencies. In Sections 4.1 and 4.2, we use the basic variant of our method (i.e., fixed exploration scale and update coefficient hyper-parameters throughout the training). This will facilitate a better understanding of our core method's capabilities without any additional tweaks. See the Supplementary Material for a comparison on traditional gym environments, where the basic variant of TDPO works similarly to existing methods. Section 4.3 includes the most difficult setting in our paper, where we use the advanced variant of our method (i.e., with line-search the update coefficient and adaptive exploration scales).

## 4.1    An Environment with Non-Local Rewards [1]

The first environment that we consider is a simple pendulum. The transition function is standard—the states are joint angle and joint velocity, and the action is joint torque. The reward function is non-

---

[1]Non-local rewards are reward functions of the entire trajectory whose payoffs cannot be decomposed into the sum of terms such as $\eta = \sum_t f_t(s_t, a_t)$, where functions $f_t$ only depend on nearby states and actions. An example non-local reward is one that depends on the Fourier transform of the complete trajectory signal.

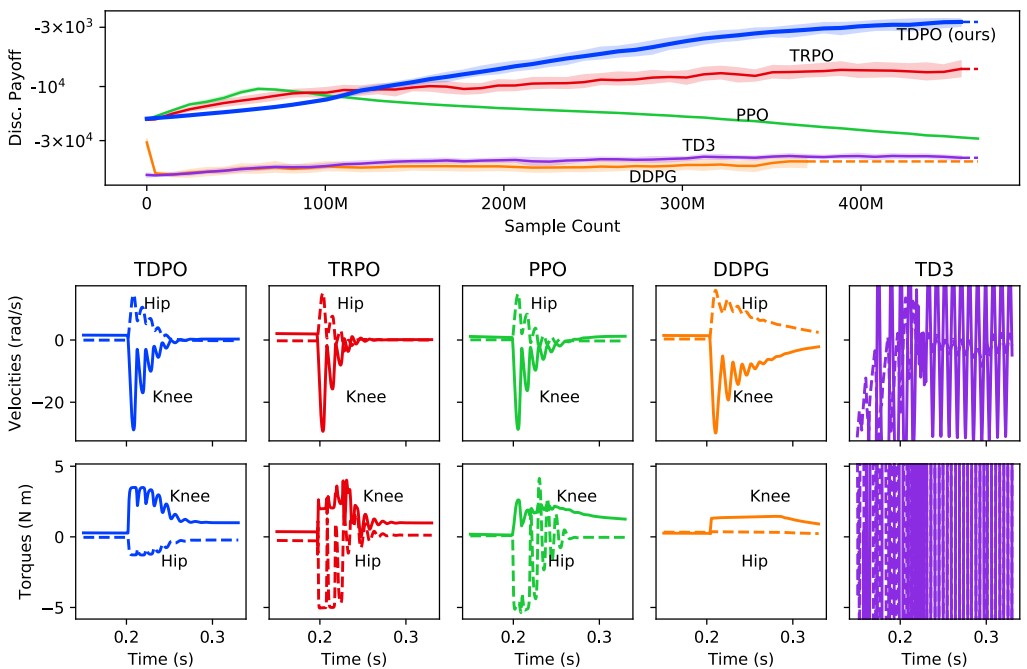

Figure 2: Results for the leg environment with a long horizon and resonant frequencies due to ground compliance. Upper panel: training curves with empirical discounted payoffs. Lower panel: partial trajectories, restricted to times shortly before and after impact with the ground. Note the oscillations at about 100 Hz that appear just after the impact at 0.2 s—these oscillations are evidence of a resonant frequency. The basic variant of our method (non-adaptive exploration scales and update coefficients) was used in this experiment.

standard—rather than define a local reward in the time domain with the goal of making the pendulum stand upright (for example), we define a non-local reward in the frequency domain with the goal of making the pendulum oscillate with a desired frequency and amplitude about a desired offset. In particular, we compute this non-local reward by taking the Fourier transform of the joint angle signal over the entire trajectory and by penalizing differences between the resulting power spectrum and a desired power spectrum. We apply this non-local reward at the last time-step of the trajectory. All methods here only used the current state of the systems, despite the fact that these environments are in all cases partially observable or history-dependent. Nevertheless, the agents are able to achieve high-reward behaviors, since these environments have weak history dependence. Note that this environment is a toy-problem to illustrate frequency dependence, and may as such be solved using Wavelet or short-term Fourier transformations in conjunction with the existing PG methods. However, our focus is on the representative features of this example, rather than this particular problem itself. Implementation details and similar results for more variants are left to the Supplementary Material.

Figure 1 shows training curves for TDPO (our method) as compared to TRPO, PPO, DDPG, and TD3. These results were averaged over 25 experiments in which the desired oscillation frequency was 1.7 Hz (different from the pendulum's natural frequency of 0.5 Hz), the desired oscillation amplitude was 0.28 rad, and the desired offset was 0.52 rad. Figure 1 also shows trajectories obtained by the best agents from each method. TDPO (our method) was able to learn high-reward behavior and to achieve the desired frequency, amplitude, and offset. TRPO was able to learn the correct offset but did not produce any oscillatory behavior. TD3 also learned the correct offset, but not the desired oscillation. PPO and DDPG failed to learn any desired behavior. We hypothesize that our method was able to learn good behaviors here because using a deterministic policy for exploration avoids excitation of spurious frequencies and the DeVine estimator is accurate even for the non-local reward.

## 4.2    An Environment with Long Horizon and Resonant Frequencies[2]

The second environment that we consider is a single leg from a quadruped robot [44]. This leg has two joints, a "hip" and a "knee," about which it is possible to exert torques. The hip is attached to a

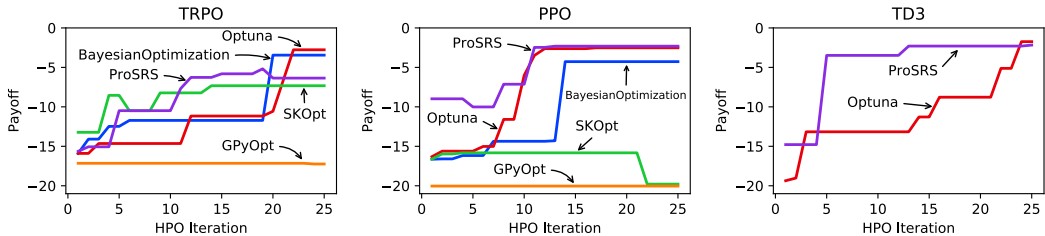

Figure 3: The best payoff vs. the Hyper-Parameter Optimization (HPO) iteration on a short-horizon variant of the legged robotic environment. The HPOs are performed for each of the TRPO, PPO, and TD3 methods in a separate panel. DDPG is a special case of TD3 with HPO. Since TD3 was considerably more expensive, we only show Optuna and ProSRS for it.

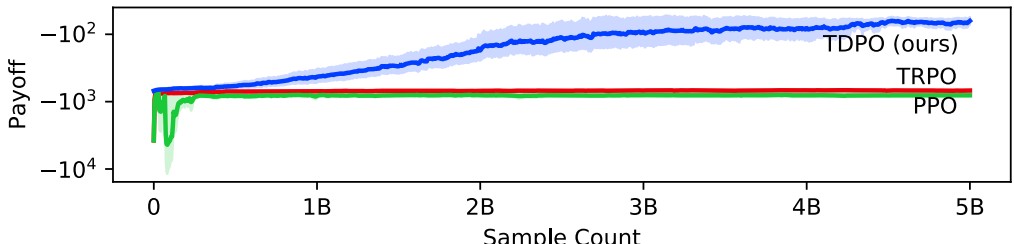

Figure 4: Post Hyper-Parameter Optimization (HPO) training curves with the best settings found for TRPO and PPO compared to the advanced variant of our method (TDPO with adaptive exploration scales and line search). TD3 had a significantly poor performance in the initial parameter sweeps. Due to resource limitations and poor initial performance, we excluded TD3 from this experiment.

slider that confines motion to a vertical line above flat ground. We assume the leg is dropped from some height above the ground and the task is to recover from this drop and to stand upright at rest after impact. States given to the agent are the angle and velocity of each joint (slider position and velocity are hidden), and actions are the joint torques. The reward function penalizes difference from an upright posture, slipping or chattering at the contact between the foot and the ground, non-zero joint velocities, and steady-state joint torque deviations. We use the open-source MuJoCo software for simulation [55], with high-fidelity models of ground compliance, motor nonlinearity, and joint friction. The control loop rate is $4000$ Hz and the rollout length is $2$ s, resulting in a horizon of $8000$ steps. Implementation details are left to the Supplementary Material.

Figure 2 shows training curves for TDPO (our method) as compared to TRPO, PPO, DDPG and TD3. These results were averaged over 75 experiments. A discount factor of $\gamma = 0.99975$ was chosen for all methods, where $(1 - \gamma)^{-1}$ is half the trajectory length. Similarly, the GAE factors for PPO and TRPO were scaled up to $0.99875$ and $0.9995$, respectively, in proportion to the trajectory length. Figure 2 also shows trajectories obtained by the best agents from each method. TDPO (our method) was able to learn high-reward behavior. TRPO, PPO, DDPG, and TD3 were not.

We hypothesize that the reason for this difference in performance is that TDPO better handles the combination of two challenges presented by the leg environment—an unusually long time horizon (8000 steps) and the existence of a resonant frequency that results from compliance between the foot and the ground (note the oscillations at a frequency of about $100$ Hz that appear in the trajectories after impact). Both high-speed control loops and resonance due to ground compliance are common features of real-world legged robots to which TDPO seems to be more resilient.

---

[2]Resonant frequencies are a concept from control theory. In the frequency domain, signals of certain frequencies are excited more than others when applied to a system. This is captured by the frequency-domain transfer function of the system, which may have a peak of magnitude greater than one. The resonant frequency is the frequency at which the frequency-domain transfer function has the highest amplitude. Common examples of systems with a resonant frequency include the undamped pendulum, which oscillates at its natural frequency, and RLC circuits which have characteristic frequencies at which they are most excitable. See Chapter 8 of Kuo and Golnaraghi [34] for more information.

### 4.3 Practical Training and Hardware Implementation

For the most realistic setting, we take the environment from the previous section and make it highly stochastic by (a) injecting physical modeling noise into the transition dynamics $P$, and (b) making the initial state distribution $\mu$ as random as physically possible. We also systematically perform Hyper-Parameter Optimization (HPO) on all methods to allow the most fair comparison.

The choice of the HPO method can have a significant impact on the RL agent's performance. We consider a list of five off-the-shelf HPO implementations and run them in their default settings: Optuna [2], BayesianOptimization [42], Scikit-Optimize [19], GPyOpt [17], and ProSRS [53]. These implementations include a range of HPO methods, including Gaussian processes and tree Parzen estimators. For better performance, HPO methods need a reasonable set of initial hyper-parameter guesses. For this, we perform a one-variable-at-a-time parameter sweep along every hyper-parameter near the RL method's default hyper-parameters. These parameter sweep results are then input to each HPO method for full optimization. Using all HPO algorithms for all RL methods in the long-horizon environment (where each full training run takes 5 billion samples) is computationally infeasible. To pick the best HPO method, we benchmark a short-horizon environment with only 200 time-steps in a trajectory. The result is shown in Figure 3 (see the Supplementary Material for full details

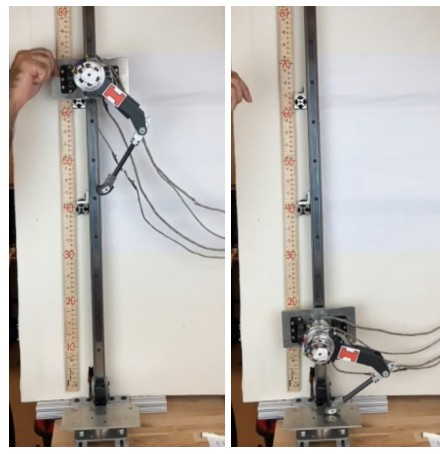

Figure 5: The simulation-to-real transfer of the best TDPO agent to perform a successful drop test at 4 kHz control rate.

on the HPO methods). Overall, we found that Optuna and ProSRS are the best HPO methods on the test problem. Since Optuna is widely-tested and arguably the most popular HPO library, we pick it as the main HPO method for our long-horizon environment.

We repeat the same HPO procedure on the long-horizon environment using Optuna, and pick the best hyper-parameters found in the course of HPO for a final training. Figure 4 shows this final training. TDPO shows superior performance in this highly stochastic environment, and such benefits cannot be obtained by merely performing HPO on other methods. To showcase the practicality of our method, we picked the best TDPO trained agent, and implemented it on the physical hardware. The transferred agent was able to successfully perform drop-and-catch tests on the robot system at 4 kHz, with both global control and suppression of high-frequency transients. Figure 5 shows a glimpse of this test, and a short video is also included in the code repository.

## 5 Discussion

We proposed a deterministic policy gradient method (TDPO: Truly Deterministic Policy Optimization) based on the use of a deterministic Vine (DeVine) gradient estimator and the Wasserstein metric. We proved monotonic payoff guarantees for our method, and defined a novel surrogate for policy optimization. We believe that using deterministic policies for exploration and avoiding the need for consistent noise injection results in lower gradient estimation variances, enabling our method to solve tasks with longer horizons and non-local rewards. We introduced several realistic robotic control tasks that have such features and we showed that TDPO performs well on them, in contrast to existing policy gradient methods.

There are a number of limitations of this paper. First, we assumed continuous environments and required a state reset capability of the environment, which is commonly available in simulators but would prevent learning on hardware. Second, TDPO relies on local gradients and thus may not be able to learn effectively in environments with payoffs that are sparse in either action or state. Such environments typically need substantial exploration which TDPO may not be able to achieve. Third, the analysis in this paper does not specifically pinpoint the reasons why existing methods fail on the long-horizon non-local-reward robotic control environments. For instance, such failures might be more specifically attributed to either (1) the frequency-based nature of the reward, (2) the non-locality of the reward signal, or (3) the long horizon. Understanding this is an important prerequisite for future work.

# 6 Acknowledgement

We sincerely thank Chenzhuang Li for his valuable help, feedback, and expertise with the physical robotic system. We also thank Hae-Won Park for sending the robotic platform to facilitate the experiments in this article, and the valuable advice for system identification. This research is part of the Blue Waters sustained-petascale computing project, which is supported by the National Science Foundation (awards OCI-0725070 and ACI-1238993) the State of Illinois, and as of December, 2019, the National Geospatial-Intelligence Agency. Blue Waters is a joint effort of the University of Illinois at Urbana-Champaign and its National Center for Supercomputing Applications [8, 33]. This work also utilizes resources supported by the National Science Foundation's Major Research Instrumentation (MRI) program, grant Number 1725729.

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
