# A Appendix

## A.1 Tables of Notation

The same mathematical definitions and notations used in the paper were re-introduced and summarized in two tables; Table 1 describes the mathematical functions and operators used throughout the paper, and Table 2 describes the notations needed to define the Markov Decision Process (MDP). The tables consist of two columns; one showing or defining the notation, and the other includes the name in which the same notation was called in the paper.

| Name | Mathematical Definition or Description |
|---|---|
| Value function | $V^\pi(s) := \frac{1}{1-\gamma} \mathbb{E}_{\substack{s_t \sim \rho_\mu^\pi \\ a_t \sim \pi(s_t)}} [R(s_t, a_t)]$ $= \mathbb{E}[\sum_{t=1}^\infty \gamma^{t-1} R(s_t, a_t)\|s_1 = s, a_t \sim \pi(s_t), s_{t+1} \sim P(s_t, a_t)].$ |
| Q-Value function | $Q^\pi(s, a) := R(s, a) + \gamma \cdot \mathbb{E}_{s' \sim P(s,a)}[V^\pi(s')]$ |
| Advantage function | $A^\pi(s, a) := Q^\pi(s, a) - V^\pi(s).$ |
| Advantage function | $A^\pi(s, \pi') := \mathbb{E}_{a \sim \pi'(s)}[A^\pi(s, a)].$ |
| Arbitrary functions | $f$ and $g$ are arbitrary functions used next. |
| Arbitrary distributions | $\nu$ and $\zeta$ are arbitrary probability distributions used next. |
| Hilbert inner product | $\langle f, g \rangle_x := \int f(x)g(x)\mathrm{d}x$ |
| Kulback-Liebler (KL) divergence | $D_{\mathrm{KL}}(\zeta\|\nu) := \langle \zeta(x), \log(\frac{\zeta(x)}{\nu(x)}) \rangle_x = \int \zeta(x) \log(\frac{\zeta(x)}{\nu(x)})\mathrm{d}x$ |
| Total Variation (TV) distance | $\mathrm{TV}(\zeta, \nu) := \frac{1}{2}\langle \|\zeta(x) - \nu(x)\|, 1 \rangle_x = \frac{1}{2}\int \|\zeta(x) - \nu(x)\|\mathrm{d}x.$ |
| Coupling set | $\Gamma(\zeta, \nu)$ is the set of couplings for $\zeta$ and $\nu$. |
| Wasserstein distance | $W(\zeta, \nu) = \inf_{\gamma \in \Gamma(\zeta,\nu)} \langle \|x - y\|, \gamma(x, y) \rangle_{x,y}.$ |
| Policy Wasserstein distance | $W(\pi_1, \pi_2) := \sup_{s \in \mathcal{S}} W(\pi_1(\cdot\|s), \pi_2(\cdot\|s)).$ |
| Lipschitz Constant | $\mathrm{Lip}(f(x, y); x) := \sup_x \|\nabla_x f(x, y)\|_2.$ |
| Rubinstein-Kantrovich (RK) duality | $\|\langle \zeta(x) - \nu(x), f(x) \rangle_x\| \leq W(\zeta, \nu) \cdot \mathrm{Lip}(f; x).$ |

Table 1: The mathematical notations used throughout the paper.

| Mathematical Notation | Name and Description |
|---|---|
| $\mathcal{S}$ | This is the state space of the MDP. |
| $\mathcal{A}$ | This is the action space of the MDP. |
| $\gamma$ | This is the discount factor of the MDP. |
| $R : \mathcal{S} \times \mathcal{A} \to \mathbb{R}$ | This is the reward function of the MDP. |
| $\mu$ | This is the initial state distribution of the MDP over the state space. |
| $\Delta$ | $\Delta(\mathcal{F})$ is the set of all probability distributions over the arbitrary set $\mathcal{F}$ (otherwise known as the Credal set of $\mathcal{F}$). |
| $\pi$ | In general, $\pi$ denotes the policy of the MDP. However, the output argument type could vary in the text. See the next lines. |
| $\pi : \mathcal{S} \to \Delta(\mathcal{A})$ | Given a state $s \in \mathcal{S}$, $\pi(s)$ and $\pi(\cdot\|s)$ denote the action distribution suggested by the policy $\pi$. 

 In other words, $a \sim \pi(s)$ and $a \sim \pi(\cdot\|s)$. |
| $\pi_{\text{det}} : \mathcal{S} \to \mathcal{A}$ | For a deterministic policy $\pi_{\text{det}}$, the unique action $a$ suggested by the policy given the state $s$ can be denoted by $\pi(s)$ specially. 

 In other words, $a = \pi_{\text{det}}(s)$. |
| $\Pi$ | $\Pi$ is the set of all policies (i.e., $\forall \pi : \pi \in \Pi$). |
| $P$ | In general, $P$ denotes the transition dynamics model of the MDP. However, the input argument types could vary throughout the text. See the next lines for more clarification. |
| $P : \mathcal{S} \times \mathcal{A} \to \Delta(\mathcal{S})$ | Given a particular state $s$ and action $a$, $P(s, a)$ will be the next state distribution of the transition dynamics (i.e. $s' \sim P(s, a)$ where $s'$ denotes the next state after applying $s, a$ to the transition $P$). |
| $P : \Delta(\mathcal{S}) \times \mathcal{A} \to \Delta(\mathcal{S})$ | This is a generalization of the transition dynamics to accept state distributions as input. In other words, $P(\nu_s, a) := \mathbb{E}_{s \sim \nu_s}[P(s, a)]$. |
| $P : \mathcal{S} \times \Delta(\mathcal{A}) \to \Delta(\mathcal{S})$ | This is a generalization of the transition dynamics to accept action distributions as input. In other words, $P(s, \nu_a) := \mathbb{E}_{a \sim \nu_a}[P(s, a)]$. |
| $\mathbb{P} : \Delta(\mathcal{S}) \times \Pi \to \Delta(\mathcal{S})$ | This is a generalization of the transition dynamics to accept a state distribution and a policy as input. Given an arbitrary state distribution $\nu_s$ and a policy $\pi$, and $\mathbb{P}(\nu_s, \pi)$ will be the next state distribution given that the state is sampled from $\nu_s$ and the action is sampled from the $\pi(s)$ distribution. 

 In other words, we have $\mathbb{P}(\nu_s, \pi) := \mathbb{E}_{\substack{s \sim \nu_s \\ a \sim \pi(s)}}[P(s, a)]$. |
| $\mathbb{P}^t : \Delta(\mathcal{S}) \times \Pi \to \Delta(\mathcal{S})$ | This is the $t$-step transition dynamics generalization. Given an arbitrary state distribution $\nu_s$ and a policy $\pi$ and non-negative integer $t$, one can define $\mathbb{P}^t$ recursively as $\mathbb{P}^0(\nu_s, \pi) := \nu_s$ and $\mathbb{P}^t(\nu_s, \pi) := \mathbb{P}(\mathbb{P}^{t-1}(\nu_s, \pi), \pi)$. |
| $\rho_\mu^\pi$ | The discounted visitation frequency $\rho_\mu^\pi$ is a distribution over $\mathcal{S}$, and can be defined as $\rho_\mu^\pi := (1 - \gamma) \sum_{t=0}^\infty \gamma^t \mathbb{P}^t(\mu, \pi)$. |

Table 2: The MDP notations used throughout the paper.

## A.2 Brief Introduction to Policy Gradient Methods

Conservative Policy Iteration (CPI) [25] was one of the early dimensionally consistent methods with a surrogate of the form $\mathcal{L}_{\pi_1}(\pi_2) = \eta_{\pi_1} + \frac{1}{1-\gamma} \cdot \mathbb{E}_{s\sim\rho_\mu^{\pi_1}}[A^{\pi_1}(s,\pi_2)] - \frac{C}{2}\mathrm{TV}^2(\pi_1,\pi_2)$. The $C$ coefficient guarantees non-decreasing payoffs. However, CPI is limited to linear function approximation classes due to the update rule $\pi_{\mathrm{new}} \leftarrow (1-\alpha)\pi_{\mathrm{old}} + \alpha\pi'$. This lead to the design of the Trust Region Policy Optimization (TRPO) [50] algorithm.

TRPO exchanged the bounded squared TV distance with the KL divergence by lower bounding it using the Pinsker inequality. This made TRPO closer to the Natural Policy Gradient algorithm[26], and for Gaussian policies, the modified terms had similar Taylor expansions within small trust regions. Confined trust regions are a stable way of making large updates and avoiding pessimistic coefficients. For gradient estimates, TRPO used Importance Sampling (IS) with a baseline shift:

$$\nabla_{\theta_2}\mathbb{E}_{s\sim\rho_\mu^{\pi_1}}[A^{\pi_1}(s,\pi_2)]\Big|_{\theta_2=\theta_1} = \nabla_{\theta_2}\mathbb{E}_{\substack{s\sim\rho_\mu^{\pi_1}\\a\sim\pi_1(\cdot|s)}}\left[Q^{\pi_1}(s,a)\frac{\pi_2(a|s)}{\pi_1(a|s)}\right]\Big|_{\theta_2=\theta_1}. \tag{12}$$

While empirical $\mathbb{E}[A^{\pi_1}(s,\pi_2)]$ and $\mathbb{E}[Q^{\pi_1}(s,\pi_2)]$ estimates yield identical variances in principle, the importance sampling estimator in (12) imposes larger variances. Later, Proximal Policy Optimization (PPO) [51] proposed utilizing the Generalized Advantage Estimation (GAE) method for variance reduction and incorporated first-order smoothing like ADAM [32]. GAE employed Temporal-Difference (TD) learning [7] for variance reduction. Although TD-learning was not theoretically guaranteed to converge and could add bias, it improved the estimation accuracy.

As an alternative to IS, deterministic policy gradient estimators were also utilized in an actor-critic fashion. Deep Deterministic Policy Gradient (DDPG) [35] generalized deterministic gradients by employing Approximate Dynamic Programming (ADP) [39] for variance reduction. Twin Delayed Deterministic Policy Gradient (TD3) [15] used twin critic networks and reduced the actor update frequency to address the value over-estimation phenomenon observed in DDPG. Although both methods used deterministic policies, they still performed stochastic search by adding stochastic noise to the deterministic policies to force exploration.

Other lines of stochastic policy optimization were later proposed. Wu et al. [59] used a Kronecker-factored approximation for curvatures. Haarnoja et al. [18] proposed a maximum entropy actor-critic method for stochastic policy optimization.

## A.3 Reinforcement Learning Challenges

We will briefly describe a few challenges in modern reinforcement learning: (a) the problem of non-local rewards, (b) scalability to longer horizons, and (c) observation or action delay.

### A.3.1 Non-local Rewards

An underlying assumption in the MDP framework is that the desired payoff can be decomposed into a (discounted) sum of time-step rewards. This leaves out practical payoff functions that cannot be expressed in this form. Non-local rewards are reward functions of the entire trajectory whose payoffs cannot be decomposed into the sum of terms such as $\eta = \sum_t f_t(s_t, a_t)$, where functions $f_t$ only depend on nearby states and actions. An example non-local reward is one that depends on the Fourier transform of the complete trajectory signal. Other examples include trajectory statistics (e.g., the median or maximum observation in a trajectory). In both examples, the reward cannot be determined without collecting the entire trajectory. While approximating non-local rewards with local ones is possible, such approximations may be difficult to engineer and may induce undesired behavior in the resulting policy. Although policy gradient methods are designed under the MDP framework and theoretically under-equipped for such challenges, being resilient to them is a desired property.

### A.3.2 Scalability to Longer Horizons

In its simplified and un-discounted form, reinforcement learning aims at optimizing the $\eta = \sum_{t=1}^{T} r_t$ payoff by determining the proper actions. It is insightful to contemplate the difficulty of this goal relative to the time-horizon $T$. With $T = 1$ the optimal policy is to take the greedy action at each time-step. However, with larger $T$ finding the optimal policy becomes more challenging.

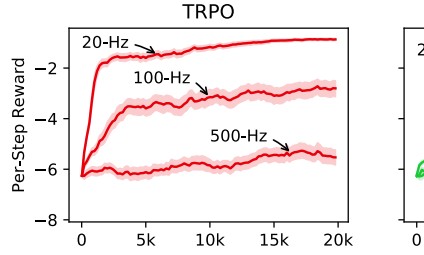 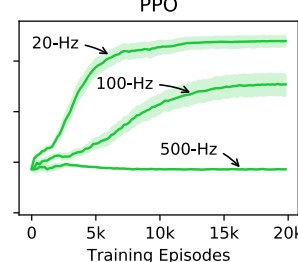 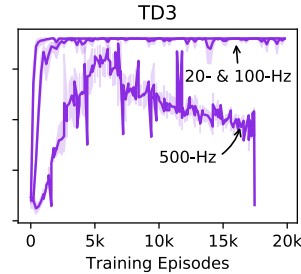

Figure 6: The effect of soft horizon scaling on the typical pendulum continuous control task. Three environments are defined at different control frequencies. All environments try to achieve the same goal of making the same pendulum stand up-right within 10 seconds. The original environment runs at 20 Hz control frequency. We also show two similar environments running at 100 Hz and 500 Hz control frequency. To make the tasks comparable, the horizontal axis shows the training episodes and the vertical axis shows the normalized reward per time-step. The environment and training hyper-parameters are given in Table 3. Evidently, all methods suffer from the curse of horizon.

| General Hyper-Parameters | Control Frequency | | |
|---|---|---|---|
| | 20 hz | 100 hz | 500 hz |
| Control Time-step | 50 ms | 10 ms | 2 ms |
| Trajectory Duration | 10 s | 10 s | 10 s |
| Parallel Workers | 4 | 4 | 4 |
| Training Episodes | 20K | 20K | 20K |
| Episode Time-steps | 200 | 1000 | 5000 |

| TD3 Hyper-Parameters | Control Frequency | | |
|---|---|---|---|
| | 20 hz | 100 hz | 500 hz |
| MDP Discount | 0.99 | 0.998 | 0.9996 |
| Replay Buffer Size | 50K | 250K | 1.25M |
| Initial Random Steps | 100 | 500 | 2500 |
| Training Interval | 100 | 500 | 2500 |
| Opt. Batch Size | 128 | 640 | 3200 |

| TRPO Hyper-Parameters | Control Frequency | | |
|---|---|---|---|
| | 20 hz | 100 hz | 500 hz |
| Sampling Batch Size | 1024 | 5120 | 25600 |
| MDP Discount | 0.99 | 0.998 | 0.9996 |
| GAE Discount | 0.98 | 0.996 | 0.9992 |
| Value Batch Size | 128 | 640 | 3200 |

| PPO Hyper-Parameter | Control Frequency | | |
|---|---|---|---|
| | 20 hz | 100 hz | 500 hz |
| Sampling Batch Size | 256 | 1280 | 6400 |
| MDP Discount | 0.99 | 0.998 | 0.9996 |
| GAE Discount | 0.95 | 0.99 | 0.998 |
| Opt. Batch Size | 64 | 320 | 1600 |

Table 3: The settings and hyper-parameters used to produce Figure 6. The top-left table shows the common settings used to define the environment and the run the training. The scaled hyper-parameters for each of the TD3, TRPO, and PPO methods are given in the top-right, bottom-left, and bottom-right corner, respectively. Other hyper-parameters were set to their default value in all methods.

In infinite-horizon discounted MDPs, $1/(1-\gamma)$ can be considered the counterpart of $T$. In particular, we have $\sum_{i=1}^{T} \gamma^i / \sum_{i=1}^{\infty} \gamma^i = 1 - \gamma^T \simeq 1 - e^{-T(1-\gamma)}$; although the MDP framework defines infinite time-steps, the cumulative weight of time-steps after $T$ in the payoff decays exponentially with a $1/(1-\gamma)$ time constant. For instance, with $\gamma = 0.99$ the first 200 steps constitute 87% of the infinite-length trajectory payoff, whereas with $\gamma = 0.999$ the first 2000 steps constitute the same portion of the payoff. This is why $1/(1-\gamma)$ appears in most theoretical sample-complexity analyses and higher bounds in an exponential capacity [27, 31, 30]. Practically, longer horizons can appear in at least two forms: (a) preserving the task complexity but increasing the decision-making frequency, and (b) increasing the task complexity. We call these forms *soft* and *hard* horizon scalability, respectively.

**Soft Horizon Scalability:** In physical systems, one can preserve the task complexity but increase the control frequency. This increases the policy's agility in adapting to changes in the observation. Each time-step can be divided into $k$ smaller time-steps, resulting in a $k$-fold increase of time-steps per trajectory. This is what we call *soft horizon scalability*. Intuitively, we expect soft horizon scaling to improve the optimal policy's performance; the smaller time-step policy can be faster in response to observation changes, and the policy will have the freedom to choose different actions in the $k$

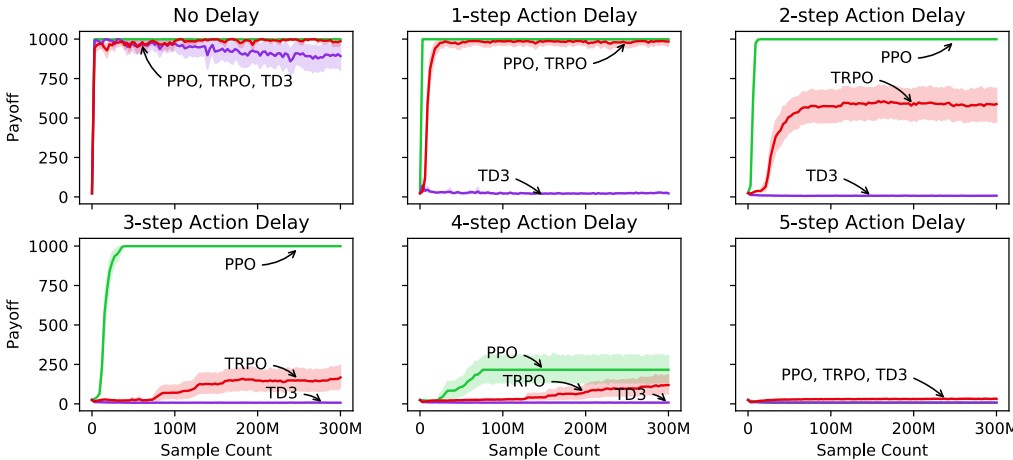

Figure 7: The effect of action delay on the inverted pendulum continuous control task where early-termination was disabled. The top left plot shows each method's performance on the original problem, and the rest simulate different amounts of action delay time-steps. All methods' lose performance with 5 time-steps of action delay.

smaller time-steps rather than being constrained to apply the same action during the entire $k$ smaller time-steps. However, this expectation may not be satisfied in practice.

To showcase this effect, we consider a typical pendulum benchmark problem. Existing methods can solve this task with their default settings in less than a million serial time-steps. By making the control time-steps 5 or 25 times smaller, one may hope to achieve higher per-step rewards. Of course, some hyper-parameters (e.g., the sampling batch-size) must be scaled proportionally to have comparable settings. Table 3 summarizes such scalings. Figure 6 shows the training curves for each method and control frequency. Evidently, all methods suffer from the curse of horizon. In particular, on-policy methods (TRPO and PPO) are most vulnerable to soft-horizon scaling. TD3, on the other hand, is closer to off-policy algorithms. This problem has usually been addressed with the *frame-skip* trick, where the same action is zero-held for multiple time-steps. The performance deterioration with soft horizon scaling can be attributed to the higher variance of estimated gradients in reinforcement learning methods with longer horizons.

**Hard Horizon Scalability**   When multiple tasks are stacked in the time horizon and are conditioned upon the completion of each other, *hard horizon scalability* is achieved. Consider a treasure hunt game where the next clue is conditioned upon solving the current task. Stacking more tasks makes winning the game exponentially more difficult; a single mistake in any step can result in failure. Reinforcement learning methods can suffer the same way with composite tasks. It is difficult to resolve hard horizon scalability without being resilient to soft horizon scalability in the first place. Overall, hard horizon scalability is a difficult challenge and beyond the scope of our work.

### A.3.3   Action or Observation Delay

Delay in sensing the observation or applying the desired actuation is a challenging artifact in physical systems. Such delays have been a favorite topic of research in traditional control theory [34]. Such delays are most influential in high-bandwidth control systems. Although the MDP framework does not address observation or action delays, being resilient to them is a desirable feature for policy gradient methods. To show the effect of delay on PG methods, we simulated a typical inverted pendulum task and delayed the proposed actions by the agent for different numbers of time-steps. The training curves are shown in Figure 7. With no action delay, all methods produce high-performance agents. However, at 5 time-steps of action delay, the resulting agents are almost indistinguishable from the initial policies performance-wise. In particular, TD3 is most vulnerable to delay, while PPO and TRPO could tolerate a few time-steps of delay. We speculate that this is due to TD3 being closely related to the TD(0) methods, whereas PPO and TRPO are closer to TD(1) when estimating the state values in their training processes. Overall, observation and action delays are unresolved topics in modern reinforcement learning.

## B  Theoretical Proofs and Derivations

Two theoretical results from the main paper were left to be discussed here. The bulk of our theoretical derivations (Sections B.1-B.6) belongs to the payoff improvement lower-bound of Theorem B.7, which was used in Algorithm 1 of the main paper to regulate the policy updates. Figure 8 shows a flow-chart of the theoretical steps necessary to prove Theorem B.7, and Section B.8 discusses the assumptions used in the theoretical derivations. On a separate note, Section B.7 is dedicated to proving Theorem 3.1, which shows that the DeVine advantage estimator (Algorithm 2 of the main paper) can provide exact policy gradient estimates under certain conditions.

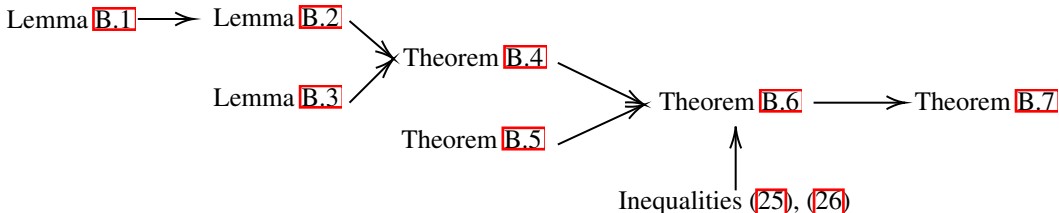

Figure 8: The theoretical derivations flow-chart to prove Theorem B.7.

### B.1  Bounding $W(\mathbb{P}^t(\mu, \pi_1), \mathbb{P}^t(\mu, \pi_2))$

To review, the dynamical smoothness assumptions were

$$W(\mathbb{P}(\mu, \pi_1), \mathbb{P}(\mu, \pi_2)) \leq L_\pi \cdot W(\pi_1, \pi_2),$$
$$W(\mathbb{P}(\mu_1, \pi), \mathbb{P}(\mu_2, \pi)) \leq L_\mu \cdot W(\mu_1, \mu_2).$$

The following lemma states that these two assumptions are equivalent to a more concise assumption. This will be used to bound the $t$-step visitation distance and prove Lemma B.2.

**Lemma B.1.** *Assumptions (4) and (5) are equivalent to having*

$$W(\mathbb{P}(\mu_1, \pi_1), \mathbb{P}(\mu_2, \pi_2)) \leq L_\mu \cdot W(\mu_1, \mu_2) + L_\pi \cdot W(\pi_1, \pi_2). \tag{13}$$

*Proof.* To prove the (4), (5) $\Rightarrow$ (13) direction, the triangle inequality for the Wasserstein distance gives

$$W(\mathbb{P}(\mu_1, \pi_1), \mathbb{P}(\mu_2, \pi_2)) \leq W(\mathbb{P}(\mu_1, \pi_1), \mathbb{P}(\mu_2, \pi_1)) + W(\mathbb{P}(\mu_2, \pi_1), \mathbb{P}(\mu_2, \pi_2)) \tag{14}$$

and using (4), (5), and (14) then implies

$$W(\mathbb{P}(\mu_1, \pi_1), \mathbb{P}(\mu_2, \pi_2)) \leq L_\mu \cdot W(\mu_1, \mu_2) + L_\pi \cdot W(\pi_1, \pi_2). \tag{15}$$

The other direction is trivial. $\qquad\square$

**Lemma B.2.** *Under Assumptions (4) and (5) we have the bound*

$$W(\mathbb{P}^t(\mu, \pi_1), \mathbb{P}^t(\mu, \pi_2)) \leq L_\pi \cdot (1 + L_\mu + \cdots + L_\mu^{t-1}) \cdot W(\pi_1, \pi_2), \tag{16}$$

*where $\mathbb{P}^t(\mu, \pi)$ denotes the state distribution after running the MDP for $t$ time-steps with the initial state distribution $\mu$ and policy $\pi$.*

*Proof.* For $t = 1$, the lemma is equivalent to Assumption (4). This paves the way for the lemma to be proved using induction. The hypothesis is

$$W(\mathbb{P}^{t-1}(\mu, \pi_1), \mathbb{P}^{t-1}(\mu, \pi_2)) \leq L_\pi \cdot (1 + L_\mu + \cdots + L_\mu^{t-2}) \cdot W(\pi_1, \pi_2), \tag{17}$$

and for the induction step we write

$$W(\mathbb{P}^t(\mu, \pi_1), \mathbb{P}^t(\mu, \pi_2)) = W(\mathbb{P}(\mathbb{P}^{t-1}(\mu, \pi_1), \pi_1), \mathbb{P}(\mathbb{P}^{t-1}(\mu, \pi_2), \pi_2)). \tag{18}$$

Using Assumption (13), which according to Lemma B.1 is equivalent to Assumptions (4) and (5), we can combine (17) and (18) into

$$W(\mathbb{P}^t(\mu, \pi_1), \mathbb{P}^t(\mu, \pi_2)) \leq L_\pi \cdot W(\pi_1, \pi_2) + L_\mu \cdot W(\mathbb{P}^{t-1}(\mu, \pi_1), \mathbb{P}^{t-1}(\mu, \pi_2)). \tag{19}$$

Thus, by applying the induction Hypothesis (17), we have

$$W(\mathbb{P}^t(\mu, \pi_1), \mathbb{P}^t(\mu, \pi_2)) \leq L_\pi \cdot W(\pi_1, \pi_2) + L_\mu \cdot L_\pi \cdot (1 + L_\mu + \cdots + L_\mu^{t-2}) \cdot W(\pi_1, \pi_2), \tag{20}$$

which can be simplified into the lemma statement (i.e., Inequality (16)). $\qquad\square$

## B.2 Bounding $W(\rho_\mu^{\pi_1}, \rho_\mu^{\pi_2})$

Lemma B.2 suggests making the $\gamma L_\mu < 1$ assumption and paves the way for Theorem B.4. The $\gamma L_\mu < 1$ assumption is overly restrictive and unnecessary but makes the rest of the proof easier to follow. This assumption can be relaxed by a general transition dynamics stability assumption which is discussed in more detail later in section B.8.3, and an equivalent $\gamma \bar{L}_\mu < 1$ assumption is introduced to replace $\gamma L_\mu < 1$.

First, we need to introduce Lemma B.3 first, which will be used in the proof of Theorem B.4.

**Lemma B.3.** *The Wasserstein distance between linear combinations of distributions can be bounded as* $W(\beta \cdot \mu_1 + (1 - \beta) \cdot \nu_1, \beta \cdot \mu_2 + (1 - \beta) \cdot \nu_2) \leq \beta \cdot W(\mu_1, \mu_2) + (1 - \beta) \cdot W(\nu_1, \nu_2)$.

*Proof.* Plugging $\gamma = \beta \cdot \gamma_{(\mu_1, \mu_2)} + (1 - \beta) \cdot \gamma_{(\nu_1, \nu_2)}$ in the Wasserstein definition yields the result. $\square$

**Theorem B.4.** *Assuming (4), (5), and* $\gamma L_\mu < 1$, *we have the inequality*

$$W(\rho_\mu^{\pi_1}, \rho_\mu^{\pi_2}) \leq \frac{\gamma L_\pi}{1 - \gamma L_\mu} \cdot W(\pi_1, \pi_2). \tag{21}$$

*Proof.* Using Lemma B.3 and the definition of $\rho_\mu^\pi$, we can write

$$W(\rho_\mu^{\pi_1}, \rho_\mu^{\pi_2}) \leq (1 - \gamma) \sum_{t=0}^{\infty} \gamma^t \cdot W(\mathbb{P}^t(\mu, \pi_1), \mathbb{P}^t(\mu, \pi_2)). \tag{22}$$

Using Lemma B.2, we can take another step to relax the inequality (22) and write

$$W(\rho_\mu^{\pi_1}, \rho_\mu^{\pi_2}) \leq \frac{L_\pi(1 - \gamma)W(\pi_1, \pi_2)}{(L_\mu - 1)} \sum_{t=0}^{\infty} ((\gamma L_\mu)^t - \gamma^t). \tag{23}$$

Due to the $\gamma L_\mu < 1$ assumption, the right-hand summation in (23) is convergent. This leads us to

$$W(\rho_\mu^{\pi_1}, \rho_\mu^{\pi_2}) \leq \frac{L_\pi(1 - \gamma)W(\pi_1, \pi_2)}{(L_\mu - 1)} \left( \frac{1}{1 - \gamma L_\mu} - \frac{1}{1 - \gamma} \right). \tag{24}$$

Inequality (24) can then be simplified to give the result. $\square$

## B.3 Steps to Bound the Second-order Term

The RK duality yields the following bound:

$$|\langle \rho_\mu^{\pi_2} - \rho_\mu^{\pi_1}, A^{\pi_1}(\cdot, \pi_2) \rangle_s| \leq W(\rho_\mu^{\pi_1}, \rho_\mu^{\pi_2}) \cdot \sup_s \|\nabla_s A^{\pi_1}(s, \pi_2)\|_2. \tag{25}$$

To facilitate the further application of the RK duality, any advantage can be rewritten as the following inner product: $A^{\pi_1}(s, \pi_2) = \langle \pi_2(a|s) - \pi_1(a|s), Q^{\pi_1}(s, a) \rangle_a$. Taking derivatives of both sides with respect to the state variable and applying the triangle inequality produces the bound

$$\sup_s \|\nabla_s A^{\pi_1}(s, \pi_2)\|_2 \leq \sup_s \|\langle \nabla_s(\pi_2(a|s) - \pi_1(a|s)), Q^{\pi_1}(s, a) \rangle_a\|_2$$
$$+ \sup_s \|\langle \pi_2(a|s) - \pi_1(a|s), \nabla_s Q^{\pi_1}(s, a) \rangle_a\|_2. \tag{26}$$

The second term of the RHS in (26) is compatible with the RK duality. However, the form of the first term does not warrant an easy application of RK. For this, we introduce Theorem B.5.

**Theorem B.5.** *Assuming the existence of* $\mathrm{Lip}(Q^{\pi_1}(s, a); a)$, *we have the bound*

$$\left\| \langle \nabla_s(\pi_2(a|s) - \pi_1(a|s)), Q^{\pi_1}(s, a) \rangle_a \right\|_2 \tag{27}$$

$$\leq 2 \cdot \mathrm{Lip}(Q^{\pi_1}(s, a); a) \cdot \left\| \nabla_{s'} W\left( \frac{\pi_2(a|s') + \pi_1(a|s)}{2}, \frac{\pi_2(a|s) + \pi_1(a|s')}{2} \right) \Big|_{s'=s} \right\|_2.$$

*Proof.* By definition, we have

$$\big\|\big\langle \nabla_s(\pi_2(a|s) - \pi_1(a|s)), Q^{\pi_1}(s,a)\big\rangle_a\big\|_2$$

$$= \sqrt{\sum_{j=1}^{\dim(\mathcal{S})} \left(\left\langle \frac{\partial}{\partial s^{(j)}}(\pi_2(a|s) - \pi_1(a|s)), Q^{\pi_1}(s,a)\right\rangle_a\right)^2}. \qquad (28)$$

For better insight, we will write the derivative using finite differences:

$$\left\langle \frac{\partial}{\partial s^{(j)}}(\pi_2(a|s) - \pi_1(a|s)), Q^{\pi_1}(s,a)\right\rangle_a$$

$$= \lim_{\delta s \to 0} \frac{1}{\delta s}\Big[\big\langle(\pi_2(a|s + \delta s \cdot \mathbf{e}_j) \quad -\pi_1(a|s + \delta s \cdot \mathbf{e}_j) \quad ), Q^{\pi_1}(s,a)\big\rangle_a$$

$$-\big\langle(\pi_2(a|s) \quad\quad -\pi_1(a|s) \quad\quad ), Q^{\pi_1}(s,a)\big\rangle_a\Big]. \qquad (29)$$

We can rearrange the finite difference terms to get

$$\left\langle \frac{\partial}{\partial s^{(j)}}(\pi_2(a|s) - \pi_1(a|s)), Q^{\pi_1}(s,a)\right\rangle_a$$

$$= \lim_{\delta s \to 0} \frac{1}{\delta s}\Big[\big\langle(\pi_2(a|s + \delta s \cdot \mathbf{e}_j) \quad +\pi_1(a|s) \quad\quad ), Q^{\pi_1}(s,a)\big\rangle_a$$

$$-\big\langle(\pi_2(a|s) \quad\quad +\pi_1(a|s + \delta s \cdot \mathbf{e}_j) \quad ), Q^{\pi_1}(s,a)\big\rangle_a\Big]. \qquad (30)$$

Equivalently, we can divide and multiply the inner products by a factor of 2, to make the inner product arguments resemble mixture distributions:

$$\left\langle \frac{\partial}{\partial s^{(j)}}(\pi_2(a|s) - \pi_1(a|s)), Q^{\pi_1}(s,a)\right\rangle_a$$

$$= \lim_{\delta s \to 0} \frac{2}{\delta s}\left[\left\langle\frac{\pi_2(a|s + \delta s \cdot \mathbf{e}_j) + \pi_1(a|s)}{2}, Q^{\pi_1}(s,a)\right\rangle_a\right.$$

$$\left. -\left\langle\frac{\pi_2(a|s) + \pi_1(a|s + \delta s \cdot \mathbf{e}_j)}{2}, Q^{\pi_1}(s,a)\right\rangle_a\right]. \qquad (31)$$

The RK duality can now be used to bound this difference as

$$\left|\left\langle \frac{\partial}{\partial s^{(j)}}(\pi_2(a|s) - \pi_1(a|s)), Q^{\pi_1}(s,a)\right\rangle_a\right| \qquad (32)$$

$$\leq \lim_{\delta s \to 0} \frac{2}{\delta s}\left[W\left(\frac{\pi_2(a|s + \delta s \cdot \mathbf{e}_j) + \pi_1(a|s)}{2}, \frac{\pi_2(a|s) + \pi_1(a|s + \delta s \cdot \mathbf{e}_j)}{2}\right) \cdot \mathrm{Lip}(Q^{\pi_1}(s,a); a)\right],$$

which can be simplified as

$$\left|\left\langle \frac{\partial}{\partial s^{(j)}}(\pi_2(a|s) - \pi_1(a|s)), Q^{\pi_1}(s,a)\right\rangle_a\right|$$

$$\leq 2 \cdot \mathrm{Lip}(Q^{\pi_1}(s,a); a) \cdot \frac{\partial}{\partial s'^{(j)}} W\left(\frac{\pi_2(a|s') + \pi_1(a|s)}{2}, \frac{\pi_2(a|s) + \pi_1(a|s')}{2}\right)\bigg|_{s'=s}. \qquad (33)$$

Combining Inequality (33) and Equation (28), we obtain the bound in the theorem. □

## B.4  The Preliminary Payoff Improvement Bound

Combining the results of Inequality (26) and Theorems B.5 and B.4 leads us to define the regularization terms and coefficients:

$$\mathrm{Lip}(\nabla_s Q^{\pi}(s,a); a) := \sqrt{\sum_{k=1}^{|\mathcal{S}|} \mathrm{Lip}(\frac{\partial}{\partial s_k} Q^{\pi_1}(s,a); a)^2},$$

$$C_1' := \sup_s \frac{2 \cdot \text{Lip}(Q^{\pi_1}(s,a);a) \cdot \gamma \cdot L_\pi}{(1-\gamma)(1-\gamma L_\mu)}, \qquad C_2' := \sup_s \frac{\text{Lip}(\nabla_s Q^\pi(s,a);a) \cdot \gamma \cdot L_\pi}{(1-\gamma)(1-\gamma L_\mu)},$$

$$\mathcal{L}_{WG}(\pi_1, \pi_2; s) := W(\pi_2(a|s), \pi_1(a|s))$$
$$\times \left\| \nabla_{s'} W\left( \frac{\pi_2(a|s') + \pi_1(a|s)}{2}, \frac{\pi_2(a|s) + \pi_1(a|s')}{2} \right)\Big|_{s'=s} \right\|_2. \tag{34}$$

This gives us the following novel lower bound for payoff improvement.

**Theorem B.6.** *Defining $C_1'$, $C_2'$, and $\mathcal{L}_{WG}(\pi_1, \pi_2; s)$ as in (34), we have $\eta_{\pi_2} \geq \mathcal{L}_{\pi_1}^{\text{sup}'}(\pi_2)$, where*

$$\mathcal{L}_{\pi_1}^{\text{sup}'}(\pi_2) := \eta_{\pi_1} + \frac{1}{1-\gamma} \mathbb{E}_{s \sim \rho_\mu^{\pi_1}} [A^{\pi_1}(s, \pi_2)] - C_1' \cdot \sup_s \left[ \mathcal{L}_{WG}(\pi_1, \pi_2; s) \right]$$
$$- C_2' \cdot \sup_s \left[ W(\pi_2(a|s), \pi_1(a|s))^2 \right]. \tag{35}$$

*Proof.* By first inserting Theorem B.5 into Inequality (26), and then applying it to Inequality (25) along with Theorem B.4, one can obtain the result according to the advantage decomposition lemma given in Equation (6) of the main paper. □

## B.5 Quadratic Modeling of Policy Sensitivity Regularization

First, we will build insight into the nature of the

$$\mathcal{L}_{WG}(\pi_1, \pi_2; s) = W(\pi_2(a|s), \pi_1(a|s)) \times$$
$$\left\| \nabla_{s'} W\left( \frac{\pi_2(a|s') + \pi_1(a|s)}{2}, \frac{\pi_2(a|s) + \pi_1(a|s')}{2} \right)\Big|_{s'=s} \right\|_2 \tag{36}$$

term. It is fairly obvious that

$$W(\pi_2(a|s), \pi_1(a|s))\big|_{\pi_2=\pi_1} = 0. \tag{37}$$

If $\pi_2 = \pi_1$, then the two distributions $\frac{\pi_2(a|s') + \pi_1(a|s)}{2}$ and $\frac{\pi_2(a|s) + \pi_1(a|s')}{2}$ will be the same no matter what $s'$ is. In other words,

$$\pi_1 = \pi_2 \Rightarrow \forall s' : W\left( \frac{\pi_2(a|s') + \pi_1(a|s)}{2}, \frac{\pi_2(a|s) + \pi_1(a|s')}{2} \right) = 0. \tag{38}$$

This means that

$$\left\| \nabla_{s'} W\left( \frac{\pi_2(a|s') + \pi_1(a|s)}{2}, \frac{\pi_2(a|s) + \pi_1(a|s')}{2} \right)\Big|_{s'=s} \right\|_2 \Bigg|_{\pi_2=\pi_1} = 0. \tag{39}$$

The Taylor expansion of the squared Wasserestein distance can be written as

$$W(\pi_2(a|s), \pi_1(a|s))^2 \big|_{\theta_2 = \theta_1 + \delta\theta} = \frac{1}{2} \delta\theta^T H_2 \delta\theta + \text{h.o.t.}. \tag{40}$$

Considering (38) and similar to the previous point, one can write the following Taylor expansion

$$\left\| \nabla_{s'} W\left( \frac{\pi_2(a|s') + \pi_1(a|s)}{2}, \frac{\pi_2(a|s) + \pi_1(a|s')}{2} \right)\Big|_{s'=s} \right\|_2^2 \Bigg|_{\theta_2 = \theta_1 + \delta\theta} = \delta\theta^T H_1 \delta\theta + \text{h.o.t.}. \tag{41}$$

According to above, $\mathcal{L}_{WG}$ is the geometric mean of two functions of quadratic order. Although this makes $\mathcal{L}_{WG}$ of quadratic order (i.e., $\lim_{\delta\theta \to 0} \frac{\mathcal{L}_{WG}(\alpha\delta\theta)}{\mathcal{L}_{WG}(\delta\theta)} = \alpha^2$ holds for any constant $\alpha$), this does not guarantee that $\mathcal{L}_{WG}$ is twice continuously differentiable w.r.t. the policy parameters, and may not have a defined Hessian matrix (e.g., $f(x_1, x_2) = |x_1 x_2|$ is of quadratic order, yet is not twice differentiable). To avoid this issue, we compromise on the local model. According to the arithmetic mean and geometric mean inequality, for all $x_1, x_2 \geq 0$ and any non-zero $\alpha$, we have

$$x_1 x_2 \leq \frac{\alpha^2 x_1^2 + \alpha^{-2} x_2^2}{2}. \tag{42}$$

Therefore, we can bound the $\mathcal{L}_{WG}$ term into two quadratic terms:

$$\mathcal{L}_{WG}(\pi_1, \pi_2; s) \leq \frac{1}{2}\left( \alpha^2 \cdot \left\| \nabla_{s'} W \left( \frac{\pi_2(a|s') + \pi_1(a|s)}{2}, \frac{\pi_2(a|s) + \pi_1(a|s')}{2} \right) \right|_{s'=s} \right\|_2^2 +$$

$$\alpha^{-2} \cdot W(\pi_2(a|s), \pi_1(a|s))^2 \bigg). \tag{43}$$

### B.6 The Final Payoff Improvement Guarantee

Inequality (43) paves the way for our final payoff lower bound theorem.

**Theorem B.7.** *By defining* $C_1 := \frac{C_1' \cdot \alpha^2}{2}$, $C_2 := (C_2' + \frac{C_1'}{2\alpha^2})$ *with any non-zero* $\alpha$, *and*

$$\mathcal{L}_{G^2}(\pi_1, \pi_2; s) := \left\| \nabla_{s'} W \left( \frac{\pi_2(a|s') + \pi_1(a|s)}{2}, \frac{\pi_2(a|s) + \pi_1(a|s')}{2} \right) \right|_{s'=s} \right\|_2^2, \tag{44}$$

*we have* $\eta_{\pi_2} \geq \mathcal{L}_{\pi_1}^{\sup}(\pi_2)$, *where*

$$\mathcal{L}_{\pi_1}^{\sup}(\pi_2) := \frac{1}{1-\gamma} \cdot \mathbb{E}_{s \sim \rho_\mu^{\pi_1}}[A^{\pi_1}(s, \pi_2)] - C_1 \cdot \sup_s \left[ \mathcal{L}_{G^2}(\pi_1, \pi_2; s) \right]$$

$$- C_2 \cdot \sup_s \left[ W(\pi_2(a|s), \pi_1(a|s))^2 \right]. \tag{45}$$

*Proof.* Applying Inequality (43) into Theorem B.6 gives the result. $\square$

$C_1$ and $C_2$ will be the corresponding regularization coefficients to the ones defined in Theorem B.6. Due to the arbitrary $\alpha$ used in the bounding process, no constrain governs the $C_1$ and $C_2$ coefficients. Therefore, $C_1$ and $C_2$ can be chosen without constraining each other.

### B.7 Proof of Theorem 3.1

Essentially, DeVine rolls out a trajectory and computes the values of each state. Since the transition dynamics and the policy are deterministic, these values are exact. Then, it picks a perturbation state $s_t$ according to the visitation frequencies. A state-reset to $s_t$ is made, a $\sigma$-perturbed action is applied for a single time-step, followed by $\pi_1$ policy. This exactly produces $Q^{\pi_1}(s_t, a_t + \sigma)$. Then, $A^{\pi_1}(s_t, a_t + \sigma)$ can be computed by subtracting the value baseline. Finally, $A^{\pi_1}(s_t, a_t) = 0$ and $A^{\pi_1}(s_t, a_t + \sigma)$ define a two-point linear $A^{\pi_1}(s_t, a)$ model with respect to the action. Parallelization can be used to have as many states of the first roll-out included in the estimator as desired. The parameter $\sigma$ acts as an exploration parameter and a finite difference to establish derivatives. While $\sigma \simeq 0$ can produce exact gradients, larger $\sigma$ can build stabler interpolations.

We restate Theorem 3.1 below for reference and now prove it.

**Theorem 4.1.** *Assume a finite horizon MDP with both deterministic transition dynamics $P$ and initial distribution $\mu$, with maximal horizon length of $H$. Define $K = H \cdot \dim(\mathcal{A})$ and $\nu := \nu_{det}$, where $\nu_{det}$ always returns the complete covering of $\{1, \cdots, \dim(\mathcal{A})\} \times \{1, \cdots, H\}$. Then we have*

$$\lim_{\sigma \to 0} \nabla_{\pi_2} \mathbb{A}^{\pi_1}(\pi_2) \big|_{\pi_2 = \pi_1} = \nabla_{\pi_2} \eta_{\pi_2} \big|_{\pi_2 = \pi_1}. \tag{46}$$

*Proof.* According to the advantage decomposition lemma, we have

$$\nabla_{\pi_2} \eta_{\pi_2} \big|_{\pi_2 = \pi_1} = \frac{1}{1-\gamma} \mathbb{E}_{s \sim \rho_\mu^{\pi_1}} [\nabla_{\pi_2} A^{\pi_1}(s, \pi_2)] \big|_{\pi_2 = \pi_1}. \tag{47}$$

Due to the fact that the transition dynamics, policies $\pi_1$ and $\pi_2$, and initial state distribution are all deterministic, we can simplify Equation (47) to

$$\nabla_{\pi_2} \eta_{\pi_2} \big|_{\pi_2 = \pi_1} = \sum_{t=0}^{H-1} \gamma^t \cdot \nabla_{\pi_2} A^{\pi_1}(s_t, \pi_2) \big|_{\pi_2 = \pi_1}, \tag{48}$$

where $s_t$ is the state after applying the policy $\pi_1$ for $t$ time-steps. We can use the chain rule to write

$$
\begin{aligned}
\nabla_{\pi_2} A^{\pi_1}(s_t, \pi_2)\big|_{\pi_2=\pi_1} &= \nabla_{\pi_2} A^{\pi_1}(s_t, a_t)\big|_{\substack{a_t=\pi_2(s_t)\\ \pi_2=\pi_1}} \\
&= \sum_{j=1}^{\dim(\mathcal{A})} \nabla_{\pi_2} a_t^{(j)} \cdot \frac{\partial}{\partial a_t^{(j)}} A^{\pi_1}(s_t, a_t)\big|_{\substack{a_t=\pi_2(s_t)\\ \pi_2=\pi_1}}.
\end{aligned}
\tag{49}
$$

To recap, Equations (48), (48), and (49) can be summarized as

$$
\nabla_{\pi_2} \eta_{\pi_2}\big|_{\pi_2=\pi_1} = \sum_{t=0}^{H-1} \gamma^t \sum_{j=1}^{\dim(\mathcal{A})} \nabla_{\pi_2} a_t^{(j)} \cdot \frac{\partial}{\partial a_t^{(j)}} A^{\pi_1}(s_t, a_t)\big|_{\substack{a_t=\pi_2(s_t)\\ \pi_2=\pi_1}}.
\tag{50}
$$

Under the assumption that the $(j, t)$ pairs are sampled to exactly cover $\mathcal{X}_K = \{1, \ldots, \dim(\mathcal{A})\} \times \{1, \ldots, H\}$, we can simplify the DeVine oracle to

$$
\begin{aligned}
\mathbb{A}^{\pi_1}(\pi_2) = \frac{1}{K} \sum_{t=0}^{H-1} \sum_{j=1}^{\dim(\mathcal{A})} &\left[ \frac{\dim(\mathcal{A}) \cdot H \cdot \gamma^t}{\nu_{\det}(\mathcal{X}_K)} \cdot \frac{(\pi_2(s_t) - a_t)^T (a_t' - a_t)}{(a_t' - a_t)^T (a_t' - a_t)} \right. \\
&\left. \cdot A^{\pi_1}(s_t, q(s_t; j, \sigma)) \right].
\end{aligned}
\tag{51}
$$

From the definition, we have $a_t' - a_t = \sigma \mathbf{e}_j$ and $(a_t' - a_t)^T (a_t' - a_t) = \sigma^2$. Since $\nu$ is uniform (i.e., $\nu_{\det}(\mathcal{X}_K) = 1$) and $K = H \cdot \dim(\mathcal{A})$, we can take the policy gradient of Equation (51) and simplify it into

$$
\nabla_{\pi_2} \mathbb{A}^{\pi_1}(\pi_2)\big|_{\pi_2=\pi_1} = \sum_{t=0}^{H-1} \sum_{j=1}^{\dim(\mathcal{A})} \left[ \gamma^t \cdot \nabla_{\pi_2}(\pi_2(s_t) - \pi_1(s_t))^T \mathbf{e}_j \cdot \frac{A^{\pi_1}(s_t, \pi_1(s_t) + \sigma \mathbf{e}_j)}{\sigma} \right].
\tag{52}
$$

Since, $A^{\pi_1}(s_t, \pi_1(s_t)) = 0$, we can write

$$
\begin{aligned}
\lim_{\sigma \to 0} \frac{A^{\pi_1}(s_t, \pi_1(s_t) + \sigma \mathbf{e}_j)}{\sigma} &= \lim_{\sigma \to 0} \frac{A^{\pi_1}(s_t, \pi_1(s_t) + \sigma \mathbf{e}_j) - A^{\pi_1}(s_t, \pi_1(s_t))}{\sigma} \\
&= \frac{\partial}{\partial a_t^{(j)}} A^{\pi_1}(s_t, a_t)\big|_{a_t=\pi_1(s_t)}.
\end{aligned}
\tag{53}
$$

Also, by the definition of the gradient, we can write

$$
\nabla_{\pi_2}(\pi_2(s_t) - \pi_1(s_t))^T \mathbf{e}_j = \nabla_{\pi_2} a_t^{(j)}.
\tag{54}
$$

Combining Equations (53) and (54), and applying them to Equation (52), yields

$$
\lim_{\sigma \to 0} \nabla_{\pi_2} \mathbb{A}^{\pi_1}(\pi_2)\big|_{\pi_2=\pi_1} = \sum_{t=0}^{H-1} \sum_{j=1}^{\dim(\mathcal{A})} \gamma^t \cdot \nabla_{\pi_2} a_t^{(j)} \cdot \frac{\partial}{\partial a_t^{(j)}} A^{\pi_1}(s_t, a_t)\big|_{\substack{a_t=\pi_2(s_t)\\ \pi_2=\pi_1}}.
\tag{55}
$$

Finally, the theorem can be obtained by comparing Equations (55) and (50).

$\qquad\qquad\qquad\qquad\qquad\qquad\qquad\qquad\qquad\qquad\qquad\qquad\qquad\qquad\qquad\qquad\qquad\qquad$ $\square$

## B.8 The Discussion of Assumptions

There are three key groups of assumptions made in the derivation of our policy improvement lower bound. First is the existence of $Q^\pi$-function Lipschitz constants. Second is the transition dynamics Lipschitz-continuity assumptions. Finally, we make an assumption about the stability of the transition dynamics. Next, we will discuss the meaning and the necessity of these assumptions.

### B.8.1 On the Existence of the $\mathrm{Lip}(Q^\pi, a)$ Constant

The $\mathrm{Lip}(Q^\pi, a)$ constant may be undefined when either the reward function or the transition dynamics are discontinuous. Examples of known environments with undefined $\mathrm{Lip}(Q^\pi, a)$ constants include those with grazing contacts which define a discontinuous transition dynamics. In practice, even for environments that do not satisfy Lipschitz continuity assumptions, there are mitigating factors; practical $Q^\pi$ functions are reasonably narrow-bounded in a small trust-region neighborhood, and since we use non-vanishing exploration scales and trust regions, a bounded interpolation slope can still model the $Q$-function variation effectively. We should also note that a slightly stronger version of this assumption is frequently used in the context of Lipschitz MDPs [45, 48, 5]. In practice, we have not found this to be a substantial limitation.

### B.8.2 The Transition Dynamics Lipschitz Continuity Assumption

Assumptions 4 and 5 of the main paper essentially represent the Lipschitz continuity assumptions of the transition dynamics with respect to actions and states, respectively. If the transition dynamics and the policy are deterministic, then these assumptions are exactly equivalent to the Lipschitz continuity assumptions. Assumptions 4 and 5 only generalize the Lipschitz continuity assumptions in a distributional sense.

The necessity of these assumptions is a consequence of using metric measures for bounding errors. Traditional non-metric bounds force the use of full-support stochastic policies where all actions have non-zero probabilities (e.g., for the KL-divergence of two policies to be defined, TRPO needs to operate on full-support policies such as the Gaussian policies). In those analyses, since all policies share the same support, the next state distribution automatically becomes smooth and Lipschitz continuous with respect to the policy measure even if the transition dynamics were not originally smooth with respect to the input actions. However, metric measures are also defined for policies of non-overlapping support. To be able to provide closeness bounds for future state visitations of two similar policies with non-overlapping support, it becomes necessary to assume that close-enough actions or states must be yielding close-enough next states. In fact, this is a very common assumption in the framework of Lipschitz MDPs (See Section 2.2 of Rachelson and Lagoudakis [48], Section 3 of Asadi et al. [5], and Assumption 1 of Pirotta et al. [45]).

### B.8.3 The Transition Dynamics Stability Assumption

Before moving to relax the $\gamma L_\mu < 1$ assumption, we will make a few definitions and restate the previous lemmas and theorems under these definitions. We define $L_{\mu_1, \mu_2, \pi}$ to be the infimum non-negative value that makes the equation $W(\mathbb{P}(\mu_1, \pi), \mathbb{P}(\mu_2, \pi)) = L_{\mu_1, \mu_2, \pi} W(\mu_1, \mu_2)$ hold. Similarly, $L_{\mu_1, \mu_2, \pi}$ is defined as the infimum non-negative value that makes the equation $W(\mathbb{P}(\mu, \pi_1), \mathbb{P}(\mu, \pi)) = L_{\mu, \pi_1, \pi_2} W(\pi_1(\cdot|\mu), \pi_2(\cdot|\mu))$ hold. For notation brevity, we will also denote $L_{\mathbb{P}^t(\mu,\pi_1), \mathbb{P}^t(\mu,\pi_2), \pi_2}$ and $L_{\mathbb{P}^t(\mu,\pi_1), \pi_1, \pi_2}$ by $\tilde{L}^{(t)}_{\mu, \pi_1, \pi_2}$ and $\hat{L}^{(t)}_{\mu, \pi_1, \pi_2}$, respectively.

Under these definitions, Lemma B.1 evolves into

$$W(\mathbb{P}(\mu_1, \pi_1), \mathbb{P}(\mu_2, \pi_2)) \leq L_{\mu_1, \mu_2, \pi} W(\mu_1, \mu_2) + L_{\mu_1, \pi_1, \pi_2} W(\pi_1, \pi_2). \tag{56}$$

We can apply a time-point recursion to this lemma and have

$$W(\mathbb{P}(\mathbb{P}^t(\mu, \pi_1), \pi_1), \mathbb{P}(\mathbb{P}^t(\mu, \pi_2), \pi_2))$$
$$\leq L_{\mathbb{P}^t(\mu,\pi_1), \pi_1, \pi_2} W(\pi_1, \pi_2) + L_{\mathbb{P}^t(\mu,\pi_1), \mathbb{P}^t(\mu,\pi_2), \pi_2} W(\mathbb{P}^t(\mu, \pi_1), \mathbb{P}^t(\mu, \pi_2)) \tag{57}$$

, which can be notationally simplified to

$$W(\mathbb{P}^t(\mu, \pi_1), \mathbb{P}^t(\mu, \pi_2)) \leq \hat{L}^{(t-1)}_{\mu, \pi_1, \pi_2} W(\pi_1, \pi_2) + \tilde{L}^{(t-1)}_{\mu, \pi_1, \pi_2} W(\mathbb{P}^{t-1}(\mu, \pi_1), \mathbb{P}^{t-1}(\mu, \pi_2)). \tag{58}$$

These modifications lead Lemma B.2 to be updated accordingly into

$$W(\mathbb{P}^t(\mu, \pi_1), \mathbb{P}^t(\mu, \pi_2)) \leq C^{(t)}_{L; \mu, \pi_1, \pi_2} \cdot W(\pi_1, \pi_2) \tag{59}$$

, where we have

$$C^{(t)}_{L; \mu, \pi_1, \pi_2} := \sum_{k=1}^{t} \hat{L}^{(t-k)}_{\mu, \pi_1, \pi_2} \prod_{i=1}^{k-1} \tilde{L}^{(t-i)}_{\mu, \pi_1, \pi_2}. \tag{60}$$

By a simple change of variables, we can have the equivalent definition of

$$C_{L;\mu,\pi_1,\pi_2}^{(t)} := \sum_{k=1}^{t} \hat{L}_{\mu,\pi_1,\pi_2}^{(k-1)} \prod_{i=k+1}^{t-1} \tilde{L}_{\mu,\pi_1,\pi_2}^{(i)}. \tag{61}$$

Now, we would replace the $\gamma L_\mu < 1$ assumption with the following assumption.

**The Transition Dynamics Stability Assumption**: A transition dynamics $P$ is called stable if and only if the induced $\{\tilde{L}_{\mu,\pi_1,\pi_2}^{(t)}\}_{t\geq 0}$ and $\{\hat{L}_{\mu,\pi_1,\pi_2}^{(t)}\}_{t\geq 0}$ sequences satisfy

$$C_L := \sup_{\mu,\pi_1,\pi_2,t} C_{L;\mu,\pi_1,\pi_2}^{(t)} = \sup_{\mu,\pi_1,\pi_2,t} \sum_{k=1}^{t} \hat{L}_{\mu,\pi_1,\pi_2}^{(k-1)} \prod_{i=k+1}^{t-1} \tilde{L}_{\mu,\pi_1,\pi_2}^{(i)} < \infty. \tag{62}$$

To help understand which $\{\tilde{L}_{\mu,\pi_1,\pi_2}^{(t)}\}_{t\geq 0}$ and $\{\hat{L}_{\mu,\pi_1,\pi_2}^{(t)}\}_{t\geq 0}$ sequences can satisfy this assumption, we will provide some examples:

- Having $\forall t : \tilde{L}_{\mu,\pi_1,\pi_2}^{(t)} = c_1 > 1, \hat{L}_{\mu,\pi_1,\pi_2}^{(t)} = c_2$ violates the dynamics stability assumption.
- Having $\forall t : \tilde{L}_{\mu,\pi_1,\pi_2}^{(t)} \leq 1, \hat{L}_{\mu,\pi_1,\pi_2}^{(t)} = O(\frac{1}{t^2})$ sequences satisfy the dynamics stability assumption.
- Having $\sup_t \tilde{L}_{\mu,\pi_1,\pi_2}^{(t)} < 1$ guarantees the dynamics stability assumption.
- Having $\forall t \geq t_0 : \tilde{L}_{\mu,\pi_1,\pi_2}^{(t)} < 1$ guarantees the dynamics stability assumption no matter (1) how big $t_0$ is (as long as it is finite), or (2) how big the members of the finite set $\{\tilde{L}_{\mu,\pi_1,\pi_2}^{(t)}|t < t_0\}$ are.

If the dynamics stability assumption holds with a constant $C_L$, one can define a $\bar{L}_\mu$ constant such that $C_L = L_\pi \sum_{t=0}^{\infty} (\gamma \bar{L}_\mu)^t$. Then, we can replace all the $L_\mu$ instances in the rest of the proof with the corresponding $\bar{L}_\mu$ constant, and the results will remain the same without any change of format.

The $\tilde{L}_{\mu,\pi_1,\pi_2}^{(t)}$ and $\hat{L}_{\mu,\pi_1,\pi_2}^{(t)}$ constants can be thought as tighter versions of $L_\mu$ and $L_\pi$, but with dependency on $\pi_1$, $\pi_2$, $\mu$ and the time-point of application. Having $\gamma L_\mu < 1$ is a sufficient yet unnecessary condition for this dynamics stability assumption to hold. Vaguely speaking, $L_\mu$ is an expansion rate for the state distribution distance; it tells you how much the divergence in the state distribution will expand after a single application of the transition dynamics. Having effective expansion rates that are larger than one throughout an infinite horizon trajectory is a sign of the system instability; some change in the initial state's distribution could cause the observations to diverge exponentially. While controlling unstable systems is an important and practical challenge, none of the existing reinforcement learning methods is capable of learning effective policies in such environments. Roughly speaking, having the dynamics stability assumption guarantees that the expansion rates cannot be consistently larger than one for infinite time-steps.

# C Implementation Details and Supplementary Results

## C.1 Implementation Details for the Environment with Non-local Rewards

We used the stable-baselines implementation [20], which has the same structure as the original OpenAI baselines [11] implementation. We used the "ppo1" variant since no hardware acceleration was necessary for automatic differentiation and MPI parallelization was practically efficient. TDPO, TRPO, and PPO used the same function approximation architecture with two hidden layers, 64 units in each layer, and the tanh activation. TRPO, PPO, DDPG, and TD3 used their default hyper-parameter settings. We used the same method of network initialization as TRPO and PPO (Xavier initialization [16] with default gains for the inner layers, and smaller gain for the output layer). TD3's baseline implementation was amended to support MPI parallelization just like TRPO, PPO, and DDPG. We performed one-dimensional hyper-parameter tuning for DDPG and TD3 both with and without the tanh final activation function that is common for DDPG and TD3 (this causes the difference in initial payoff in the figures). We could not find a consistent training payoff improvement over the default setting, so we used the default hyper-parameters of DDPG and TD3 in our experiments. Mini-batch selection was unnecessary for TDPO since optimization for samples generated by DeVine was fully tractable. The confidence intervals in all figures were generated using 1000 samples of the statistics of interest.

For designing the environment, we used Dhariwal et al. [11]'s pendulum dynamics and relaxed the torque thresholds to be as large as $80$ N m. The environment also had the same episode length of 200 time-steps. We used the reward function described by the following equations:

$$
\begin{aligned}
R(s_t, a_t) &= C_R \cdot R(\tau) \cdot \mathbf{1}\{t = 200\} \\
R(\tau) &= R_{\text{Freq}}(\tau) + R_{\text{Offset}}(\tau) + R_{\text{Amp}}(\tau) \\
R_{\text{Freq}}(\tau) &= 0.1 \cdot \left[ \sum_{f=f_{\min}}^{f_{\max}} \Theta_{\text{std}}^{+}(f)^2 - 1 \right] \\
R_{\text{Offset}}(\tau) &= -\left| \frac{\Theta(f = 0)}{200} - \theta_{\text{Target Offset}} \right| = -\left| \left( \frac{1}{200} \sum_{t=0}^{199} \theta_t \right) - \theta_{\text{Target Offset}} \right| \\
R_{\text{Amp}}(\tau) &= h_{\text{piecewise}} \left( \frac{\Theta_{\text{AC}}}{\theta_{\text{Target Amp.}}} - 1 \right)
\end{aligned}
\tag{63}
$$

where

- $\theta$ is the pendulum angle signal in the time domain.
- $\Theta$ is the magnitude of the Fourier transform of $\theta$.
- $\Theta^{+}$ is the same as $\Theta$ only for the positive frequency components.
- $\Theta_{\text{AC}}$ is the normalized oscillatory spectrum of $\Theta$:

$$
\Theta_{\text{AC}} = \frac{\sqrt{\Theta^{+T}\Theta^{+}}}{200}.
\tag{64}
$$

- $h_{\text{piecewise}}$ is a piece-wise linear error penalization function:

$$
h_{\text{piecewise}}(x) = -x \cdot \mathbf{1}\{x \geq 0\} + 10^{-4}x \cdot \mathbf{1}\{-x \geq 0\}.
\tag{65}
$$

- $\Theta_{\text{std}}^{+}$ is the standardized positive amplitudes vector:

$$
\Theta_{\text{std}}^{+} = \frac{\Theta^{+}}{\sqrt{\Theta^{+T}\Theta^{+} + 10^{-6}}}.
\tag{66}
$$

- $C_R = 1.3 \times 10^4$ is a reward normalization coefficient and was chosen to yield approximately the same payoff as a null policy would yield in the typical pendulum environment of Dhariwal et al. [11].

- $\theta_{\text{Target Offset}}$ is the target offset, $\theta_{\text{Target Amp.}}$ is the target amplitude, and $[f_{\min}, f_{\max}]$ is the target frequency range of the environment.

All methods used 48 parallel workers. The machines used Xeon E5-2690-v3 processors and 256 GB of memory. Each experiment was repeated 25 times for each method, and each run was given 6 hours or 500 million samples to finish.

## C.2 Implementation Details for the Environment with Long Horizon and Resonant Frequencies

For the robotic leg, we used exactly the same algorithms with the same parameters as described in Section C.1 above.

We used the reward function described by the following equations:

$$R = R_{\text{posture}} + R_{\text{velocity}} + R_{\text{foot offset}} + R_{\text{foot height}} + R_{\text{ground force}} + R_{\text{knee height}} + R_{\text{on-air torques}} \quad (67)$$

with

$$
\begin{aligned}
R_{\text{posture}} &= -1 \times \left[ \left| \theta_{\text{knee}} + \frac{\pi}{2} \right| + \left| \theta_{\text{hip}} + \frac{\pi}{4} \right| \right] \\
R_{\text{velocity}} &= -0.08 \times \left[ |\omega_{\text{knee}}| + |\omega_{\text{hip}}| \right] \\
R_{\text{foot offset}} &= -10 \times \left[ |x_{\text{foot}}| \cdot \mathbf{1}\{ z_{\text{knee}} < 0.2 \} \right] \\
R_{\text{ground force}} &= -1 \times \left[ |f_z - mg| \cdot \mathbf{1}\{ f_z < mg \} \cdot \mathbf{1}_{\text{touchdown}} \right] \\
R_{\text{foot height}} &= -1 \times \left[ |z_{\text{foot}}| \cdot \mathbf{1}_{\text{touchdown}} \right] \\
R_{\text{knee height}} &= -15 \times \left[ \left| z_{\text{knee}} - z_{\text{knee}}^{\text{target}} \right| \cdot \mathbf{1}_{\text{touchdown}} \right] \\
R_{\text{on-air torques}} &= -10^{-4} \times \left[ (\tau_{\text{knee}}^2 + \tau_{\text{hip}}^2) \cdot (1 - \mathbf{1}_{\text{touchdown}}) \right] \quad (68)
\end{aligned}
$$

where

- $\theta_{\text{knee}}$ and $\theta_{\text{hip}}$ are the knee and hip angles in radians, respectively.
- $\omega_{\text{knee}}$ and $\omega_{\text{hip}}$ are the knee and hip angular velocities in radians per second, respectively.
- $x_{\text{foot}}$ and $z_{\text{foot}}$ are the horizontal and vertical foot offsets in meters from the desired standing point on the ground, respectively.
- $x_{\text{knee}}$ and $z_{\text{knee}}$ are the horizontal and vertical knee offsets in meters from the desired standing point on the ground, respectively.
- $f_z$ is the vertical ground reaction force on the robot in Newtons.
- $m$ is the robot weight in kilograms (i.e., $m = 0.76$ kg).
- $g$ is the gravitational acceleration in meters per second squared.
- $\mathbf{1}_{\text{touchdown}}$ is the indicator function of whether the robot has ever touched the ground.
- $z_{\text{knee}}^{\text{target}}$ is a target knee height of 0.1 m.
- $\tau_{\text{knee}}$ and $\tau_{\text{hip}}$ are the knee and hip torques in Newton meters, respectively.

All methods used 72 full trajectories between each policy update, and each run was given 16 hours of wall time, which corresponded to almost 500 million samples. This experiment was repeated 75 times for each method. The empirical means of the discounted payoff values were reported without any performance or seed filtration. The same hardware as the non-local rewards experiments (i.e., Xeon E5-2690-v3 processors and 256 GB of memory) was used.

## C.3 Gym Suite Benchmarks

While it is clear that our deterministic policy gradient performs well on the new control environments we consider, one may naturally wonder about its performance on existing RL control benchmarks. To show our method's core capability, we ran our method on a suite of Gym environments and include six representative examples in Figure 9. Broadly speaking, our method (TDPO) performs similar to or slightly worse than others, but occasionally performs better as seen in the Swimmer-v3 environment.

We speculate that many of these gym environments are reasonably robust to any injected noise, and this may mean that stochastic policy gradients can more rapidly and efficiently explore the policy space than in our new control environments.

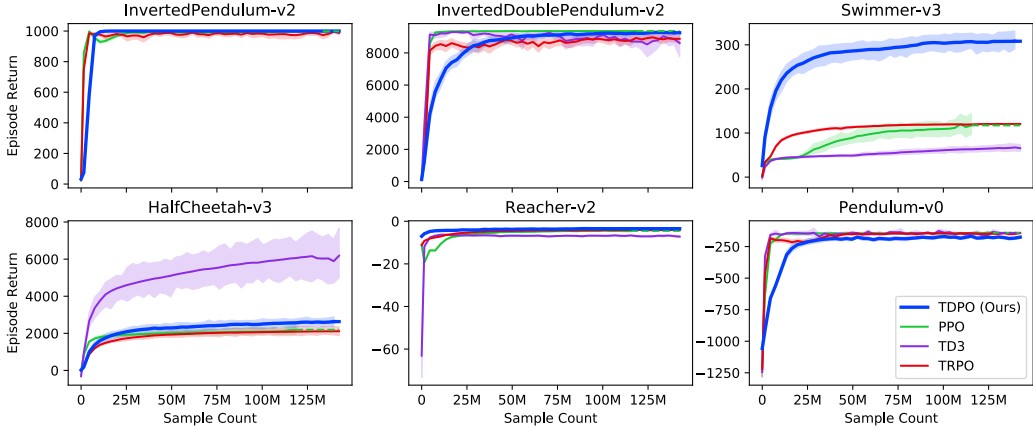

Figure 9: Results for the gym suite benchmarks. In this setting, we ran our method, TDPO, without line search or adaptive exploration parameters. The experiments granted each method 72 parallel MPI workers for about 144 million steps (i.e., 2 million sequential steps), and the returns were averaged over 100 different seeds for each method. Since the computational cost of running both DDPG and TD3 was high, we only included TD3.

## C.4    Running Time Comparison

Figure 10 depicts a comparison of each method's running time per million steps. These plots show the combination of both the simulation (i.e., environment sampling) and the optimization (i.e., computing the policy gradient and running the conjugate gradient solver) time. Our method's DeVine gradient estimator summarizes two full trajectories into a single state-action-advantage tuple, which saves on computational resources and makes our method faster. That being said, these relative comparisons could vary to a large extent (1) under different processor architectures, (2) with more (or less) efficient implementations, or (3) when running environments whose simulation time constitutes a significantly larger (or smaller) portion of the total running time.

## C.5    Other Swinging Pendulum Variants

Multiple variants of the pendulum with non-local rewards were used, each with different frequency targets and the same reward structure. Table 4 summarizes the target characteristics of each variant. The main variant was shown in the paper. Figures 11, 12, 13, 14, 15, 17, 18, and 19 show similar results for the second to ninth variants. Also, we show the performance of our method (TDPO) on all variants in Figure 16.

## C.6    Non-locality or the Frequency-based Nature of the Pendulum Reward?

Solving the frequency-domain problem of Figure 1 in the main paper is challenging. The frequency-based nature of the reward may be the main contributor to this difficulty. Alternatively, the non-locality of the reward may be playing the key factor in this challenge. To investigate this, we simulate a typical MDP reward on the same pendulum model, and make it artificially non-local by accumulating the rewards every 2, 20, or 200 time-steps. Figure 20 shows the training curves for each case. Other methods perform better than ours when the rewards are only accumulated for 2 time-steps. However, our method is more resilient to higher non-locality of the reward. Since this reward is not defined in the frequency domain, yet is still challenging to solve, we speculate that the non-locality of the reward is a more influential factor than the frequency-based nature of the reward used in Figure 1. This explanation is consistent with the observation that a 5-fold change in the desired frequency between the main paper's variant, and the second and the third variants shown in Figures 11 and 12 could not resolve the issue for the existing methods.

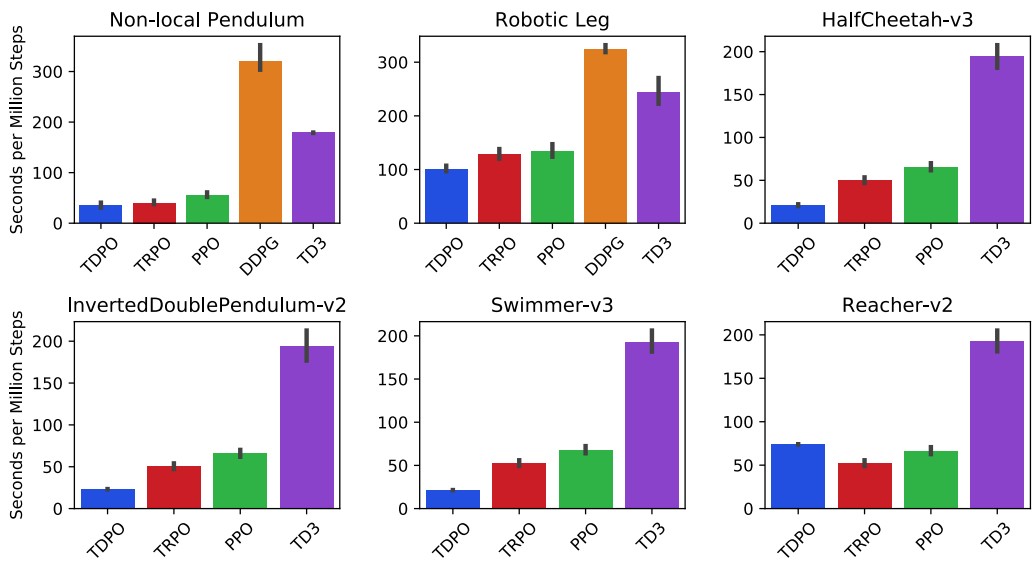

Figure 10: Training time comparison in different environments. The lower the bar, the faster the method. The vertical axis shows the time in seconds needed to consume one million state-action pairs for training. Each environment was shown separately in a different subplot.

| Pendulum Variant | Desired Frequency | Desired Offset | Desired Amplitude |
|---|---|---|---|
| Main | 1.7–2 Hz | 0.524 rad | 0.28 rad |
| Second | 0.5–0.7 Hz | 1.571 rad | 1.11 rad |
| Third | 2.5–3 Hz | 0.524 rad | 0.28 rad |
| Fourth | 2–2.4 Hz | 0.785 rad | 0.28 rad |
| Fifth | 2–2.4 Hz | 1.571 rad | 0.74 rad |
| Sixth | 2–2.4 Hz | 0.524 rad | 0.28 rad |
| Seventh | 2–2.4 Hz | 1.047 rad | 0.28 rad |
| Eighth | 2–2.4 Hz | 0.785 rad | 0.74 rad |
| Ninth | 2–2.4 Hz | 1.309 rad | 0.28 rad |

Table 4: The target oscillation characteristics defining different pendulum swinging environments.

### C.7 On the Interpretation of the Surrogate Function

For deterministic policies, the squared Wasserstein distance $W(\pi_2(a|s), \pi_1(a|s))^2$ degenerates to the Euclidean distance over the action space. Any policy defines a sensitivity matrix at a given state $s$, which is the Jacobian matrix of the policy output with respect to $s$. The policy sensitivity term $\mathcal{L}_{G^2}(\pi_1, \pi_2; s)$ is essentially the squared Euclidean distance over the action-to-observation Jacobian matrix elements. In other words, our surrogate prefers to step in directions where the action-to-observation sensitivity is preserved within updates.

Although our surrogate uses a metric distance instead of the traditional non-metric measures for regularization, we do not consider this sole replacement a major contribution. The squared Wasserstein distance and the KL divergence of two identically-scaled Gaussian distributions are the same up to a constant (i.e., $D_{\mathrm{KL}}(\mathcal{N}(m_1, \sigma) \| \mathcal{N}(m_2, \sigma)) = W(\mathcal{N}(m_1, \sigma), \mathcal{N}(m_2, \sigma))^2/2\sigma^2$). On the other hand, our surrogate's compatibility with deterministic policies makes it a valuable asset for our policy gradient algorithm; both $W(\pi_2(a|s), \pi_1(a|s))^2$ and $\mathcal{L}_{G^2}(\pi_1, \pi_2; s)$ can be evaluated for two deterministic policies $\pi_1$ and $\pi_2$ numerically without any approximations to overcome singularities.

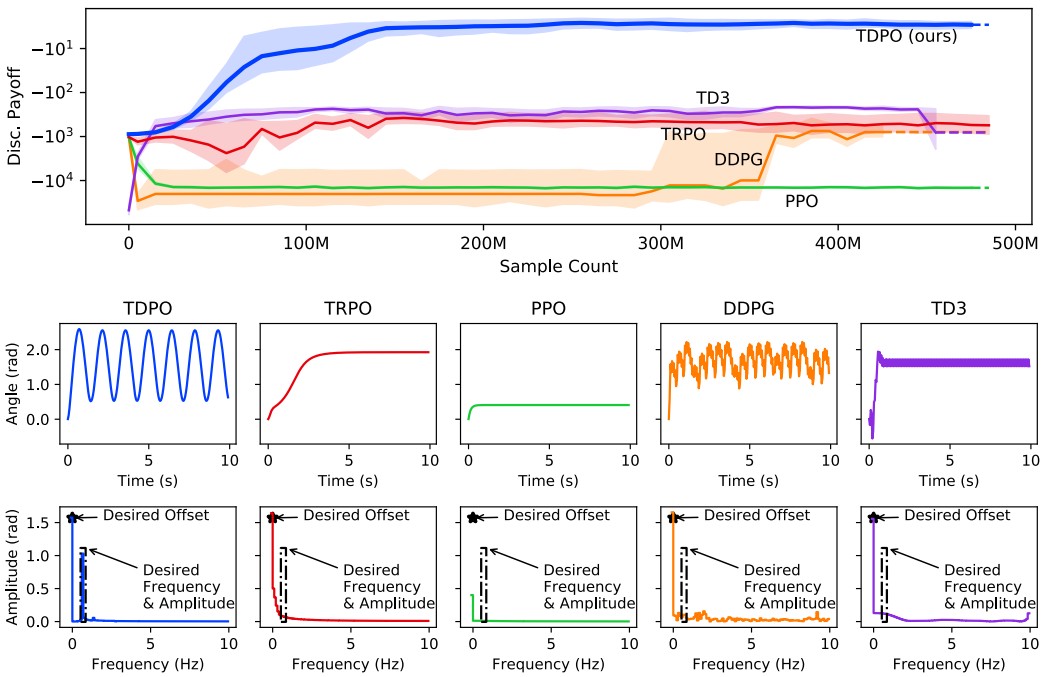

Figure 11: Results for the second variant of the simple pendulum with non-local rewards. Upper panel: training curves with empirical discounted payoffs. Lower panels: trajectories in both the time domain and frequency domain, showing target values of oscillation frequency, amplitude, and offset.

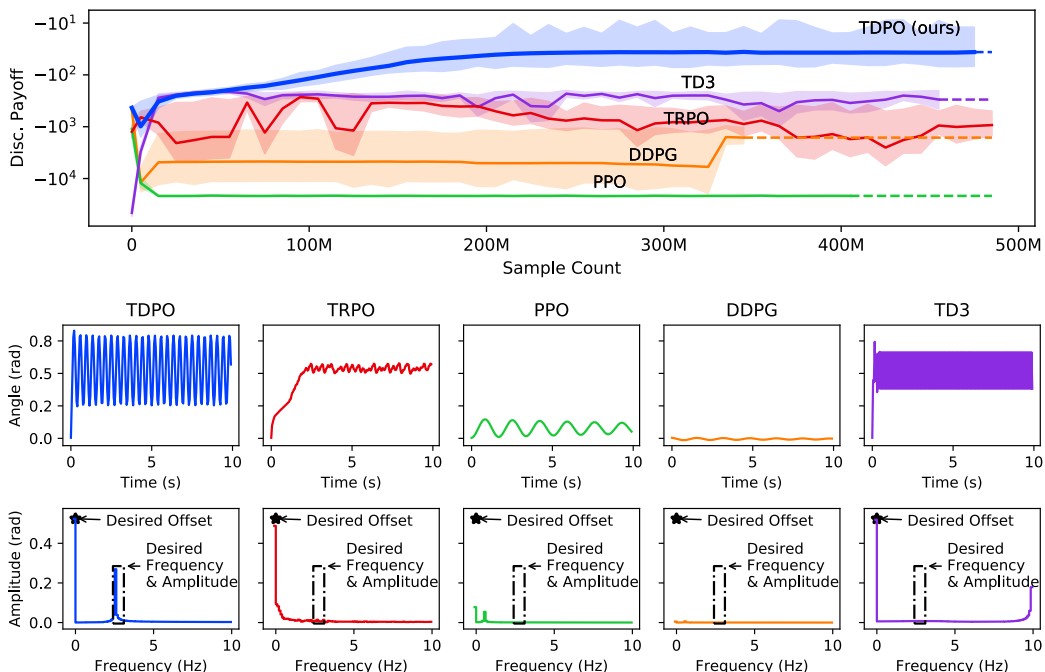

Figure 12: Results for the third variant of the simple pendulum with non-local rewards. Upper panel: training curves with empirical discounted payoffs. Lower panels: trajectories in both the time domain and frequency domain, showing target values of oscillation frequency, amplitude, and offset.

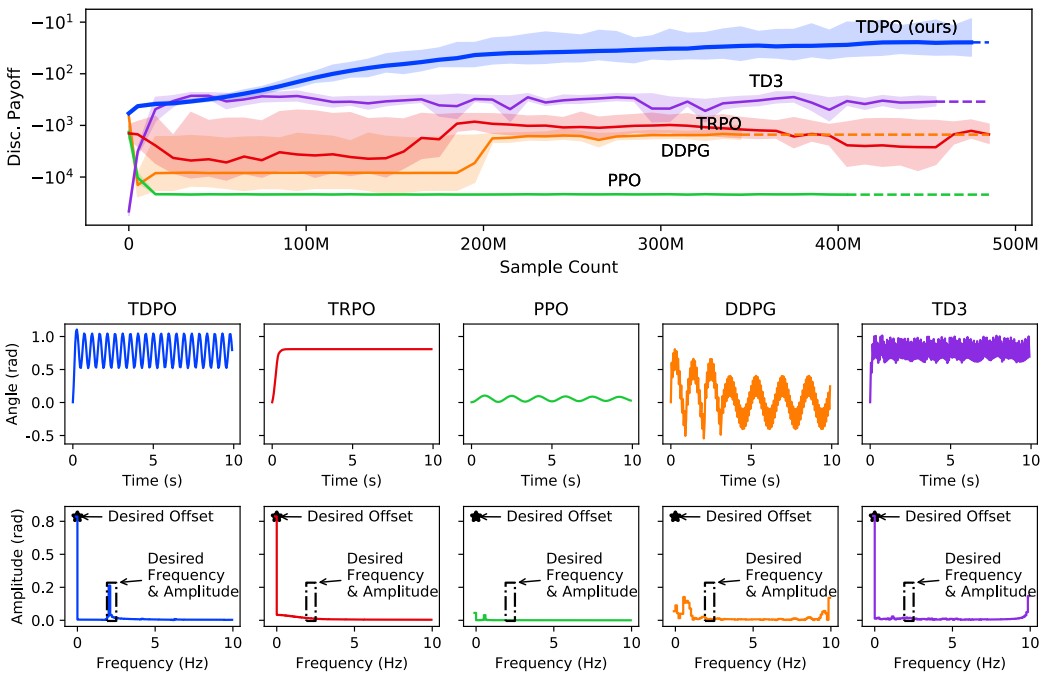

Figure 13: Results for the fourth variant of the simple pendulum with non-local rewards. Upper panel: training curves with empirical discounted payoffs. Lower panels: trajectories in both the time domain and frequency domain, showing target values of oscillation frequency, amplitude, and offset.

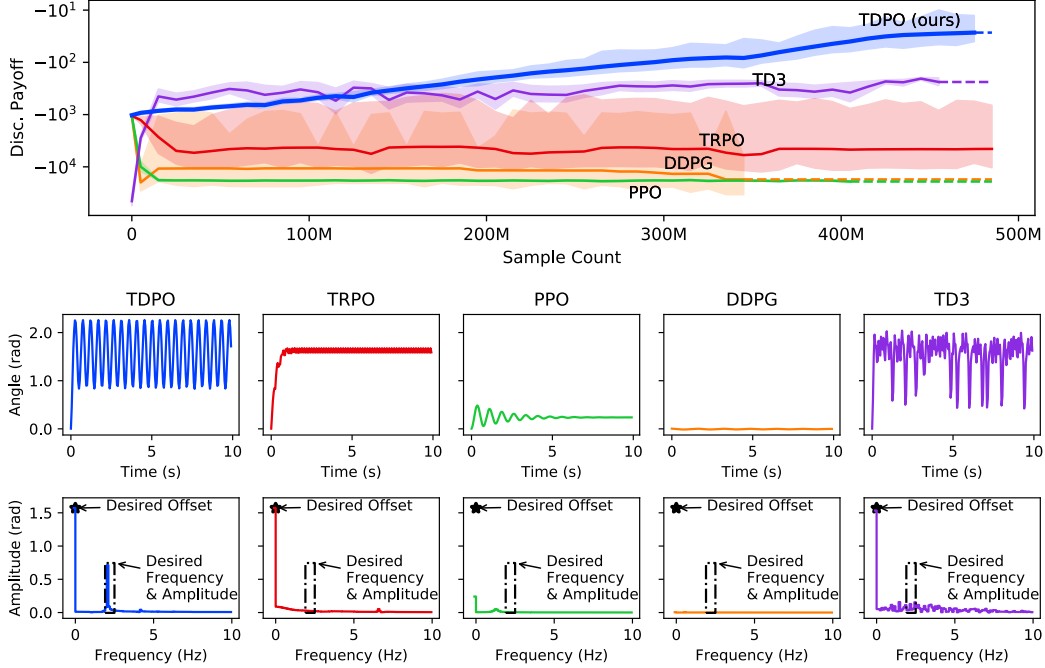

Figure 14: Results for the fifth variant of the simple pendulum with non-local rewards. Upper panel: training curves with empirical discounted payoffs. Lower panels: trajectories in both the time domain and frequency domain, showing target values of oscillation frequency, amplitude, and offset.

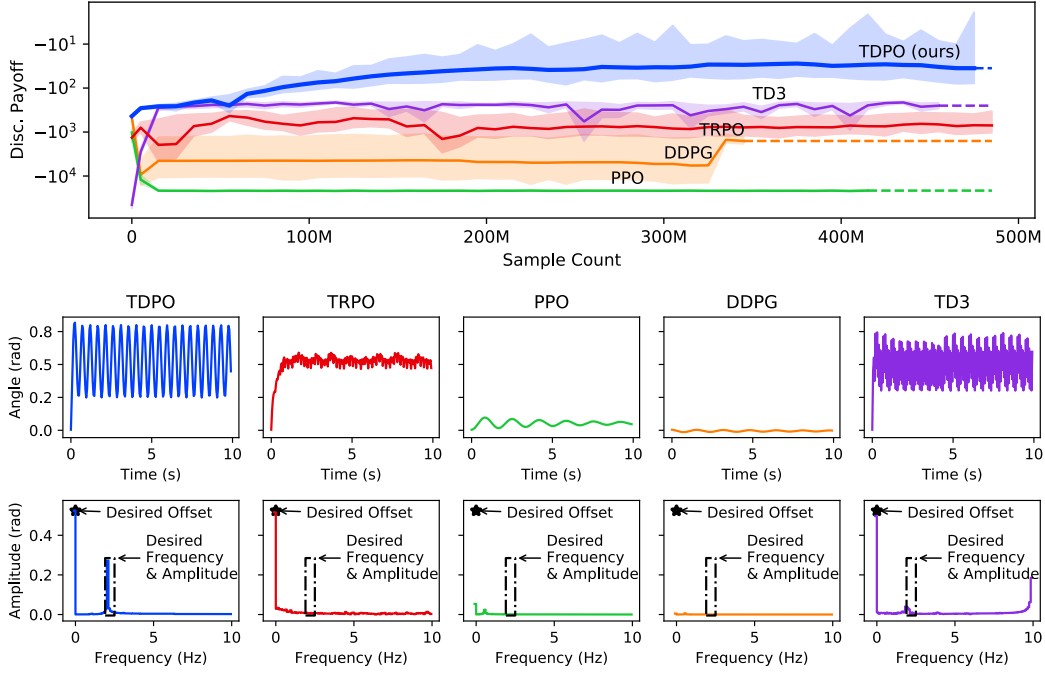

Figure 15: Results for the sixth variant of the simple pendulum with non-local rewards. Upper panel: training curves with empirical discounted payoffs. Lower panels: trajectories in both the time domain and frequency domain, showing target values of oscillation frequency, amplitude, and offset.

## C.8 Notes on How to Implement TDPO

In short, our method (TDPO) is structured in the same way TRPO was structured; both TDPO and TRPO use policy gradient estimation, and a conjugate-gradient solver utilizing a Hessian-vector product machinery. On the other hand, there are some algorithmic differences that distinguish the basic variant of TDPO from TRPO. TDPO uses the DeVine advantage estimator, which requires storing and reloading pseudo-random generator states. Furthermore, the Hessian-vector product machinery used in TDPO computes Wasserstein-vector products, which is slightly different from those used in TRPO. The hyper-parameter settings and notes on how to choose them were discussed in Sections C.1, C.9, and C.2. We will describe how to implement TDPO, and focus on the differences between TDPO and TRPO next.

As for the state-reset capability, our algorithm does not require access to a reset function for arbitrary states. Instead, we only require to be able to start from the prior trajectory's initial state. Many environments, including the Gym environments, instantiate their own pseudo-random generators and only utilize that pseudo-random generator for all randomized operations. This facilitates a straightforward implementation of the DeVine oracle; in such environments, implementing an arbitrary state-reset functionality is unnecessary, and only reloading the pseudo-random generator to its configuration prior to the trajectory would suffice. In other words, the DeVine oracle can store the initial configuration of the pseudo-random generator before asking for a trajectory reset and then start sampling. Once the main trajectory is finished, the pseudo-random generator can be reloaded, thus producing the same initial state upon a reset request. Other time-step states can then be recovered by applying the same proceeding action sequence.

To optimize the quadratic surrogate, the conjugate gradient solver was used. Implementing the conjugate gradient algorithm is fairly straightforward, and is already included in many common automatic differentiation libraries. The conjugate gradient solver is perfect for situations where (1) the Hessian matrix is larger than can efficiently be stored in the memory, and (2) the Hessian matrix includes many nearly identical eigenvalues. Both of these conditions apply for TDPO, as well as for TRPO. Instead of requiring the full Hessian matrix to be stored, the conjugate gradient solver only requires a Hessian-vector product machinery $v \to Hv$, which must be specifically implemented for

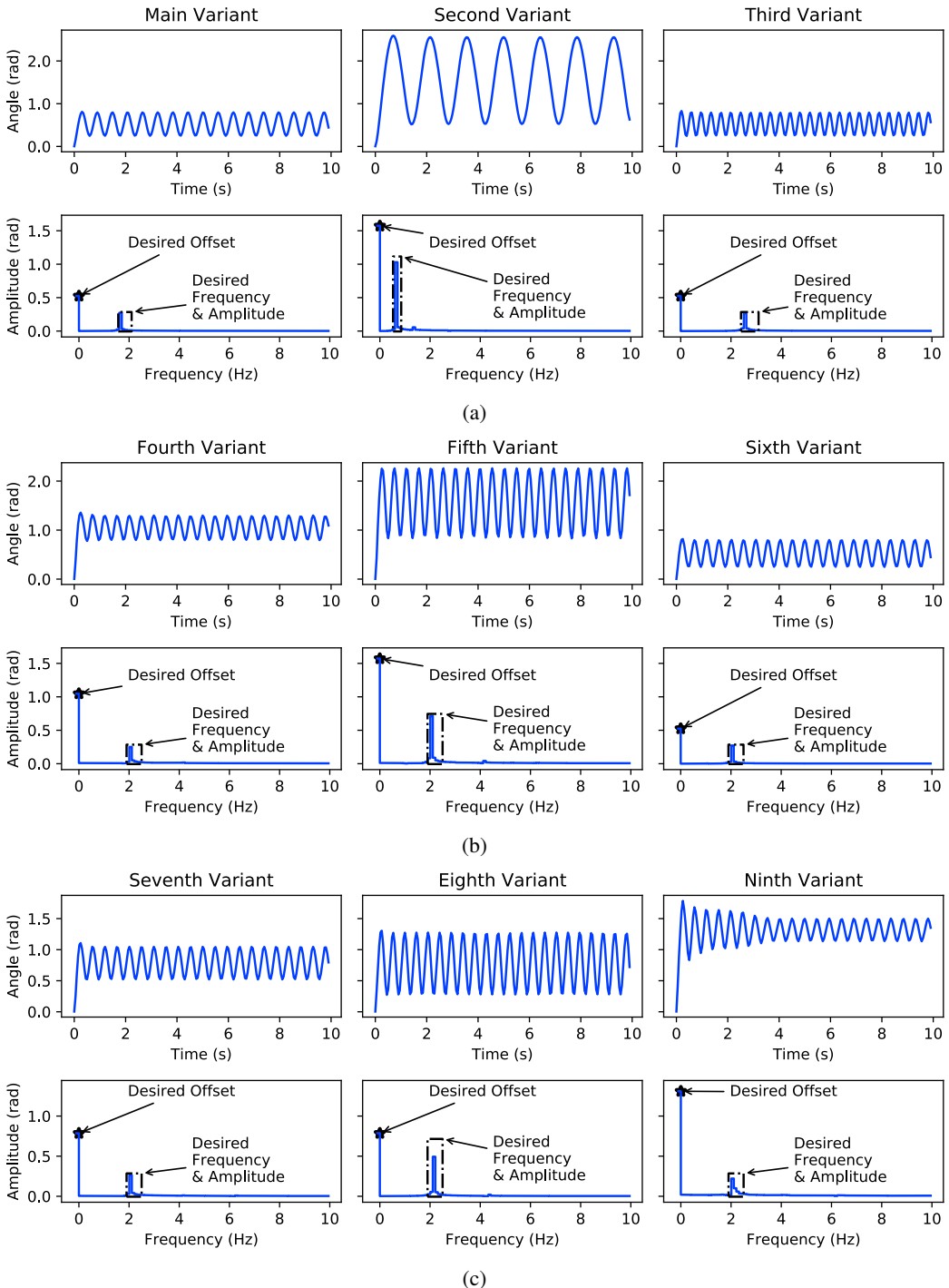

Figure 16: Time and frequency domain trajectories for our method (TDPO) on multiple variants of the simple pendulum with non-local rewards. (a) The high-reward trajectories for the first group of variants, (b) the high-reward trajectories for the second group of variants, and (c) the high-reward trajectories for the third group of variants. Target values of oscillation frequency, amplitude, and offset were annotated in the frequency domain plots.

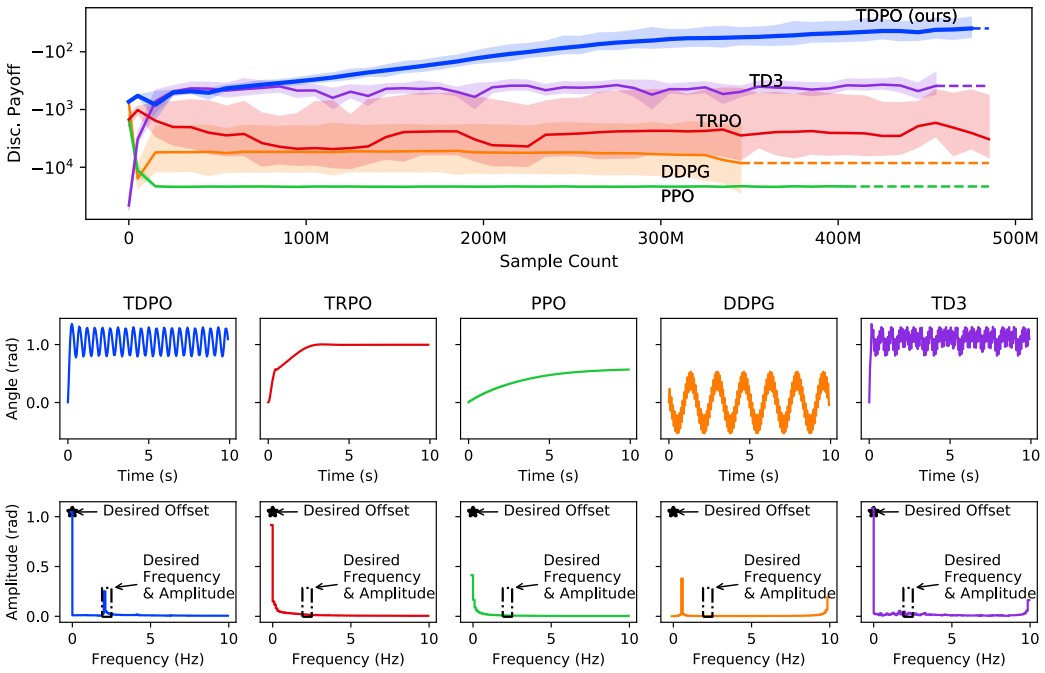

Figure 17: Results for the seventh variant of the simple pendulum with non-local rewards. Upper panel: training curves with empirical discounted payoffs. Lower panels: trajectories in both the time domain and frequency domain, showing target values of oscillation frequency, amplitude, and offset.

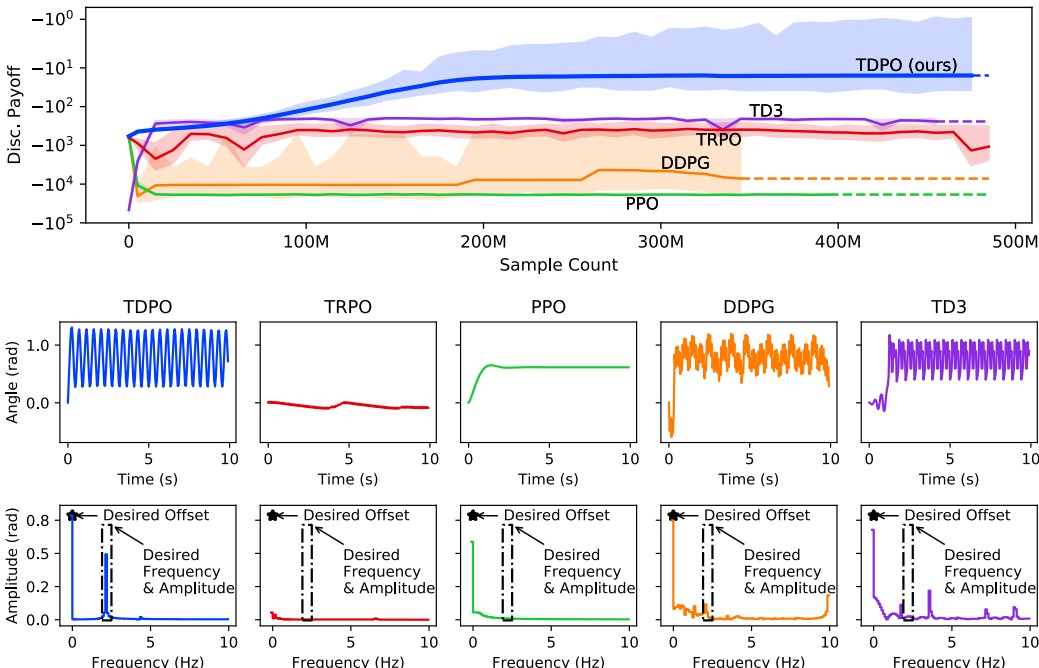

Figure 18: Results for the eighth variant of the simple pendulum with non-local rewards. Upper panel: training curves with empirical discounted payoffs. Lower panels: trajectories in both the time domain and frequency domain, showing target values of oscillation frequency, amplitude, and offset.

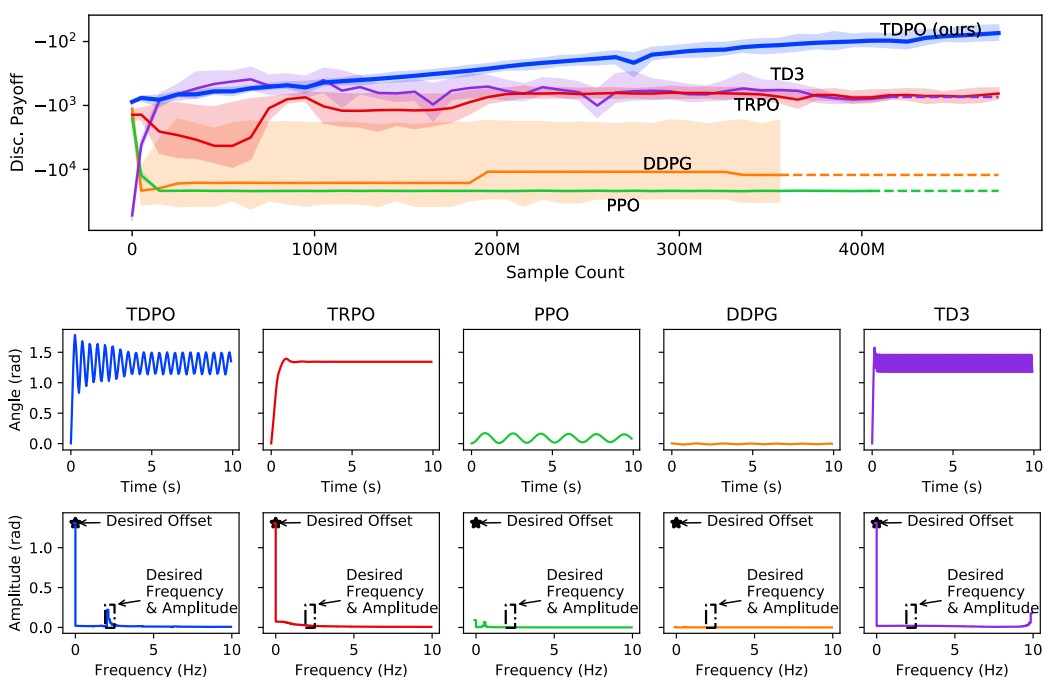

Figure 19: Results for the ninth variant of the simple pendulum with non-local rewards. Upper panel: training curves with empirical discounted payoffs. Lower panels: trajectories in both the time domain and frequency domain, showing target values of oscillation frequency, amplitude, and offset.

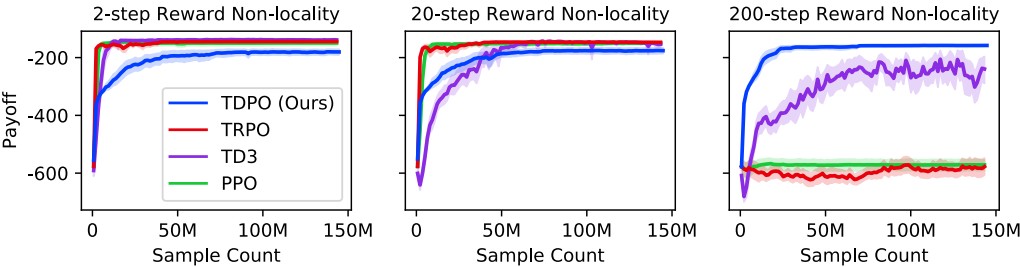

Figure 20: The training curves vs. different levels of reward non-locality for the continuous control pendulum benchmark in gym. The reward for the default standing-up task was used in here (rather than using the frequency-domain reward for the swinging pendulum). The trajectory length was set to 200 time-steps in all charts. To simulate non-locality of the reward, the rewards were accumulated every 2, 20, and 200 steps in the left, center, and the right chart, respectively, and then applied at once. For example, the trajectories used in the middle chart have 20 evenly-spaced non-zero rewards, and another 180 time-steps with zero rewards. The rewards were scaled so that all payoffs would be comparable.

TDPO. Our surrogate function can be viewed as

$$\mathcal{L}(\delta\theta) = g^T\delta\theta + \frac{C_2'}{2}\delta\theta^T H\delta\theta$$

where the Hessian matrix can be defined as

$$H = H_2 + \frac{C_1'}{C_2'}H_1, \qquad H_1 := \nabla_{\theta'}^2 \mathbb{E}_{s\sim\rho_\mu^{\pi_k}}\left[\mathcal{L}_{G^2}(\pi', \pi_k; s)\right],$$

$$H_2 := \nabla_{\theta'}^2 \mathbb{E}_{s\sim\rho_\mu^{\pi_k}}\left[W(\pi'(a|s), \pi_k(a|s))^2\right]. \tag{69}$$

---

**Algorithm 3** Wasserstein-Vector-Product Machinery

---

**Require:** Current Policy $\pi_1$ with parameters $\theta_1$.
**Require:** The vector $v$ with the same dimensions as $\theta_1$.
**Require:** An observation $s$.
1: Compute the action for the observation $s$ with $|A|$ elements.

$$a_{|A|\times 1} := \begin{bmatrix} \pi^{(1)}(s) \\ \vdots \\ \pi^{(|A|)}(s) \end{bmatrix}. \tag{70}$$

This vector should be capable of propagating gradients back to the policy parameters when used in automatic differentiation software.

2: Define $t$ to be a constant vector with the same shape as $a$. It could be populated with any values such as all ones.
3: Define the scalar $\tilde{a} := a^T t$.
4: Using back-propagation, find the gradient

$$\nabla_\theta \tilde{a} = \sum_{i=1}^{|A|} t_i \nabla_\theta a_i = \sum_{i=1}^{|A|} t_i \begin{bmatrix} \frac{\partial a_i}{\partial \theta_1} & \cdots & \frac{\partial a_i}{\partial \theta_{|\Theta|}} \end{bmatrix}. \tag{71}$$

5: Compute the following dot-product:

$$\langle \nabla_\theta \tilde{a}, v \rangle = \left( \sum_{i=1}^{|A|} t_i \cdot \frac{\partial a_i}{\partial \theta_1} \right) \cdot v_1 + \cdots + \left( \sum_{i=1}^{|A|} t_i \cdot \frac{\partial a_i}{\partial \theta_{|\Theta|}} \right) \cdot v_{|\Theta|}. \tag{72}$$

6: Using automatic differentiation, take the gradient w.r.t. the $t$ vector.

$$\tilde{a}_{\theta,v} := \nabla_t \langle \nabla_\theta \tilde{a}, v \rangle = \begin{bmatrix} \frac{\partial a_1}{\partial \theta_1} \cdot v_1 + \cdots + \frac{\partial a_1}{\partial \theta_{|\Theta|}} \cdot v_{|\Theta|} \\ \vdots \\ \frac{\partial a_{|A|}}{\partial \theta_1} \cdot v_1 + \cdots + \frac{\partial a_{|A|}}{\partial \theta_{|\Theta|}} \cdot v_{|\Theta|} \end{bmatrix} = \begin{bmatrix} \frac{\partial a_1}{\partial \theta_1} & \cdots & \frac{\partial a_1}{\partial \theta_{|\Theta|}} \\ \vdots & & \vdots \\ \frac{\partial a_{|A|}}{\partial \theta_1} & \cdots & \frac{\partial a^{|A|}}{\partial \theta_{|\Theta|}} \end{bmatrix} v \tag{73}$$

7: Compute the dot product $\langle \tilde{a}_{\theta,v}, \tilde{a} \rangle$.
8: Using back-propagation, take the gradient w.r.t. $\theta$, and return it as the gain-vector-product.

$$\nabla_\theta \langle \tilde{a}_{\theta,v}, \tilde{a} \rangle = \begin{bmatrix} \frac{\partial a_1}{\partial \theta_1} & \cdots & \frac{\partial a_1}{\partial \theta_{|\Theta|}} \\ \vdots & & \vdots \\ \frac{\partial a_{|A|}}{\partial \theta_1} & \cdots & \frac{\partial a^{|A|}}{\partial \theta_{|\Theta|}} \end{bmatrix}^T \begin{bmatrix} \frac{\partial a_1}{\partial \theta_1} & \cdots & \frac{\partial a_1}{\partial \theta_{|\Theta|}} \\ \vdots & & \vdots \\ \frac{\partial a_{|A|}}{\partial \theta_1} & \cdots & \frac{\partial a^{|A|}}{\partial \theta_{|\Theta|}} \end{bmatrix} v \tag{74}$$

---

In order to construct a Hessian-vector product machinery $v \to Hv$, one can design an automatic-differentiation procedure that returns the Hessian-vector product. Many automatic-differentiation packages already include functionalities that can provide a Hessian-vector product machinery of a given scalar loss function without computing the Hessian matrix. This can be used to implement the Hessian-vector product machinery in a straightforward manner; one only needs to provide the scalar quadratic terms of our surrogate and would obtain the Hessian-vector product machinery in return. On the other hand, this may not be the most computationally efficient approach, as our problem exhibits a more specific structure. Alternatively, one can implement a more elaborate and specifically designed Hessian-vector product machinery by following these three steps:

- Compute the Wasserstein-vector product $v \to H_2 v$ according to Algorithm 3.

- Compute the Sensitivity-vector product $v \to H_1 v$ according to Algorithm 4.

- Return the weighted sum of $H_1 v$ and $H_2 v$ as the final Hessian-vector product $Hv$.

**Algorithm 4** Sensitivity-Vector-Product Machinery

**Require:** Current Policy $\pi_1$ with parameters $\theta_1$.
**Require:** The vector $v$ with the same dimensions as $\theta_1$.
**Require:** An observation $s$.
1: Compute the action to observation Jacobian matrix

$$
J_{|A| \times |S|} := \begin{bmatrix} \frac{\partial \pi^{(1)}(s)}{\partial s^1} & \cdots & \frac{\partial \pi^{(1)}(s)}{\partial s^{(|S|)}} \\ \vdots & & \vdots \\ \frac{\partial \pi^{(|A|)}(s)}{\partial s^{(1)}} & \cdots & \frac{\partial \pi^{(|A|)}(s)}{\partial s^{|S|}} \end{bmatrix}. \tag{75}
$$

This can either be done using finite-differences in the observation using

$$
\frac{\partial \pi^{(i)}(s)}{\partial s^{(j)}} \simeq \frac{\pi^{(i)}(s + ds \cdot \mathbf{e_j}) - \pi^{(i)}(s)}{ds} \tag{76}
$$

(which may be numerically inaccurate), or using automatic differentiation. In any case, this matrix should be a parameter tensor capable of propagating gradients back to the parameters when used in automatic differentiation software.
2: Define $\tilde{J}$ to be the vectorized (i.e., reshaped into a column) $J$ matrix, with $|AS| = |A| \times |S|$ rows and one column.
3: Define $t$ to be a constant vector with the same shape as $\tilde{J}$. It could be populated with any values such as all ones.
4: Define the scalar $J_t := \tilde{J}^T t$.
5: Using back-propagation, find the gradient

$$
\nabla_\theta J_t = \sum_{i=1}^{|A|} \sum_{j=1}^{|S|} t_{i,j} \nabla_\theta J_{i,j} = \sum_{i=1}^{|A|} \sum_{j=1}^{|S|} t_{i,j} \begin{bmatrix} \frac{\partial J_{i,j}}{\partial \theta_1} & \cdots & \frac{\partial J_{i,j}}{\partial \theta_{|\Theta|}} \end{bmatrix}. \tag{77}
$$

6: Compute the following dot-product.

$$
\langle \nabla_\theta J_t, v \rangle = \left( \sum_{i=1}^{|A|} \sum_{j=1}^{|S|} t_{i,j} \cdot \frac{\partial J_{i,j}}{\partial \theta_1} \right) \times v_1 + \cdots + \left( \sum_{i=1}^{|A|} \sum_{j=1}^{|S|} t_{i,j} \cdot \frac{\partial J_{i,j}}{\partial \theta_{|\Theta|}} \right) \times v_{|\Theta|} \tag{78}
$$

7: Using automatic differentiation, take the gradient w.r.t. the $t$ vector.

$$
(\nabla_\theta J) v := \nabla_t \langle \nabla_\theta J_t, v \rangle = \begin{bmatrix} \frac{\partial J_{1,1}}{\partial \theta_1} \cdot v_1 + \cdots + \frac{\partial J_{1,1}}{\partial \theta_{|\Theta|}} \cdot v_{|\Theta|} \\ \vdots \\ \frac{\partial J_{|A|,|S|}}{\partial \theta_1} \cdot v_1 + \cdots + \frac{\partial J_{|A|,|S|}}{\partial \theta_{|\Theta|}} \cdot v_{|\Theta|} \end{bmatrix}
$$

$$
= \begin{bmatrix} \frac{\partial J^{(1,1)}}{\partial \theta_1} & \cdots & \frac{\partial J^{(1,1)}}{\partial \theta_{|\Theta|}} \\ \vdots & & \vdots \\ \frac{\partial J^{(|A|,|S|)}}{\partial \theta_1} & \cdots & \frac{\partial J^{(|A|,|S|)}}{\partial \theta_{|\Theta|}} \end{bmatrix} v \tag{79}
$$

8: Reshape $(\nabla_\theta J) v$ into a column vector and name it $\tilde{J}_{\theta,v}$.
9: Compute the dot product $\langle \tilde{J}_{\theta,v}, \tilde{J} \rangle$.
10: Using back-propagation, take the gradient w.r.t. $\theta$, and return it as the gain-vector-product.

$$
\nabla_\theta \langle \tilde{J}_{\theta,v}, \tilde{J} \rangle = \begin{bmatrix} \frac{\partial J^{(1,1)}}{\partial \theta_1} & \cdots & \frac{\partial J^{(1,1)}}{\partial \theta_{|\Theta|}} \\ \vdots & & \vdots \\ \frac{\partial J^{(|A|,|S|)}}{\partial \theta_1} & \cdots & \frac{\partial J^{(|A|,|S|)}}{\partial \theta_{|\Theta|}} \end{bmatrix}^T \begin{bmatrix} \frac{\partial J^{(1,1)}}{\partial \theta_1} & \cdots & \frac{\partial J^{(1,1)}}{\partial \theta_{|\Theta|}} \\ \vdots & & \vdots \\ \frac{\partial J^{(|A|,|S|)}}{\partial \theta_1} & \cdots & \frac{\partial J^{(|A|,|S|)}}{\partial \theta_{|\Theta|}} \end{bmatrix} v \tag{80}
$$

One may also need to add a conjugate gradient damping to the conjugate gradient solver (i.e., return $\beta v + Hv$ for some small $\beta$ as opposed to returning $Hv$), which is also done in the TRPO method. This may be important when the number of policy parameters is much larger than the sample size, which makes the $H$ matrix low-rank. Setting $\beta = 0$ may yield poor numerical stability if $H$ had small eigenvalues, and setting large $\beta$ will cause the conjugate gradient optimizer to mimic the gradient descent optimizer by making updates in the same direction as the gradient. The optimal conjugate gradient damping may depend on the problem and other hyper-parameters such as the sample size.

Once the conjugate gradient solver returned the optimal update direction $H^{-1}g$, it must be scaled down by a factor of $C_2'$ (i.e., $\delta\theta^* = H^{-1}g/C_2'$). If $\delta\theta^*$ satisfied the trust region criterion (i.e., $\frac{1}{2}\delta\theta^{*T} H \delta\theta^* \leq \delta_{\max}^2$), then one can make the parameter update (i.e., $\theta_{\text{new}} = \theta_{\text{old}} + \delta\theta^*$) and proceed to the next iteration. Otherwise, the proposed update $\delta\theta^*$ must be scaled down further, namely by $\alpha$, such that the trust region condition would be satisfied (i.e., $\frac{1}{2}(\alpha\delta\theta^*)^T H(\alpha\delta\theta^*) = \delta_{\max}^2$) before making the update $\theta_{\text{new}} = \theta_{\text{old}} + \alpha\delta\theta^*$. A line search can also be implemented by sampling from the environment for policy evaluation at different update scales.

### C.9    Manual Choice of $C_1$ and $C_2$

Since the TDPO algorithm operates using the metric Wasserstein distance, thinking about how normalizing actions and rewards affect the corresponding optimization objective builds insight into how to set these coefficients properly. Say we use the same dynamics, only the new actions are scaled up by a factor of $\beta$, and the rewards are scaled up by a factor of $\alpha$:

$$a_{\text{new}} = \beta \cdot a_{\text{old}} \qquad r_{\text{new}} = \alpha \cdot r_{\text{old}}. \tag{81}$$

If the policy function approximation class remained the same, the policy gradient would be scaled by a factor of $\frac{\alpha}{\beta}$ (i.e., $\frac{\partial \eta_{\text{new}}}{\partial a_{\text{new}}} = \frac{\alpha}{\beta} \cdot \frac{\partial \eta_{\text{old}}}{\partial a_{\text{old}}}$). Therefore, one can easily show that the corresponding new regularization coefficient and trust region sizes can be obtained by

$$C_{\text{new}} = \frac{\alpha}{\beta^2} \cdot C_{\text{old}} \tag{82}$$

and

$$\delta_{\max}^{\text{new}} = \beta \cdot \delta_{\max}^{\text{old}}. \tag{83}$$

The following process can provide a starting point for hyper-parameter optimization: (1) Define $C \propto \alpha \cdot \beta^{-2}$, $\delta_{\max} \propto \beta$ and $\sigma_q \propto \beta$ (where $\sigma_q$ is the action disturbance parameter used for DeVine), (2) using prior knowledge or by trial and error determine appropriate action and reward normalization coefficients. The reward normalization coefficient $\alpha$ can approximate the average per-step discounted reward difference between a null policy and an optimal policy.

### C.10    Implementation Details for the Practical Training of the Robotic Leg

Section C.10.1 details the one variable at a time parameter sweep experiments which served as the initial guess set for HPO and defined the hyper-parameter search domain. Section C.10.2 details the individual training settings for each set of proposed hyper-parameters. Section C.10.3 discusses the HPO and the log-space search details. Finally, Section C.10.4 describes the specifications of the transition dynamics, the reward definition, and the imposed stochasticity in the observations and the initial state distribution. We used TDPO with line search and adaptive exploration scale parameters. We excluded TD3 from Figure 4 since it did not produce payoffs above $-10^3$ after a few iterations.

### C.10.1    One Variable at a Time Parameter Sweeps

For better performance, HPO methods need a reasonable set of initial hyper-parameter guesses. For this, we perform a one-variable-at-a-time parameter sweep around the central (default) setting of the reinforcement learning method. The central HP values, the search domain, and the sweep values are shown in Tables 5, 6, and 7 for the PPO, TRPO, and TD3 methods, respectively. Each HP was swept with the candidate values while keeping all the other HPs fixed at their central value. The test problem for determining the best HPO method had a short horizon, so the default HPs of each method were used as the central value. The only exception for this was the entropy coefficient, which was set to zero since the default value caused performance deterioration for TRPO and PPO in this particular test

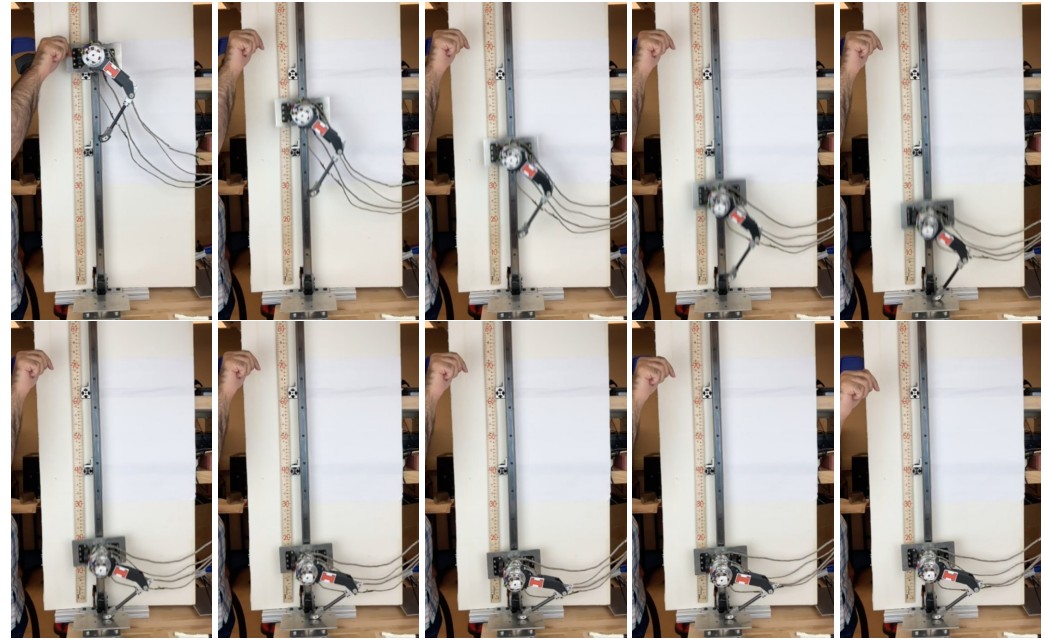

Figure 21: The frame sequence of a physical drop-and-catch test from a height of 0.7 m using the best agent trained by our method (TDPO) at 4 kHz control frequency. A short video of multiple drops from different heights is also included in our code repository.

problem. For TD3, we implemented two common exploration noise types: the Ornstein-Uhlenbeck and the pink noise. Each of these noises was parametrized by their relative bandwidth, where the relative bandwidth lies within $[0, 1]$. This relative bandwidth was treated as a hyper-parameter for TD3. Since the Ornstein-Uhlenbeck noise with a relative bandwidth of 1 is the same as the white Gaussian noise (TD3's default exploration noise), we set it as the default. All settings were repeated with 3 random seeds, and the best agent's return during each individual training was reported as the performance metric for HPO. The parameter sweep values were extended on both ends until a clear peek in performance was detected. These parameter sweeps created an initial set of HPs with their corresponding performances, which were initially input to all HPO methods.

Since there was a 40-fold increase in the number of time-steps per trajectory between the short- and long-horizon environments, relevant HPs were proportionally scaled in the long-horizon. In particular, the central value, domain, and parameter sweep values for all (a) batch-sizes, (b) initial pre-training samples, and (c) training intervals were multiplied by 40. Due to computational resource limitations, we did not scale TD3's optimization batch-size proportional to the horizon. We also adjusted the MDP and GAE discount factors so that their respective horizon lengths are multiplied by 40.

### C.10.2 Individual Training Details

In the short-horizon test problem, we observed that the network architecture used in PPO, TRPO, and TDPO improved TD3's performance over its default architecture (i.e., when using ReLU activations in the hidden layers and a `tanh` output activation followed by normalizing the actions). Therefore, we used the same neural architecture (a 3-layer MLP with 64 units in the hidden layers and `tanh` activation) for all methods. As shown in Figure 3, using this architecture, TD3 managed to outperform the other methods upon full HPO on the short-horizon test benchmark.

**The short-horizon test environment:** We used four parallel workers in all trainings. Therefore, all the relevant batch-sizes in Tables 5, 6, and 7 must be quadrupled to reveal the collective values. Each training was performed for one million time-steps per worker (i.e., four million collective training steps). Since (a) some HPO methods were intolerant of evaluation stochasticity and (b) performing HPO with many seeds was intractable, the HPO's metric was defined as the average performance on 3 pre-determined random seeds. This environment defined trajectories with a duration of 2 seconds and a control frequency of 100 Hz, corresponding to a total of 200 time-steps per trajectory.

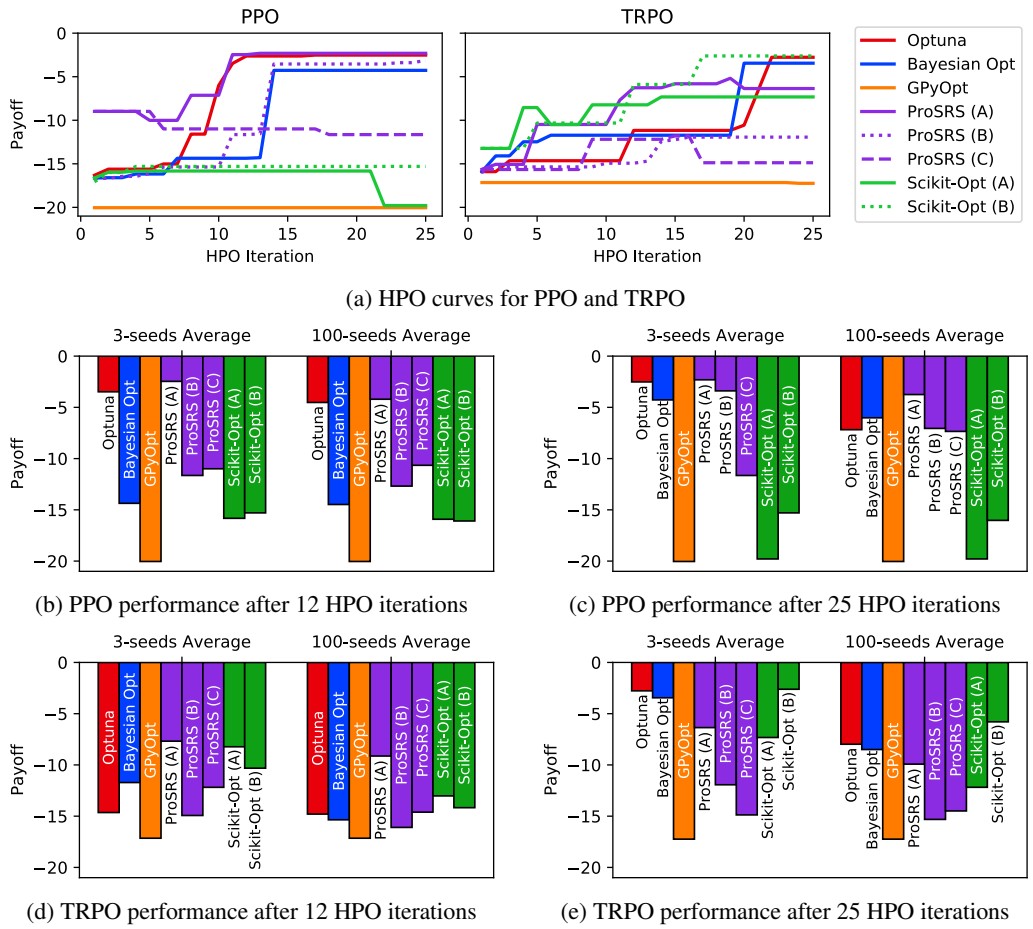

(a) HPO curves for PPO and TRPO

(b) PPO performance after 12 HPO iterations

(c) PPO performance after 25 HPO iterations

(d) TRPO performance after 12 HPO iterations

(e) TRPO performance after 25 HPO iterations

Figure 22: The top line-plots show the HPO payoff curves for PPO and TRPO. For each iteration, the best agent obtained so far was evaluated using the 3 HPO seeds. In the top-left bar plot, the best agent after 12 iterations of HPO was selected and evaluated twice for each method; once only with the 3 HPO seeds, and once with 100 different random seeds. The shorter the bars, the better the agent's performance. Three ProSRS and two Scikit-Opt variants were considered. The A variants made 30 parallel proposals per iteration, where each proposal was evaluated with a single seed randomly chosen from the 3 HPO seeds. The B variant made 10 parallel proposals per iteration, where each proposal was evaluated with all 3 seeds. Variant C made 30 parallel proposals per iteration, where each proposal was evaluated with a single seed randomly chosen between 1 and 100. Similarly, Sub-figures the top-right, bottom-left, and bottom-right bar plots were drawn for PPO after 25 iterations, TRPO after 12 iterations, and TRPO after 25 iterations, respectively. Overall, Variant A offers the best balance between the number of proposals and the induced stochasticity. The bar plots suggest a high ordering correlation between the evaluations on 3 seeds and 100 seeds, which further validates the "optimize the hyper-parameters for 3-seeds and finally evaluate on many seeds" approach.

**The long-horizon leg environment:** We used 144 parallel workers in this setting, which means that the collective batch-sizes can be obtained by multiplying the values in Table 8 by 144. We trained each set of proposed hyper-parameters for 5 billion collective time-steps. Similar to the short-horizon environment, the HPO's metric was defined as the average performance on 3 pre-determined random seeds. However, the final set of best hyper-parameters for each method was trained with 25 random seeds to be shown in Figure 4 of the main paper. This environment defined rollout durations of 2 s and a control frequency of 4 kHz, corresponding to a total of 8000 time-steps per trajectory.

### C.10.3 The HPO Details

We used the default settings with each implementation. Optuna used a Tree Parzen Estimation (TPE) method, Bayesian Optimization used a Gaussian Process (GP) with Upper Confidence Bound (UCB)

| Hyper-Param. | Center | Domain | Parameter Sweep Values |
|---|---|---|---|
| MDP Discount | 0.99 | [0.36, 0.99984375] | 0.36, 0.68, 0.84, 0.92, 0.96, 0.98, 0.99, 0.995, 0.9975, 0.99875, 0.999375, 0.9996875, 0.99984375 |
| GAE Discount | 0.98 | [0.36, 0.99984375] | 0.36, 0.68, 0.84, 0.92, 0.96, 0.98, 0.99, 0.995, 0.9975, 0.99875, 0.999375, 0.9996875, 0.99984375 |
| Sampling BS | 256 | [64, 16384] | 64, 128, 256, 512, 1024, 2048, 4096, 8192, 16384 |
| Clip Param. | 0.2 | [0.02, 200] | 0.02, 0.06, 0.2, 0.6, 2.0, 6, 20, 60, 200 |
| Entropy Coef. | 0 | [0, 0.1] | 0, 0.001, 0.01, 0.1 |
| Opt. Epochs | 4 | [1, 128] | 1, 2, 4, 8, 16, 32, 64, 128 |
| Opt. MBs | 4 | [1, 64] | 1, 2, 4, 8, 16, 32, 64 |
| Opt. LR | $10^{-3}$ | $[10^{-5}, 10^{-1}]$ | $10^{-5}, 3 \times 10^{-5}, 10^{-4}, 3 \times 10^{-4}, 10^{-3},$ $3 \times 10^{-3}, 10^{-2}, 3 \times 10^{-2}, 10^{-1}$ |
| ADAM $\epsilon$ | $10^{-5}$ | $[10^{-8}, 10^{-4}]$ | $10^{-8}, 3 \times 10^{-8}, 10^{-7}, 3 \times 10^{-7}, 10^{-6},$ $3 \times 10^{-6}, 10^{-5}, 3 \times 10^{-5}, 10^{-4}$ |
| LR Schedule | Linear | - | Constant, Linear |

Table 5: The one-variable-at-a-time parameter sweep details for the PPO method on the short-horizon leg benchmark environment. BS, MB, and LR are short for batch-size, mini-batch, and learning rate, respectively. See Section C.10.1 for more information.

| Hyper-Param. | Center | Domain | Parameter Sweep Values |
|---|---|---|---|
| Sampling BS | 1024 | [64, 16384] | 64, 128, 256, 512, 1024, 2048, 4096, 8192, 16384 |
| MDP Discount | 0.99 | [0.36, 0.99984375] | 0.36, 0.68, 0.84, 0.92, 0.96, 0.98, 0.99, 0.995, 0.9975, 0.99875, 0.999375, 0.9996875, 0.99984375 |
| GAE Discount | 0.98 | [0.36, 0.99984375] | 0.36, 0.68, 0.84, 0.92, 0.96, 0.98, 0.99, 0.995, 0.9975, 0.99875, 0.999375, 0.9996875, 0.99984375 |
| Max KL | 0.01 | [0.00125, 0.64] | 0.00125, 0.0025, 0.005, 0.01, 0.02, 0.04, 0.08, 0.16, 0.32, 0.64 |
| CG Iterations | 10 | [1, 20] | 1, 2, 5, 10, 20 |
| Entropy Coef. | 0 | $[0.0, 10^{-3}]$ | $0.0, 10^{-5}, 10^{-4}, 10^{-3}$ |
| CG Damping | 0.01 | $[10^{-4}, 1]$ | $10^{-4}, 10^{-3}, 10^{-2}, 10^{-1}, 1$ |
| VF LR | 0.0003 | $[3 \times 10^{-6}, 3 \times 10^{-2}]$ | $3 \times 10^{-6}, 10^{-5}, 3 \times 10^{-5}, 10^{-4}, 3 \times 10^{-4},$ $10^{-3}, 3 \times 10^{-3}, 10^{-2}, 3 \times 10^{-2},$ |
| VF Iterations | 3 | [1, 24] | 1, 3, 6, 12, 24 |
| VF MBs | 8 | [1, 64] | 1, 2, 4, 8, 16, 32, 64 |

Table 6: The one variable at a time parameter sweep details for TRPO on the short-horizon leg benchmark environment. BS, MB, LR, VF, and CG are short for batch-size, mini-batch, learning-rate, value function, and conjugate gradient, respectively. See Section C.10.1 for more information.

acquisitions and the Mattern kernel, Scikit-Opt used a GP with a hedge acquisition function (i.e., automatically determining the acquisition function from a pre-defined set), ProSRS used a GP with Radial Basis Function (RBF) acquisitions, and GPyOpt used a GP with Local Penalization (LP), UCB acquisitions, and a white noise kernel. All HPO parameters were left as their default values.

ProSRS and Scikit-Opt HPO implementations could tolerate evaluation noise. That is, these implementations could tolerate stochasticity in their evaluation metric according to their documentation. On the other hand, Optuna, Bayesian Optimization, and GPyOpt's documentation suggested running them on non-noisy evaluation functions only. Since each HPO training in the long-horizon environment could take 10 hours, running thousands of sequential HPO iterations is impractical. Therefore, we used the HPO implementations in a "batched" capacity, where each HPO method proposed multiple sets of HPs for parallel evaluation and then received their performance values simultaneously. We allowed all HPO methods to ask for 30 parallel trainings in each proposal.

| Hyper-Param. | Center | Domain | Parameter Sweep Values |
|---|---|---|---|
| MDP Discount | 0.99 | $[0.36, 0.99984375]$ | $0.36, 0.68, 0.84, 0.92, 0.96, 0.98, 0.99, 0.995, 0.9975,$ $0.99875, 0.999375, 0.9996875, 0.99984375$ |
| Buffer Size | 50000 | $[3125, 800000]$ | $3125, 6250, 12500, 25000, 50000, 100000,$ $200000, 400000, 800000$ |
| Pre-training | 100 | $[25, 6400]$ | $25, 50, 100, 200, 400, 800, 1600, 3200, 6400$ |
| Train Interval | 100 | $[25, 6400]$ | $25, 50, 100, 200, 400, 800, 1600, 3200, 6400$ |
| Opt. BS | 128 | $[8, 2048]$ | $8, 16, 32, 64, 128, 256, 512, 1024, 2048$ |
| Opt. LR | 0.0003 | $[3 \times 10^{-6}, 3 \times 10^{-2}]$ | $3 \times 10^{-6}, 10^{-5}, 3 \times 10^{-5}, 10^{-4},$ $3 \times 10^{-4}, 10^{-4}, 3 \times 10^{-4}, 10^{-2}, 3 \times 10^{-2}$ |
| GD Iterations | 100 | $[6, 400]$ | $6, 12, 24, 50, 100, 200, 400$ |
| Soft Update Coefficient | 0.005 | $[0.000625, 0.08]$ | $0.000625, 0.00125, 0.0025, 0.005,$ $0.01, 0.02, 0.04, 0.08$ |
| Policy Delay | 2 | $[1, 16]$ | $1, 2, 4, 8, 16$ |
| Noise Type | Ornstein | - | Ornstein, Pink |
| Noise RFB | 1 | $[2^{-8}, 1]$ | $2^{-8}, 2^{-7}, 2^{-6}, 2^{-5}, 2^{-4}, 2^{-3}, 2^{-2}, 2^{-1}, 1$ |
| Noise std | 0.1 | $[0.00625, 0.8]$ | $0.00625, 0.0125, 0.025, 0.05, 0.1, 0.2, 0.4, 0.8$ |
| Target Noise std | 0.2 | $[0.00625, 3.2]$ | $0.00625, 0.0125, 0.025, 0.05, 0.1,$ $0.2, 0.4, 0.8, 1.6, 3.2$ |
| Target Noise Clipping | 0.5 | $[0.0625, 4]$ | $0.0625, 0.125, 0.25, 0.5, 1, 2, 4$ |

Table 7: The one-variable-at-a-time parameter sweep details for the TD3 method on the short-horizon robotic leg test environment. BS, LR, GD, and RFB are short for batch-size, learning rate, gradient descent, and relative frequency bandwidth, respectively. See Section C.10.1 for more information.

The noise-tolerant HPO implementations (ProSRS and Scikit-Opt) were allowed to propose 30 different HPs. We ran each of these 30 with one of the 3 pre-determined random seeds (picked at random) and returned the result to the HPO method. Figure 22 shows that this was the best configuration for ProSRS and Scikit-Opt out of those tried. On the other hand, the noise-intolerant HPO implementations (Optuna, Bayesian Optimization, and GPyOpt) could only propose 10 HP sets, since each set needed to be trained on all 3 pre-determined random seeds. This "batched" HPO approach allowed us to effectively optimize TRPO, PPO, and TD3 on the short horizon benchmark in 25 HPO iterations as shown in Figure 4. Due to resource limitations, we only ran 11 HPO iterations for PPO and TRPO on the long-horizon environment.

**Hyper-parameter pre-processing transformations:** We applied a log transformation to all numeric hyper-parameters before passing them to HPO methods. We made two exceptions to this rule. Instead of searching for the MDP and GAE discount factors in the log-space, we transformed them into their respective horizons and then searched for the horizons logarithmically. That is, instead of searching for $\log(\gamma)$, we searched for $\log(1/(1-\gamma))$. Since we wanted the HPO method to be able to set the entropy coefficient $C_{\text{entropy}}$ to zero, we searched for $\log(C_{\text{entropy}} + 10^{-5})$ instead of searching for $\log(C_{\text{entropy}})$. The search domains for all HPs were identical to the one variable at a time parameter sweep domains in Section C.10.1, where the domain bounds were extended until a clear peek in performance was detected.

### C.10.4 The Environment Specification

For the dynamics, we used the same physical model as the one in Sections 4.2 and C.2. For the initial state distribution, the hip and knee angles were uniformly chosen from the $[-180°, -30°]$ and $[-155°, -35°]$ intervals, respectively. The initial drop height of the robotic leg was uniformly chosen between 0.4 and 0.8 meters. To simulate the unmodeled physical characteristics, we extracted physical sensing noises from the hardware and used them as a template for generating stochastic observation noise in our simulated model. For a better simulation-to-real transfer, we modified the

| Method | Hyper-Param. | Center | Domain | Parameter Sweep Values |
|---|---|---|---|---|
| PPO, TRPO, TD3 | MDP Discount | 0.99975 | [0.488, 0.99996875] | 0.488, 0.744, 0.872, 0.936, 0.968, 0.984, 0.992, 0.996, 0.998, 0.999, 0.9995, 0.99975, 0.999875, 0.9999375, 0.99996875 |
| PPO, TRPO | GAE Discount | 0.9995 | [0.488, 0.9999375] | 0.488, 0.744, 0.872, 0.936, 0.968, 0.984, 0.992, 0.996, 0.998, 0.999, 0.9995, 0.9995, 0.99975, 0.999875, 0.9999375 |
| PPO, TRPO | Sampling BS | 40000 | [19, 160000] | 19, 39, 78, 156, 312, 625, 1250, 2500, 5000, 10000, 20000, 40000, 80000, 160000 |
| TD3 | Buffer Size | 2000000 | [62500, 8000000] | 62500, 125000, 250000, 500000, 1000000, 2000000, 4000000, 8000000 |
| TD3 | Pre-training, Train Interval | 4000 | [500, 64000] | 500, 1000, 2000, 4000, 8000, 16000, 32000, 64000 |
| TD3 | Opt. BS | 128 | [8, 2048] | 8, 16, 32, 64, 128, 256, 512, 1024, 2048 |

Table 8: The one-variable-at-a-time parameter sweep details for the long-horizon robotic leg environment. Only the HPs in need of horizon-scaling were given here, and the rest of the HPs used the same settings as Tables 5, 6, and 7. 144 parallel workers were used in the long-horizon environment trainings, so the collective batch-sizes are 144 times the values in this table.

reward definition to emphasize the penalty for non-optimal behavior such as violating the physical constraints. In particular, we used the reward function described by the following equations:

$$R = R_{\text{torque sm.}} + R_{\text{foot offset}} + R_{\text{posture}} + R_{\text{velocity}} + R_{\text{torque}} + R_{\text{constraints}} \tag{84}$$

with

$$R_{\text{torque sm.}} = -2 \times \left[ (\tau_{\text{knee}} - \tau_{\text{knee}}^{\text{old}})^2 + (\tau_{\text{hip}} - \tau_{\text{hip}}^{\text{old}})^2 \right]$$
$$R_{\text{foot offset}} = -1 \times x_{\text{foot}}^2$$
$$R_{\text{posture}} = -0.1 \times \left[ (z_{\text{hip}} - z_{\text{foot}}) - z_{\text{posture}}^{\text{target}} \right]^2$$
$$R_{\text{torque}} = -10^{-7} \times \left[ \tau_{\text{knee}}^2 + \tau_{\text{hip}}^2 \right]$$
$$R_{\text{velocity}} = -10^{-4} \times \left[ \omega_{\text{knee}}^2 + \omega_{\text{hip}}^2 \right]$$
$$R_{\text{constraints}} = -0.1 \times \mathbf{1}_{\text{phys. violation}} \tag{85}$$

where

- $\omega_{\text{knee}}$ and $\omega_{\text{hip}}$ are the knee and hip angular velocities in radians per second, respectively.

- $\tau_{\text{knee}}$ and $\tau_{\text{hip}}$ are the knee and hip torques in Newton meters, respectively.

- $x_{\text{foot}}$ and $z_{\text{foot}}$ are the horizontal and vertical foot offsets in meters from the desired standing point on the ground, respectively.

- $z_{\text{hip}}$ is the vertical hip offset in meters from the desired standing point on the ground.

- $z_{\text{posture}}^{\text{target}}$ is a target posture height of 0.1 m.

- $\tau_{\text{knee}}^{\text{old}}$ and $\tau_{\text{hip}}^{\text{old}}$ are the values of $\omega_{\text{knee}}$ and $\omega_{\text{hip}}$ from the previous time-step, respectively.

- $\mathbf{1}_{\text{phys. violation}}$ is an indicator variable only being one when the agent violates the physical safety bounds of the robotic leg hardware. This involves exceeding the limits of allowed hip or knee angles and angular velocities, or the vertical offsets of the hip and knee. Such violations could result in physical damage to the hardware and were penalized during the training.

# D   Broader Impact

This work provides foundational theoretical results and builds upon reinforcement learning techniques within the area of machine learning. Reinforcement learning methods can provide stable control and decision-making processes for a range of challenging applications in robotics [23], computer vision [60], advertisement and recommendation systems [49], human search and rescue in natural disasters [41, 12], automated resource management [37], chemistry [61], computational biology [46], and even clinical surgeries [40].

Although many implications could result from the application of reinforcement learning, in this work we focused especially on settings where intelligence, precision, and speed are required for controlling robotic movements. Our work particularly investigated methods for controlling oscillation characteristics using reinforcement learning agents. Such improvements could help stabilize high-bandwidth robotic environments and may facilitate the training of intelligent agents for robotic hardware where safety is of concern [3]. The negative consequences of this work could include the removal of human decision-making from the controller design loop, the unknown existence of unforeseen loopholes in the engineered behavior, and vulnerability to policy induction attacks [6].

To mitigate the risks, we encourage further research to develop methods to provide guarantees and definitive answers about agent behavior. In other words, a general framework for making guaranteed statements about the behavior of the trained agents is missing. For instance, one cannot currently guarantee that the agent would act safely under all circumstances and perturbations even if it achieves high performances in practice. Understanding exploitation techniques of such reinforcement learning agents and designing processes to prevent such abuses could be of paramount societal concern.