# OpenReview forum: "Truly Deterministic Policy Optimization"
_NeurIPS.cc/2022/Conference — NeurIPS 2022 Accept_

### Official Review · Reviewer_3Zyz · 2022-07-07

**Rating:** 8
**Confidence:** 3
**Soundness:** 4 excellent
**Presentation:** 4 excellent
**Contribution:** 4 excellent

**Summary:**

This work proposes a gradient-based algorithm for updating a deterministic policy’s parameters in a reinforcement learning setting. While RL algorithms for training deterministic policies exist, they require the use of a stochastic exploration policy that can introduce variance into policy updates. This variance is especially problematic in environments with non-local rewards and long horizons. The TDPO algorithm uses a deterministic version of the Vine estimator to compute exact advantage estimates, and regularizes the policy updates using the Wasserstein metric, which is uniquely suited for deterministic policies. The algorithm is benchmarked against state-of-the-art baselines on robotic domains with non-local rewards and long horizons, and clearly shows a benefit in achieved performance.

**Questions:**

My primary question is: in what environments do you expect the algorithm to perform poorly? My main concern is that it would have inadequate exploration. Though $\epsilon$-greedy is a notoriously inefficient strategy, would this algorithm fair better than, say, Thompson sampling or UCB-exploration, if those algorithms could be tractably applied to the same problem setting?

**Limitations:**

The discussion of limitations in this work is restricted its scope, i.e. that the algorithm is designed for continuous state and action spaces only. These is no discussion of negative societal impact, though I think such a discussion is unnecessary for this work. Regarding scope, and as stated in the Strengths & Weaknesses section, I would add a discussion on how the algorithm would fare in settings where exploration is required, and when to avoid using it in favor of another algorithm.

**Strengths And Weaknesses:**

Strengths:
This paper is clearly written, and derives its approach with appropriate rigor. Moreover, it identifies an important problem, proposes a sensible solution, and thoroughly demonstrates the benefits of the work through experiments.

Weaknesses:
Though this is a very strong paper, I think there is inadequate discussion of its limitations. Namely, it seems to be the cases that any exploration can only result from perturbing the initial condition. It seems that some exploration is necessary in many domains, and it would be worth spelling out how TDPO would fare in such domains.

There are also a few typos, such as “Since Optuna is widely-test[ed] and…“.

---

> ### Author Response · Authors · 2022-08-02
> **Response to Reviewer 3Zyz**
>
> First, we want to thank the reviewer for their constructive comments. These helped us to improve the presentation in our paper.
>
> ## Questions
> 1. *"in what environments do you expect the algorithm to perform poorly? My main concern is that it would have inadequate exploration. Though ϵ-greedy is a notoriously inefficient strategy, would this algorithm fair better than, say, Thompson sampling or UCB-exploration, if those algorithms could be tractably applied to the same problem setting?"*
>
>     **Answer**: We think that TDPO may well have inadequate exploration and we added the following text to Section 5, Line 297:
>
>     "*Second, TDPO relies on local gradients and thus may not be able to learn effectively in environments with payoffs that are sparse in either action or state. Such environments typically need substantial exploration which TDPO may not be able to achieve.*"
>
> 2. *"The discussion of limitations in this work is restricted its scope, i.e. that the algorithm is designed for continuous state and action spaces only... Regarding scope, and as stated in the Strengths & Weaknesses section, I would add a discussion on how the algorithm would fare in settings where exploration is required, and when to avoid using it in favor of another algorithm."*
>
>     **Answer**: We thank the reviewer for their suggestion. We have expanded the discussion of our method's limitations in Section 5, Line 295:
>
>     "*There are a number of limitations of this paper. First, we assumed continuous environments and required a state reset capability of the environment, which is commonly available in simulators but would prevent learning on hardware. Second, TDPO relies on local gradients and thus may not be able to learn effectively in environments with payoffs that are sparse in either action or state. Such environments typically need substantial exploration which TDPO may not be able to achieve. Third, the analysis in this paper does not specifically pinpoint the reasons why existing methods fail on the long-horizon non-local-reward robotic control environments. For instance, such failures might be more specifically attributed to either (1) the frequency-based nature of the reward, (2) the non-locality of the reward signal, or (3) the long horizon. Understanding this is an important prerequisite for future work.*"

---

> > ### Comment · Reviewer_3Zyz · 2022-08-10
> > **Thank you for your response**
> >
> > Thank you for your response. I maintain that this is a strong paper and so will leave my score unchanged.

---

### Official Review · Reviewer_c8Yt · 2022-07-08

**Rating:** 6
**Confidence:** 3
**Soundness:** 3 good
**Presentation:** 2 fair
**Contribution:** 3 good

**Summary:**

A solid paper, which I think represents a valuable contribution to the NeurIPS community because it tackles an important problem (improving actor-critic methods) in a novel way. The major weakness is in presentation, especially of issues around exploration. With deterministic policies, we need to be very careful to explore enough, but the paper is remarkably silent about what policy is used for exploration.

The literature review section is missing OAC [1], whose deterministic version explores using deterministic policies, solving an optimisation problem that involves the Wasserstein divergence.

Typos:
- line 152: well-suited form => in a form well-suited
- line 287: widely-test => widely tested

References:
[1] Ciosek, K., Vuong, Q., Loftin, R., & Hofmann, K. (2019). Better exploration with optimistic actor critic. Advances in Neural Information Processing Systems

**Questions:**

1. Please provide more details about how the exploration policy is chosen. Moreover, can you clarify the pseudocode of Algorithm 1 to make it clear where the exploration policy comes form?
2. Can you explain what you mean by constraints in line 160? The optimisation objective in equation (7) does not seem to come with any constraints.
3. In line 116, you take the 2-norm of a Lipschitz constant. What does that mean given the Lipschitz constant is defined as a scalar?
4. In practice, can you explain how injecting noise into the transition dynamics you describe in the beginning of section 4.3 is different from injecting noise into the action distribution (which is what vanilla TD3 and DDPG do).
5. In the top part of Figure 1, can you explain why the different algorithms have vastly different performance at zero samples.

**Limitations:**

The authors do not explicitly discuss the limitations of the work, except for the last sentence in the Discussion section. In my view, the major limitations are:
1. Empirically, while the evaluation is statistically excellent (the authors use 100 random seeds), the authors only test on relatively simple environments. Moreover, other than pendulum, the used environments are non-standard and seem to have been introduced with the proposed algorithm in mind. This presents a risk that the good performance of the proposed method is due to overfitting a cherry-picked benchmark. However, the authors do alleviate these concerns to a certain extent by testing on a real robot.
2. Theoretically, the authors do not show, even in the bandit case, why their method explores better than alternative approaches. You do not have to provide a regret bound (although that would be ideal), but providing some intuitions about how the proposed method is able to explore would be nice.

**Strengths And Weaknesses:**

Strengths:
1. The paper addresses an important issue of improving policy-gradient-based RL agents. This has huge practical significance since most RL agents used in practice are based on policy gradients.
2. The empirical evaluation is limited in scope, but very solid, with 100 independent runs.
3. As far as I can tell, the proposed update to the policy is novel.
4. There is an experiment on a real robot.

Weaknesses:
1. The presentation could be improved, particularly concerning:
    - how the algorithm explores
    - the pseudocode needs to be made more specific (make it clear that algorithm 2 is invoked in line 2 of algorithm 1).
2.  The simulated benchmarks seem a bit artificial.

---

> ### Author Response · Authors · 2022-08-02
> **Response to Reviewer c8Yt - Part 2**
>
> 7. *"In practice, can you explain how injecting noise into the transition dynamics you describe in the beginning of section 4.3 is different from injecting noise into the action distribution (which is what vanilla TD3 and DDPG do)."*
>
>     **Answer**: We modified the paper (Line 256) to clarify that this is "physical modeling" noise, which is designed to model the unmodeled physical components in the mechanical and electrical sub-systems. In other words, this noise is inherent to the physical model (i.e., sensed observations and exerted actuations), is neither Gaussian nor Ornestien, and is not associated with the algorithmic RL explorations.
>
> 8. *"In the top part of Figure 1, can you explain why the different algorithms have vastly different performance at zero samples."*
>
>     **Answer**: The agents were not check-pointed for evaluation at zero samples until after one epoch. Since this is difficult to see in the figure, we clarified this point in the caption by adding the following sentence:
>
>     "*The initial agent payoffs indicate the performance after the first epoch.*"
>
> 9. *"Empirically, while the evaluation is statistically excellent (the authors use 100 random seeds), the authors only test on relatively simple environments. Moreover, other than pendulum, the used environments are non-standard and seem to have been introduced with the proposed algorithm in mind. This presents a risk that the good performance of the proposed method is due to overfitting a cherry-picked benchmark. However, the authors do alleviate these concerns to a certain extent by testing on a real robot."*
>
>     **Answer**: We like to draw the reviewer's attention to Figure 9 on Page 30 of the supplementary material, where we show that our method has similar performance to the existing PG methods on the traditional gym control tasks. We had referred to these results in the main paper in Line 200:
>
>     "*See the Supplementary Material for a comparison on traditional gym environments, where the basic variant of TDPO works similarly to existing methods.*"
>
>     We also have added a paragraph to the Section 5, Line 295, discussing our work's limitations:
>
>     "*There are a number of limitations of this paper. First, we assumed continuous environments and required a state reset capability of the environment, which is commonly available in simulators but would prevent learning on hardware. Second, TDPO relies on local gradients and thus may not be able to learn effectively in environments with payoffs that are sparse in either action or state. Such environments typically need substantial exploration which TDPO may not be able to achieve.*"
>
>     Our work with real robots suggested that the traditional gym benchmarks do not capture the challenges in realistic robotic systems. This is why we focused our attention on environments with long-horizons and non-local rewards. The inability of the existing methods to tackle these challenges motivated the design of our method.
>
> 10. *"Theoretically, the authors do not show, even in the bandit case, why their method explores better than alternative approaches. You do not have to provide a regret bound (although that would be ideal), but providing some intuitions about how the proposed method is able to explore would be nice."*
>
>     **Answer**: We would love to have more theoretical results about exploration in these environments. We added the following to Line 299:
>
>     "*Third, the analysis in this paper does not specifically pinpoint the reasons why existing methods fail on the long-horizon non-local-reward robotic control environments. For instance, such failures might be more specifically attributed to either (1) the frequency-based nature of the reward, (2) the non-locality of the reward signal, or (3) the long horizon. Understanding this is an important prerequisite for future work.*"
>
>     Also, see Line 187:
>
>     *"The DeVine estimator can be advantageous in at least two scenarios. First, in the case of rewards that cannot be decomposed into summations of immediate rewards. For example, overshoot penalizations or frequency-based rewards as used in robotic systems are non-local. DeVine can be robust to non-local rewards as it is insensitive to whether the rewards were applied immediately or after a long period. Second, DeVine can be an appropriate choice for systems that are sensitive to the injection of noise, such as high-bandwidth robots with natural resonant frequencies. In such cases, using white (or colored) noise for exploration can excite these resonant frequencies and cause instability, making learning difficult. DeVine avoids the need for constant noise injection."*

---

> > ### Comment · Reviewer_c8Yt · 2022-08-09
> > **Thank you for the response.**
> >
> > Thanks for the very detailed response - I really appreciate the clarifications, particularly concerning exploration.
> > Because of the identified limitations, I decided to stick to the current score.

---

> ### Author Response · Authors · 2022-08-02
> **Response to Reviewer c8Yt - Part 1**
>
> First, we want to thank the reviewer for their constructive comments, which helped us to improve our exposition.
>
> ## Questions
> 1. "With deterministic policies, we need to be very careful to explore enough, but the paper is remarkably silent about what policy is used for exploration."
>
>     **Answer**: Thanks for pointing out that this was not adequately explained. We made changes to Algorithm 2. We also added the following paragraph to the discussion section, Line 295:
>
>     "*There are a number of limitations of this paper. First, we assumed continuous environments and required a state reset capability of the environment, which is commonly available in simulators but would prevent learning on hardware. Second, TDPO relies on local gradients and thus may not be able to learn effectively in environments with payoffs that are sparse in either action or state. Such environments typically need substantial exploration which TDPO may not be able to achieve. Third, the analysis in this paper does not specifically pinpoint the reasons why existing methods fail on the long-horizon non-local-reward robotic control environments. For instance, such failures might be more specifically attributed to either (1) the frequency-based nature of the reward, (2) the non-locality of the reward signal, or (3) the long horizon. Understanding this is an important prerequisite for future work.*"
>
> 2. *"The literature review section is missing OAC, whose deterministic version explores using deterministic policies, solving an optimisation problem that involves the Wasserstein divergence."*
>
>     **Answer**: Thanks for pointing out this interesting additional citation. We added the following sentence to Line 53:
>
>     "*Ciosek et al. [8] introduced a method which in the deterministic mode optimized policies using the Wasserstein distance*"
>
> 3. *"Typos: line 152: well-suited form => in a form well-suited, line 287: widely-test => widely tested"*
>
>     **Answer**: We thank the reviewer for noticing these typos and we have corrected them.
>
> 4. *"Please provide more details about how the exploration policy is chosen. Moreover, can you clarify the pseudocode of Algorithm 1 to make it clear where the exploration policy comes form?"*
>
>     **Answer**: We thank the reviewer for pointing out that our description was unclear. We clarified this point in Algorithm 1 by explicitly cross-referencing Algorithm 2 next to $\mathbb{A}^{\pi_k}$. We also re-arranged Algorithm 2, introduced new notation $\nu_\text{det}$, defined the exploratory action $a_{t_k}':=\pi(s_{t_k})+\sigma\cdot \mathbf{e}_{j_k}$ in Line 5 of Algorithm 2, and added the following text to Line 174:
>
>     *"Algorithm 2 uses an exploration index sampler $\nu$, which samples a set of time-steps and action dimensions for exploration perturbation. The truly deterministic version of TDPO uses the deterministic $\nu_{\text{det}}$ which always returns the complete covering of \{$1,\cdots,\dim(\mathcal{A})$} $\times$ {$1,\cdots,H$}. Using $\nu_{\text{det}}$, DeVine produces exact policy gradients in the limit of small $\sigma$ as stated in Theorem 3.1"*
>
> 5. *"Can you explain what you mean by constraints in line 160? The optimisation objective in equation (7) does not seem to come with any constraints."*
>
>     **Answer**: We thank the reviewer for pointing out that this wasn't clear. We added the following text to Line 156:
>
>     "*However, due to the large number of “sup” terms that need to be expensively statistically estimated and implemented as constraints [48].*"
>
> 6. *"In line 116, you take the 2-norm of a Lipschitz constant. What does that mean given the Lipschitz constant is defined as a scalar?"*
>
>     **Answer**: We thank the reviewer for pointing this out. This was a typo, and we removed the 2-norm.

---

### Official Review · Reviewer_KtyC · 2022-07-10

**Rating:** 5
**Confidence:** 5
**Soundness:** 3 good
**Presentation:** 2 fair
**Contribution:** 3 good

**Summary:**

The authors present a (mostly) deterministic policy gradient method. They derive a theoretical algorithm showing it recovers the same policy gradient estimator.

This is then evaluated in practice showing that it performs similarly to other on-policy methods when tackling standard mujoco tasks. In addition, they show impressive performance gains when tackling long-horizon non-markovian rewards that rely on frequency domain matching.

**Questions:**

- DDPG and TD3 are shown to fail. These tasks are by design partially observable or history dependent. DDPG (and variants) are by design off-policy history agnostic methods. Are PPO/TRPO/TDPO utilizing memory modules or are they only observing the current state?

- What happens when the problem is shorter term with more frequent rewards? For instance, the frequency can be estimated using STFT (for instance) and combining multiple per-step rewards. This might not be the exact same objective, but it's a denser task that is interesting to investigate.

- How do the various methods compare on sparse, non-markovian, rewards that are not necessarily frequency based?

Overall, while the experiments are impressive, I feel they do not cover the full analysis expected of this method. If the point is fully deterministic, the question is where does this help the agent perform better? How does this improve performance. Is it only in sparse reward tasks? Does it demand non-markovian nature or not? Why do previous on-policy methods fail in non-markovian returns (TRPO/PPO)?

**Limitations:**

Yes.

**Strengths And Weaknesses:**

Strengths:

- They present a (mostly) deterministic method that is capable of solving long-horizon problems in the frequency domain. A task that prior works are shown to be incapable of tackling.
- Results in the frequency domain are impressive.

Weaknesses:

- The premise of the paper is a truly deterministic policy gradient method, yet Algorithm 2 is not truly deterministic. Even though q is sampled from less often than off-policy exploration schemes, this still results in additional noise injection.
- Overall the work is presented as if the major benefit is true determinism whereas I feel the major benefit here is in the ability to tackle long horizon sparse reward tasks.

Minor points:

- I argue the authors to go over the paper with a spelling/grammar correction tool (such as grammarly). There are many mistakes such as "appeare" which would be trivially solved with minimal effort.

---

> ### Author Response · Authors · 2022-08-02
> **Response to Reviewer KtyC - Part 2**
>
> 4. *"The premise of the paper is a truly deterministic policy gradient method, yet Algorithm 2 is not truly deterministic. Even though q is sampled from less often than off-policy exploration schemes, this still results in additional noise injection."*
>
>     **Answer**: Thank you for pointing out that our description was not sufficiently clear. We re-arranged Algorithm 2, introduced new notation $\nu_\text{det}$ to clarity that it can be truly deterministic, and added the following text to Line 174:
>
>     *"Algorithm 2 uses an exploration index sampler $\nu$, which samples a set of time-steps and action dimensions for exploration perturbation. The truly deterministic version of TDPO uses the deterministic $\nu_{\text{det}}$ which always returns the complete covering of $\{1,\cdots,\dim(\mathcal{A})\} \times \{1,\cdots,H\}$. Using $\nu_{\text{det}}$, DeVine produces exact policy gradients in the limit of small $\sigma$ as stated in Theorem 3.1"*
>
>
> 5. *"Overall the work is presented as if the major benefit is true determinism whereas I feel the major benefit here is in the ability to tackle long horizon sparse reward tasks."*
>
>     **Answer**: We agree with the reviewer on this point. To emphasize this point, we added the following to the first sentence of our abstract:
>
>     "*...with the goal of improving learning with long horizons and non-local rewards*".
>
>     Also, see Line 301:
>
>     "*For instance, such failures might be more specifically attributed to either (1) the frequency-based nature of the reward, (2) the non-locality of the reward signal, or (3) the long horizon. Understanding this is an important prerequisite for future work.*"
>
>
> 6. *"I argue the authors to go over the paper with a spelling/grammar correction tool (such as grammarly). There are many mistakes such as "appeare" which would be trivially solved with minimal effort."*
>
>     **Answer**: We thank the reviewer for pointing this typo out. We have fixed this instance, and followed the reviewer's suggestion.

---

> ### Author Response · Authors · 2022-08-02
> **# Response to Reviewer KtyC - Part 1**
>
> First, we want to thank the reviewer for their constructive comments. These made us think and helped us improve the paper.
>
> ## Questions
> 1. *"DDPG and TD3 are shown to fail. These tasks are by design partially observable or history dependent. DDPG (and variants) are by design off-policy history agnostic methods. Are PPO/TRPO/TDPO utilizing memory modules or are they only observing the current state?"*
>
>     **Answer**: We added the following text to the paper (Line 212) to clarify this point:
>
>     *"All methods here only used the current state of the systems, despite the fact that these environments are in all cases partially observable or history-dependent. Nevertheless, the agents are able to achieve high-reward behaviors, since these environments have weak history dependence."*.
>
> 2. *"What happens when the problem is shorter term with more frequent rewards? For instance, the frequency can be estimated using STFT (for instance) and combining multiple per-step rewards. This might not be the exact same objective, but it's a denser task that is interesting to investigate."*
>
>     **Answer**: We added the following text to the paper (Line 215) to clarify our point:
>
>     "*Note that this environment is a toy-problem to illustrate frequency dependence, and may as such be solved using Wavelet or short-term Fourier transformations in conjunction with the existing PG methods. However, our focus is on the representative features of this example, rather than this particular problem itself.*"
>
>
> 3. *"How do the various methods compare on sparse, non-markovian, rewards that are not necessarily frequency based? Overall, while the experiments are impressive, I feel they do not cover the full analysis expected of this method. If the point is fully deterministic, the question is where does this help the agent perform better? How does this improve performance. Is it only in sparse reward tasks? Does it demand non-markovian nature or not? Why do previous on-policy methods fail in non-markovian returns (TRPO/PPO)?"*
>
>     **Answer**: These are very interesting questions, which we would like to return in future work. In our work so far, we have been motivated by problems derived from our work with real-world experimental robotic systems, which inspired the design of our TDPO method. We added the following to our discussion section, Line 299:
>
>     "*Third, the analysis in this paper does not specifically pinpoint the reasons why existing methods fail on the long-horizon non-local-reward robotic control environments. For instance, such failures might be more specifically attributed to either (1) the frequency-based nature of the reward, (2) the non-locality of the reward signal, or (3) the long horizon. Understanding this is an important prerequisite for future work.*"
>
>     Also, see Line 75:
>
>     "*All in all, long horizons are challenging for all reinforcement learning methods, especially the ones suffering from excessive estimation variance due to the use of stochastic policies for exploration, and our truly deterministic method may have advantages in this respect.*"
>
>     To compare various methods on non-local rewards that are not necessarily defined in the frequency domain, we added an experiment in Section C.6 (Page 31) and Figure 20 (Page 37) of our supplementary material:
>
>     *"Stochastic methods fail at solving the frequency-domain problem of Figure1 in the main paper. They may be doing so because of the frequency-based nature of the reward. The particular desired frequency range may also be playing a significant role.  Alternatively, the non-locality of the reward may be the main reason behind their failure. To investigate this, we simulate a typical MDP reward on the same pendulum model, and make it artificially non-local by accumulating the rewards every 2, 20, or 200 time-steps. Figure 20 shows the training curves for each case. Other methods perform better than ours when the rewards are only accumulated for 2 time-steps. However, our method is much more resilient to higher non-locality of the reward. Since this reward is not defined in the frequency domain, yet the other methods significantly deteriorate in performance with higher non-locality levels, we speculate that the non-locality of the reward is a more influential factor in the failure of the existing methods than merely the frequency-based nature of the reward used in Figure 1. Between the main paper's variant, and the second and the third variants shown in Figures 11 and 12, a 5-fold change in the desired frequency (see Table 4) could not resolve the issue for the existing methods."*

---

### Meta-Review · Area_Chair_JUGc · 2022-08-26

**Recommendation:** Accept
**Confidence:** Certain

**Metareview:**

The paper addresses the problem of long-horizon policy learning with non-local rewards via PG that avoids noise injection. The reviewers are in consensus, and I concur, that this paper would be a welcome contribution to NeurIPS because it addresses an important problem, makes a solid contribution, and shows impressive results. The authors' rebuttal sufficiently addressed putting more emphasis on the long-horizon policy learning over the deterministic setting and discussion of the limitations.

If possible, it would be a nice addition to add a video of the robot experiments in the the final revision.

**Award:**

No

---

### Decision · Program_Chairs · 2022-09-14

Accept